# Hyperdimensional Probe: Decoding LLM Representations via Vector Symbolic Architectures

## Abstract

Despite their capabilities, Large Language Models (LLMs) remain opaque with limited understanding of their internal representations. Current interpretability methods either focus on input-oriented feature extraction, such as supervised probes and Sparse Autoencoders (SAEs), or on output distribution inspection, such as logit-oriented approaches. A full understanding of LLM vector spaces, however, requires integrating both perspectives, something existing approaches struggle with due to constraints on latent feature definitions. We introduce the *Hyperdimensional Probe*, a hybrid supervised probe that combines symbolic representations with neural probing. Leveraging Vector Symbolic Architectures (VSAs) and hypervector algebra, it unifies prior methods: the top-down interpretability of supervised probes, SAE's sparsity-driven proxy space, and output-oriented logit investigation. This allows deeper input-focused feature extraction while supporting output-oriented investigation. Our experiments demonstrate that our method consistently extracts meaningful concepts across different LLMs, embedding sizes, and setups; uncovering concept-driven patterns in analogy-oriented inference and QA-focused text generation. By supporting joint input–output analysis, this work advances semantic understanding of neural representations while unifying the complementary perspectives of prior methods.

FIX

## 1 Introduction

The black-box nature of LLMs restricts the interpretability of their internal representations, motivating efforts to extract human-interpretable features from their vector spaces (Park et al., 2023). Such efforts typically fall into two probing perspectives (Ghandeharioun et al., 2024): (1) *Output Distribution Inspection*, and (2) *Feature Extraction*. These serve different yet complementary goals: feature-extraction methods target latent input features, whereas output-oriented inspection examines misalignment between a model's internal state and its textual outputs, a task also known as Eliciting Latent Knowledge (ELK). Comprehensive interpretability of LLM vector spaces would however require combining both, revealing high-level input features that may underlie such misalignment.

FIX

Probing approaches for output distribution inspection, such as lens-style (Nostalgebraist, 2020; Belrose et al., 2023) and patching-based methods (Ghandeharioun et al., 2024), project model's embeddings space into its output vocabulary space, *limiting the inspection to layer-wise projected next-token predictions* in ELK-centered, output-constrained settings. In contrast, input-oriented feature extraction is currently performed using Sparse Autoencoders (SAEs), and classification-oriented supervised probes. The supervised probes map the embeddings space into latent features to assess how much information about them is encoded using a top-down probing strategy (Gurnee & Tegmark, 2023; Marks & Tegmark, 2023; Diego Simon et al., 2024). This mapping paradigm faces two key challenges, as its probes are: *tailored to task-specific features*, limiting generalizability; and *trained directly on the final features*, making it challenging to distinguish information decoding from probe learning. On the other hand, SAEs provide general-purpose feature extraction via a bottom-up probing strategy on a layer-localized proxy space. This, however, reveals an *unbounded feature space* that requires *naming* and *filtering* for interpretability. Meanwhile, comprehensive interpretability of neural embeddings would require integrating *input-driven feature extraction with output-oriented analysis* to identify the internally represented input–output features during LLM inference.

FIX

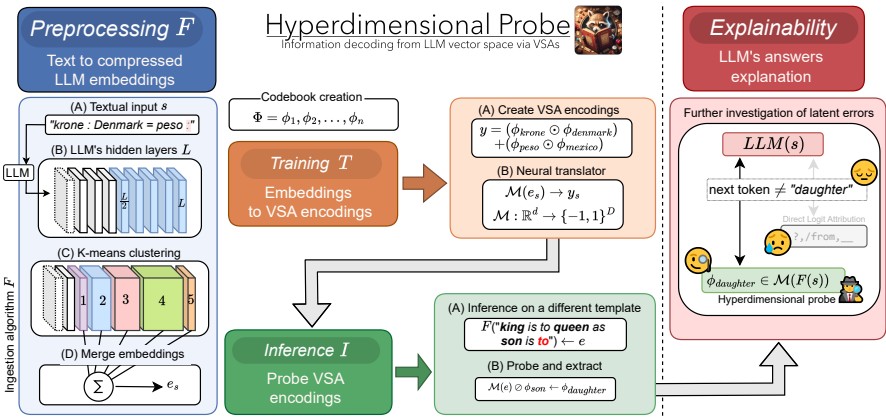

Figure 1: We first compress model's internal state using dimensionality-reduction steps on LLM embeddings ($F$, blue). Next, we train a neural VSA encoder to map these compressed neural embeddings into a bounded proxy space: VSA encodings ($T$, orange). Afterwards, we perform concept extraction by querying this proxy space using hypervector algebra ($I$, green). Lastly, extracted concepts ultimately enables deeper analysis on model's internal state and its outputs (red).

Our work uses input-oriented feature extraction for output-oriented analysis, examining which concepts are encoded in LLM vector space across two settings: (1) completing analogy-style inputs, and (2) QA-oriented text generation. We address key questions about LLM reasoning: whether models represent all in-context concepts, key–value associations, or only the output concept (RQ1–RQ3), and whether concept-oriented patterns emerge across input types (RQ4) and across diverse LLMs (RQ5; Section 5.2). During text generation, we further investigate latent patterns by uncovering how many question- and answer-related concepts appear before and after generation (RQ6; Section 6).

Unfortunately, *existing neural-embedding interpretation methods restrict this two-sided investigation*, due to their constraints on how latent features are defined. Our work introduces *Hyperdimensional Probe*, a novel probing paradigm that combines ideas from symbolic representations and neural probing. Inspired by the sparsity constraints of Sparse Autoencoder (SAEs), we exploits properties of Vector Symbolic Architectures (VSAs) and hypervector algebra to overcome major constraints in prior analysis. Functioning as a *hybrid supervised probe*, our method combines the bounded, interpretable feature space of standard supervised probes with the queryable proxy space of SAE-based methods. It also removes the single-token representation constraint of conventional output-oriented inspection, enabling feature sets with unrestricted abstraction and symbol sources.

Figure 1 shows our framework, from embedding processing, and neural VSA encoder training, to output-oriented inspection. Our work presents both methodological and experimental contributions:

- *Vector Symbolic Architectures for neural probing*: latent features of LLM vector space can be expressed through VSA encodings and hypervector algebra across diverse LLMs;
- *Hyperdimensional probe*: a novel probing paradigm that performs input-aligned feature extraction while supporting output-oriented interpretability (Section 4);
- *Effective compression of LLM embeddings*: enables probing the full residual stream (Section 4.3), removing the need for the layer-selection stage common in earlier work;
- *Concept-oriented insights into LLM inference*: Section 5.2 reveals differences in conceptual richness across inputs and models, while Section 6 exposes patterns during text generation.

## 2 RELATED WORK

The latent representations of transformers, also known as residual stream, is a high-dimensional linear vector space that aggregates the outputs of all hidden layers (Elhage et al., 2021). In recent years, three main approaches have been used to study the features encoded in this vector space (Ferrando et al., 2024): (1) supervised probes, (2) SAEs for input-focused feature extraction, and (3) logit-based methods for output distribution inspection.

**Supervised Probes** is a generic mapping paradigm that maps neural embeddings to task-relevant input features, measuring how much information about them is embedded (Tenney et al., 2019).

Conventional probes are tailored and trained directly on task-specific features, from syntactical information (Hernández López et al., 2023; Diego Simon et al., 2024), to space-time coordinates (Gurnee & Tegmark, 2023) and truthfulness (Marks & Tegmark, 2023). However, *tailoring probes* to specific tasks limits generalizability, whereas *directly learning targeted features* complicates distinguishing actual information decoding from probe-induced learning (Hewitt & Liang, 2019).

**Sparse AutoEncoders (SAEs)** provide input-oriented feature extraction using a proxy space learned via sparse dictionary learning (Olshausen & Field, 1997), uncovering superposed latent features (Cunningham et al., 2023). An autoencoder reconstructs the residual stream in an unsupervised fashion, enforcing sparsity in its learned representations. Once trained, these serve as a proxy layer for analysis. SAE activated neurons are interpreted via two strategies: identify representative tokens via logit-based methods (Kissane et al., 2024; Dunefsky et al., 2024); clustering inputs by shared SAE neurons, followed by manual (Jing et al., 2025) or automatic (Bricken et al., 2023; Lieberum et al., 2024) feature naming. The shortcomings of SAEs that hinder output-oriented investigation are threefold: (1) their *unbounded feature space* yields set of latent features that are difficult to control and align; (2) their *feature-naming* process restricts semantic grounding to either output-vocabulary labels or ambiguous, data-dependent descriptors; and (3) their dependence on *layer-localized representations* requires identifying a task-specific optimal hidden layer.

**Logit Attribution** focuses on output distribution inspection by projecting model's embedding space onto its output vocabulary through either (1) the model's unembedding matrix, also known as Direct Logit Attribution (DLA; Logit Lens by Nostalgebraist, 2020); (2) learned affine transformation (Tuned Lens by Belrose et al., 2023); or (3) patched LLM inference (Patchscope by Ghandeharioun et al., 2024). This logit-based paradigm offers insight into the output distribution (Jastrzebski et al., 2017) by generating projected logits at a chosen point in the forward pass, revealing next-token predictions under the assumption that all subsequent layers are bypassed. However, logit-attribution approaches hinder input-aligned feature extraction by (1) relying sorely on token-level surface features, which constrains *probing to ELK-style, output-aligned analysis*, and (2) restricting features to single-token representations, *limiting semantic-oriented analysis of higher-level abstractions*.

Functioning as a hybrid supervised probe, *Hyperdimensional Probe* exploits VSAs and hypervector algebra to unify the diverse perspectives of prior methods. Our novel paradigm integrates (1) the *top-down interpretability* of conventional probes, (2) the SAE's ability to learn a *sparsity-driven proxy space*, and (3) a higher-level, *jointly input–output analytic perspective* that goes beyond conventional logit-based methods. By leveraging VSA-based representations (Section 3), we avoid the unbounded feature spaces of SAEs and the need for post-hoc feature naming and filtering. In addition, our embedding-ingestion algorithm (Section 4.3) *removes the requirement to identify an optimal hidden layer of prior methods*. Our approach also mitigates the dichotomy between information decoding and probe-induced learning by learning a transformation from LLM embeddings to a controlled proxy space (Section 4.4) rather than predicting latent features directly. Lastly, it addresses the core limitations of conventional output distribution investigations by supporting feature sets of arbitrary abstraction, cardinality, and symbolic origin. Thus, our novel methodology combines the strengths of prior approaches for joint input-output feature extraction with deeper output-oriented analysis by:

1. *defining a compositional bounded feature space* (probes) rather than unbounded (SAEs);

2. *learning a sparse proxy space* (SAEs) rather than targeting task-specific feature (probes);

3. *querying a proxy space* holistically (SAEs), in contrast to classification-based probes;

4. inspecting the LLM vector space *without introducing layer selection* (probes, logit, SAEs);

5. *targeting concept-oriented latent features* (VSAs) rather than token-aligned logit features.

## 3 BACKGROUND

Vector Symbolic Architectures (VSAs; Hyperdimensional Computing) represent entities as random points in a high-dimensional space. Leveraging the concentration of measure phenomenon (Ledoux, 2001; Kanerva, 2009), exponentially many distinct concepts can be represented as nearly orthogonal random vectors. A codebook $\Phi$ maps a predefined set of concepts to their hypervector, while orthogonality and hypervector operations allow composition into more complex representations.

**VSA codebook.** We adopt the Multiply-Add-Permute architecture (MAP-Bipolar, MAP-B) from VSAs (Schlegel et al., 2022; Gayler, 1998), using bipolar hypervectors in $-1, 1^D$. Dimensionality $D$, typically $10^2$–$10^4$, depends on the number of concepts (Kanerva, 1988) and representation

complexity. MAP-B can theoretically encode $2^D$ orthogonal, independent elements (Schlegel et al., 2022). Its codebook $\Phi \in {-1, 1}^{n_c \times D}$ stores $n_c$ atomic concepts as bipolar random vectors, generated deterministically from seeds to ensure orthogonality and independence. Each vector is associated with a concept, and $\Phi$ enables evaluation of representations by comparing them with known vectors. Since MAP-B operates in the bipolar domain, cosine similarity is used (Schlegel et al., 2022).

**Hypervector algebra.** The hypervector algebra (Kanerva, 2009) relies on two operations: *binding* and *bundling*, which support representing complex cognitive structures, such as textual propositions, in a *distributed*, *noise-tolerant* manner (Gayler, 1998; Kanerva, 2009). *Binding* operation ($\odot$) encodes input features with their associated values. For example, it can associate concepts with contextual information, such as (USA $\odot$ dollar). The *bundling* operation ($+$), or superposition, creates set of (contextualized) concepts by combining multiple concepts into one, such as (USA $+$ Mexico). The resulting bundled vector is by design similar to each of its constituents, enabling retrieval. Binding is obtained via Hadamard product (element-wise) while bundling is element-wise sum. Polarization (sign) is typically required after bundling (Kleyko et al., 2020) to maintain the bipolar domain. This process irreversibly blends the parts, diminishing their similarity to the originals in proportion to their number. Conversely, *unbinding* ($\oslash$) in VSAs recovers elemental vectors from a binding operation by factoring out one vector via multiplication with its inverse (itself in MAP-B).

# 4 HYPERDIMENSIONAL PROBE 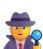

This section introduces our VSA-based framework for extracting information from the neural embeddings of LLMs on analogy-style completion tasks (RQ1–RQ5). Section 4.1 presents an analogy-style corpus as a controlled testbed for testing our method on inputs requiring varied reasoning. Section 4.2 explains how training examples are built using hypervector algebra. We next outline our three-stage pipeline: (1) processing embeddings (Section 4.3, $F$ in Figure 1); (2) mapping them into a controlled proxy space via a neural VSA encoder, producing VSA encodings (Section 4.4, $T$); and (3) extracting concepts from these encodings (Section 4.5, $I$). Finally, Section 6 applies the methodology to text generation (RQ6), and Appendix H discusses its broader applicability.

## 4.1 SYNTHETIC CORPUS

We build a textual dataset to evaluate the core components of our approach within a controlled, and interpretable testbed. Using analogy-style tasks guides LLMs toward concepts and their relations, testing diverse reasoning from syntactic patterns and key-value associations to abstract inference.

**Knowledge bases.** This work focuses on analogies, textual inputs representing pairs of concepts connected by the same type of factual, syntactic, or semantic relationship. We collect pairs of analogies from two knowledge bases: Google analogy test set (Mikolov, 2013), and the Bigger Analogy Test Set (BATS, (Gladkova et al., 2016)). These span 44 domains across five distinct categories, covering a wide range of factual and linguistic relationships, including analogies related to factual knowledge (e.g., a country's currency), semantic relations (e.g., grammatical gender), and morphological modifiers (e.g., verb+men). We also design mathematical analogies using three-digit integers and basic operations such as doubling, cubing, division, and extraction of roots.

**Textual analogies.** After collecting these pairs, we generate 114,099 distinct textual examples, denoted as $\mathcal{S}$, by combining all possible domain pairings. Each training example is formatted as:

$$\texttt{a}_1 \; : \; \texttt{a}_2 \; = \; \texttt{b}_1 \; : \; \texttt{b}_2 \tag{1}$$

where $\texttt{a}_1$ and $\texttt{b}_1$ represent the keys of the two pairs, and $\texttt{a}_2$ and $\texttt{b}_2$ are their corresponding values. For example, `Denmark:krone = Mexico:peso` for the countries currencies, and `queen:king = mother:father` for the grammatical gender. Table 11 and Table 12 in Appendix M show the domains grouped by knowledge base and category, respectively. Some concepts span multiple domains, such as `Australia` links to `Canberra`, `English`, and `Australian`. These overlaps can help mitigate the confounding effect of memorizing key-value pairs. For our experiments in Section 5, we further limit confounding effects by using the same pairs but generating a set of textual inputs ($\bar{\mathcal{S}}$) with a verbose template: $\texttt{a}_1$ is to $\texttt{a}_2$ as $\texttt{b}_1$ is to $\texttt{b}_2$. Conversely, for training (Section 4.4), we apply data augmentation strategies on $\mathcal{S}$, such as key-value swapping, effectively tripling the corpus size which results in 395,944 training inputs (Appendix M).

## 4.2 TRAINING EXAMPLES

This section describes the process of constructing VSA-based representations for training input. This procedure, illustrated with our textual templates (Equation 1), generalizes to other templates

(e.g., question-answer) or tasks (e.g., toxicity detection; Appendix H) since VSAs and hypervector algebra can encode complex structures across diverse inputs.

**Codebook construction.** The codebook defines the set of all input features; in our case, the contextually relevant concept, and is later used to construct and query VSA encodings. In our controlled setting, the codebook $\Phi$ (feature set) is constructed directly using all unique words included in the corpus, such as: $\texttt{mexico} \rightarrow \phi_{mexico} \in \Phi$, and $\texttt{krone} \rightarrow \phi_{krone} \in \Phi$. Thus, we create a matrix $\Phi \in \{-1, 1\}^{n_c \times D}$, using the $\texttt{torch-hd}$ library (Heddes et al., 2023), where $D$ is the VSA dimension and $n_c = 2,996$ is the number of concepts/features. We set $D = 4096$ as an adequate hidden dimension, given the cardinality of our codebook ($\approx 10^3$), which remains well below the theoretical capacity limit of the MAP-B architecture (Section 3). The average pairwise cosine similarity of the concepts in the codebook is $0 \pm 0.02$, confirming orthogonality (full distribution in Appendix J).

**VSA encodings.** With well-structured textual inputs, extracting input features and building their VSA-based representation is straightforward. Scalability to other input types is addressed in Appendix H.1. For each training input $s \in \mathcal{S}$, we generate its encoding by exploiting its constructive words (Equation 1), retrieving their corresponding hypervectors: $\{\phi_{a_1}, \phi_{a_2}, \phi_{b_1}, \phi_{b_2}\} \subset \Phi$. To encode an input sentence, we then exploit hypervector operations: binding and bundling (Section 3). Given that the input template represents two conceptual key–value pairs, we first bind each key to its corresponding value, such as linking each country to its currency in Equation 2. The full text is then encoded through bundling, producing a superposed set of contextualized concepts represented as key–value associations. Ultimately, we polarize it, with the sign function, to maintain the bipolar domain. The input encoding in VSA for a given sentence is then computed as:

$$y_s = (\phi_{\text{key}} \odot \phi_{\text{value}}) + (\phi_{\text{key}} \odot \phi_{\text{value}}) + \ldots = (\phi_{a_1} \odot \phi_{a_2}) + (\phi_{b_1} \odot \phi_{b_2}) \tag{2}$$
$$\text{``$\sqcup$Denmark$\sqcup$:$\sqcup$krone$\sqcup$=$\sqcup$Mexico$\sqcup$:$\sqcup$peso''} \mapsto (\phi_{\text{denmark}} \odot \phi_{\text{krone}}) + (\phi_{\text{mexico}} \odot \phi_{\text{peso}})$$

### 4.3 Processing neural embeddings $F$

FIX

The first stage of our pipeline involves feeding textual inputs to an autoregressive transformer model, followed by obtaining and preprocessing its residual stream ($F$ in Figure 1). Using our corpus, we prompt an LLM with an input sentence $s \in \mathcal{S}$, $LLM(s)$. For each textual input, its final word ($b_2$) is removed beforehand as it represents the value of the second analogy, our target concept.

**Caching token embeddings.** Our probing goal is to *inspect the complete internal state of a language model prior to its textual output*, capturing all encoded concepts without assuming their relation to the output. To this end, we examine the residual stream in the final token representation, focusing on the middle to last layers. Emerging evidence shows that transformers encode next-token information in the final token due to their autoregressive nature (Elhage et al., 2021; Olsson et al., 2022), refining it in later residual stream layers (Belrose et al., 2023; Hernandez et al., 2023). Specifically, for an autoregressive language model with $L$ hidden layers, we consider the embeddings (with size $d$) of the last token ("`:`") in the latter half, for all $l \in [L/2, \ldots, L]$, yielding a matrix in $\mathbb{R}^{L/2 \times d}$.

However, considering such a wide range of layers presents a computational challenge, as probing a high-dimensional matrix can significantly increase the computational footprint of the probing pipeline. Further, adjacent layer-wise embeddings are highly correlated (0.9) as shown in Appendix O.1, likely encoding redundant numerical patterns, and thus similar information. Here, we define representation redundancy as the approximate linear dependence among LLM hidden layer embeddings. Appendix O.2 shows that the LLM embedding space is roughly low-rank, with only a few rows/layers (or their combinations) contribute meaningful structure.

**Dimensionality reduction.** To reduce the computational cost of our approach, we lower the input dimensionality for our encoder by introducing two dimensionality-reduction steps: *k-means clustering* (Jain & Dubes, 1988), and *sum pooling*. Clustering reduces representation redundancy by grouping similar vector regions in LLM embedding space and computing centroids, accomplishing knowledge distillation. To determine the optimal range for $k$, we adopt the silhouette score (Rousseeuw, 1987). A trade-off between reduction, granularity, and model variability emerges with 3–7 clusters (Appendix O.3). We set $k = 5$ to maintain the essential data structure while supporting effective dimensionality reduction.[1] We then apply sum pooling, which consists of summing all centroid

---

[1]Appendix O.4 shows that the clusters consistently group adjacent layers.

embeddings;[2] merging group representatives (k-dimensional matrix) into a vector exploiting the additivity property of LLM embeddings demonstrated in previous work (Bronzini et al., 2024). For example, these reduction steps allows us to downsize the probed embedding space of OLMo-2:

$$\mathbb{R}^{33 \times 5120} \to \mathbb{R}^{5120}.$$

Appendix O.5 presents an ablation showing that skipping these two compression steps increases the encoder's trainable parameters tenfold. In summary, the neural representation of a textual input from a language model is processed through the ingestion procedure $F$, as summarized in Algorithm 1.

### 4.4 Neural VSA encoder $T$

We train a supervised model to map token embeddings from an autoregressive transformer into VSA encodings with a known representation ($T$ in Figure 1). We define a supervised regression model $\mathcal{M}$, a shallow feedforward neural network, to map the LLM vector space to bipolar hypervectors. The model $\mathcal{M}$ is trained on the LLM-VSA dataset generated using the corpus $\mathcal{S}$ (Section 4.1), which consists of paired LLM embeddings ($e_s$ in Algorithm 1) and their corresponding VSA representations ($y_s$ in Equation 2). The model infers latent features from the unknown LLM vector space to translate the encoded semantics into VSA representations with explicit and interpretable semantics. We define the *neural VSA encoder* model $\mathcal{M}$ as a three-layer MLP with 55M–71M parameters (depending on the input embedding size $d$; see Appendix C), performing a non-linear transformation:

$$\mathcal{M} : \mathbb{R}^d \to \{-1, 1\}^D, \quad e_s \to y_s. \tag{3}$$

We use the hyperbolic tangent function (`tanh`) in the output layer for bipolar outputs and incorporate residual connections to enhance training stability and convergence. The training process minimizes the Binary Cross-Entropy (BCE) error between the bipolar target hypervectors and the predictions. To ensure compatibility with such binary loss function, targets are temporarily converted to binary based on their sign; and predictions are smoothly mapped to the range [0, 1] using the sigmoid function. A Mean Squared Error (MSE) regularization term is added to the loss, with a coefficient of $0.1$.[3] Implementation details for the training process are reported in Appendix D.1.

**Language models.** We validate our methodology on embeddings from popular open-weight LLMs available on Hugging Face with 355M-109B parameters, experimenting with different embedding sizes and layer counts. In particular, we test the latest Meta AI's Llama 4 Scout, (AI, 2025) a multi-modal mixture of 16 experts (MoE), Llama 3.1-8B (Grattafiori et al., 2024), Microsoft's Phi-4 (Abdin et al., 2024), EleutherAI's Pythia-1.4b (Biderman et al., 2023), AllenAI's OLMo-2-32B (OLMo et al., 2024), and OpenAI's legacy GPT-2-medium (Radford et al., 2019).   **FIX**

**Performance.** The LLM-VSA dataset uses a random 70-15-15 split of $\mathcal{S}$ for training, validation, and test sets. Since our setting can be interpreted both as a vector-based regression task and a multi-label classification problem,[4] we evaluate our approach using two distinct metrics: cosine similarity and multi-label binary accuracy. For binary accuracy, targets and predictions are binarized based on sign. First, evaluating the cosine similarity between the predicted and target VSA encodings yields a test-set average score of $0.89$ (best LLM in Appendix D, Llama 3.1-8B), indicating strong numerical alignment between our encoder's outputs and the target encodings. Second, we obtain an average binary accuracy of $0.94$, which indicates robust classification accuracy after polarizing the predictions with the sign function. This means that on average, the VSA encodings produced by our trained model deviated from the targets by only $6\%$ of the vector elements, a negligible error given VSA's large tolerance to noise. All tested models exhibit consistent performance;[5] layer count has no effect, whereas reducing the embedding dimension is found to be slightly detrimental. These results support our empirical hypothesis that VSA-based representations (MAP-B) and hypervector algebra can faithfully capture the latent features encoded in the neural embedding of LLMs (RQ7).   **FIX**

### 4.5 Probing VSA encodings $I$

In the third, and experimental stage of our pipeline ($I$ in Figure 1, Section 5), we examine the VSA encodings produced by our trained neural VSA encoder $\mathcal{M}$, extracting the embedded concepts. To retrieve the embedded atomic concepts, we use the *unbinding* operation from VSA algebra ($\oslash$,

---

[2]Preliminary evidence suggests that directly summing all layers (up to 33) results in a noisier representation.

[3]Empirical results demonstrated better performance than other coefficients tested, ranging from 0.01 to 1.

[4]VSA encodings can be viewed as vectors with $D$ distinct labels, each assuming one of two possible values.

[5]Appendix D reports the training performance of our neural VSA encoder $\mathcal{M}$ for all of the six models.

Section 3). This vector operation reverses binding, which in our case links a pair's key with its corresponding value, enabling one vector to be extracted from another. Since the generated VSA encoding may encode either no or several concepts, we attempt to extract the target concept ($b_2$) by dynamically testing the unbinding operation with various candidates.

*This concept-related flexibility represents the novelty and added value of VSA-based probing*, allowing us to query our proxy space without prior assumptions on the number of concepts. Consequently, we distinguish between two scenarios: in the first, no unbinding operations are required when the model encodes none or a single concept; in the second scenario, when multiple concepts are embedded, we test the unbinding operation with different concepts to isolate a single one. For example, unbinding a VSA encoding with the concept of Mexico and obtaining Peso suggests that the probed encoding originally incorporated both the key and value of the target analogy pair:

$$
\begin{aligned}
\text{LET} \quad & s := \text{"Denmark:krone=Mexico:"} \mapsto \text{"peso"} \\
\text{COMPUTE} \quad & y_s = \mathcal{M}(F(s)) \\
\text{QUERY} \quad & y_s \oslash \phi_{\text{mexico}} = \phi_{\text{peso}} + \text{noise} \\
\text{THEN} \quad & y_s \approx (\phi_{\text{mexico}} \odot \phi_{\text{peso}})
\end{aligned}
\tag{4}
$$

When probing an encoding ($y_s$ in Equation 4), we pick in-context concepts ($\phi_{\text{denmark}}$, $\phi_{\text{krone}}$, and $\phi_{\text{mexico}}$), and their combinations, as candidates for unbinding. The best candidate was chosen by benchmarking the resulting concept after unbinding, against the in-context and target concepts through cosine similarity with a threshold for concept detection equal to 0.1. Empirical tests show relevant concepts exceeded this low threshold, while noisy ones remain below, likely due to VSA's noise tolerance. If no concepts were detected, unbinding was skipped. In the experiments reported in Section 5, 80% of unbinding operations, averaged across all models, relied on the key of the target pair. In contrast, no operation was applied in 12% of the cases. Appendix E shows the proportions of other candidates and highlights the variations among models. Meanwhile, the *unrelated baseline in Appendix F* shows results of unbinding with concept candidates that do not relate to the input.

## 5 EXPERIMENTS ON INPUT-COMPLETION TASKS

This section explores the model's internal state when completing analogy-style inputs (RQ1–RQ5). We begin by outlining the analogy-oriented setup in Section 5.1. Next, we present our findings on concept extraction using the proposed probe in Section 5.2. Lastly, Section 5.2 compares these results with traditional logit-based investigation, which exhibits more superficial probing capabilities, likely because it depends on the model's vocabulary space and, thus, token-level latent features.

### 5.1 EXPERIMENTAL SETUP

**Data.** We test our trained neural VSA encoders $\mathcal{M}$ on the set of textual inputs formatted using the verbose template (Section 4.1). Thus, we validate our methodology using inputs with syntactic structures that differ from those seen during the training stage ($\bar{\mathcal{S}} \neq \mathcal{S}$). Therefore, we also perform information decoding from the vector representation of a different token, shifting from the colon token of the training examples (Equation 1) to the token to in $\bar{\mathcal{S}}$. This further mitigates confounding effects caused by probe-induced learning, more pronounced in conventional supervised probes.

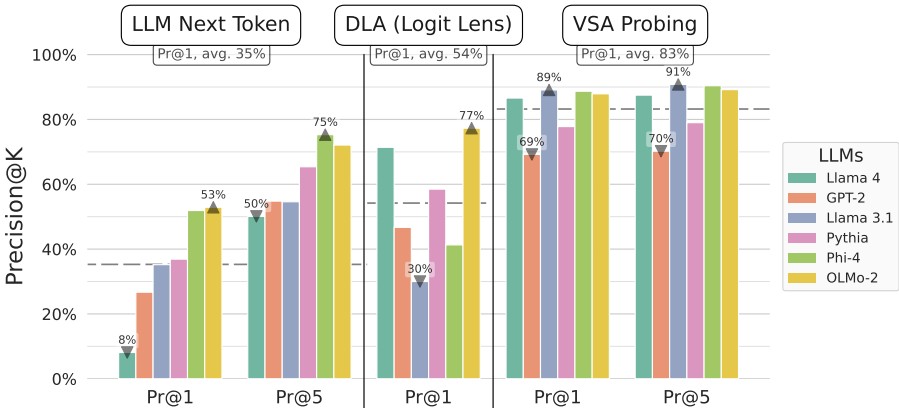

Figure 2: LLM performance on analogy completion with targets (left) and our decoding method's ability to extract targets from LLM embeddings (right), with logit-based results shown in the middle.

Table 1: **Concepts extracted by hyperdimensional probe.** `Key|Target` indicates extraction of the key ($b_1$) and value ($b_2$) of the target analogy; `Key` for only $b_1$. `Example` refers to $a_1$ and $a_2$, the in-context example's concepts; `Context|Target` for all concepts. `Out-of-context` indicates concepts unrelated to input, `Key Values` refer to analogy keys originating from a different domain. `NONE` means no concepts. *Our probe captures target-aligned combinations in roughly 80% of cases.*

| Extracted Concepts (%) | GPT-2 | Pythia | Llama 3.1 | Phi-4 | OLMo-2 | Llama4, Scout | AVERAGE |
|---|---|---|---|---|---|---|---|
| Key \| **Target** | 60.0 | 66.9 | **85.4** | 84.8 | 80.1 | 79.0 | 76.0 ± 9.4 |
| NONE | **21.9** | 16.7 | 6.9 | 7.6 | 8.5 | 11.5 | 12.2 ± 5.4 |
| Key | 4.5 | **6.1** | 0.6 | 1.0 | 1.8 | 3.4 | 2.9 ± 2.0 |
| Example | **5.8** | 2.4 | 1.5 | 1.3 | 2.0 | 0.8 | 2.3 ± 1.6 |
| Context \| **Target** | 1.1 | 1.9 | 1.5 | 1.2 | **4.4** | 2.4 | 2.1 ± 1.1 |
| Key \| Key Values | 1.3 | 1.4 | **1.8** | 1.5 | 1.1 | 0.8 | 1.3 ± 0.3 |
| Out-of-context | **1.6** | 1.2 | 0.5 | 0.7 | 0.8 | 1.1 | 1.0 ± 0.4 |
| Example Value \| Key Values | **1.5** | 1.3 | 0.3 | 0.0 | 0.2 | 0.1 | 0.6 ± 0.6 |
| Key Values \| **Target** | **0.4** | 0.1 | 0.3 | **0.4** | 0.1 | 0.1 | 0.2 ± 0.1 |
| **Target** | 0.1 | 0.1 | 0.1 | **0.2** | 0.1 | 0.1 | 0.1 ± 0.0 |

**Metrics.** Our experimental evaluation has a two-fold objective (Equation 4): we assess the performance of LLMs in the next-token prediction, and our VSA-based probing method for retrieving targets from their latent representations using *precision@k*. We measure the LLM's performance via: binary precision on the next-token prediction against the target word; softmax score of the most likely next token and the target one; and rank of the target token on the ordered softmax scores. Downstream performance of LLMs is measured using the most likely next token (*next-token@1*) and the top-5 tokens (*next-token@5*), capturing uncertainty in the model's softmax distribution. To evaluate the performance of our VSA-based probing approach, we assess the binary precision of retrieving the target VSA concept from LLM latent representations via *probing@1*, and *probing@5*.

FIX

### 5.2 CONCEPT EXTRACTION

FIX

Figure 2 reports VSA-based target concept retrieval, LLM downstream performance, and logit-based comparison. Appendix F tabulates the same results with variability and two validation tests. Table 1 presents the final results of VSA probing, with each row showing the unique concept sets produced by the extraction procedure of Section 4.5. Each set includes the concepts used for unbinding and those identified afterward; $\phi_{\text{mexico}} \mapsto$ `Key`, and $\phi_{\text{peso}} \mapsto$ `Target` for the example within Equation 4.

NEW

**High variability in LLMs' next-token predictions.** In an unexpected contrast, the largest model evaluated (109B; Llama 4, Scout) exhibited the lowest precision@1 in the next-token prediction task (8%, Figure 2), even underperforming the legacy GPT-2. Yet, its next-token@5 was comparable to others (still the lowest), but ranked among the best in probing@1. *Strong probing performance suggests the final state encodes the target concept, but the model often fails to output it*. This might be caused by exogenous (e.g., prompt design) and endogenous factors (e.g., tokenization). As shown in Appendix F.3, the model frequently predicted a space instead of the correct word, which still often appeared in its top five predictions. This emphasizes variability introduced by prompt design and tokenization, which might have greater impact on logit-based methods such as DLA.

FIX

**VSA probing exposes varying conceptual richness (RQ1–RQ5).** Regarding the concepts extracted by our hyperdimensional probe, we achieve an average probing@1 across all models equal to 83% (right side of Figure 2, Appendix F), extracting the target concept with its key for most cases (60% for GPT-2, 85% for Llama 3.1, Table 1). Notably, GPT-2 shows the highest proportion of cases with no concepts extracted (22%) followed by cases extracting only the in-context example concepts (6%), and ranks second in extracting only the target concept keys (5%). Thus, its internal states largely fall back on input-related concepts, *reflecting limited understanding needed to complete the analogy-style input correctly.* On the other hand, OLMo-2 has the highest proportion of instances in which our probing approach retrieves the target concept alongside all in-context concepts (Context | Target, 4%), *indicating its richer representation in its final state for both the input context and next word*. This latent richness is then reflected in its performance on next-token prediction, achieving the highest next token@1 equal to 48% (Figure 2). In cases where the target word was not among the top five predictions of Llama 4, nearly 50% of the instances (Appendix L.1; 28% for OLMo-2 in Appendix L.2), our probing method successfully extracted the target concept and its associated key in 70% of instances, while no concept was retrieved in 18% of cases (26% for OLMo-2). Although the first outcome supports previous observations, the absence of extracted concepts merits a more

FIX

FIX

FIX

Table 2: **Concepts extracted though VSA-based probing when DLA yields no concepts.** The table highlights VSA can also capture model's variability (e.g., in-context concepts, target concepts). The table highlights key shared items across models, covering nearly 98% of all extracted concepts.

| | GPT-2 | Pythia | Llama 3.1 | Phi-4 | OLMo-2 | Llama 4, Scout | AVERAGE |
|---|---|---|---|---|---|---|---|
| NONE: no concepts from DLA (%; $\mathcal{D} \subset \bar{\mathcal{S}}$) | 33.9 | 32.8 | **47.4** | 33.1 | 14.6 | 15.4 | $29.5 \pm 11.4$ |
| Concepts extracted from $\mathcal{D}$ by VSA (%) | | | | | | | |
| Key \| **Target** | 53.5 | 56.5 | **76.6** | 70.5 | 44.5 | 42.6 | $57.4 \pm 12.5$ |
| NONE | 26.8 | 24.3 | 13.7 | 18.6 | 41.0 | **43.2** | $27.9 \pm 10.9$ |
| Example | **6.8** | 3.0 | 2.1 | 2.1 | 3.4 | 2.0 | $3.2 \pm 1.7$ |
| Key | 5.2 | **7.3** | 0.7 | 0.9 | 1.6 | 3.0 | $3.1 \pm 2.4$ |
| Out-of-context | 2.0 | 1.8 | 1.1 | 1.9 | 3.8 | **4.5** | $2.5 \pm 1.2$ |
| Key \| Pair Values | 1.8 | 2.1 | 2.5 | **3.2** | 2.4 | 1.8 | $2.3 \pm 0.5$ |
| Context \| **Target** | 0.7 | 1.2 | 1.1 | 0.7 | **1.9** | 1.4 | $1.2 \pm 0.4$ |
| **Target** | 0.1 | 0.1 | **0.2** | 0.0 | 0.1 | 0.0 | $0.1 \pm 0.1$ |

granular analysis across analogy categories (RQ4; see also Table 12 in Appendix M). Our probe most frequently encounters conceptually-empty representations in mathematical analogies (88%, Appendix F.2, also for OLMo-2 comparison), followed by semantic hierarchies (39%). Factual and morphological analogies show much lower rates, at 5.5% and 1.1%, respectively. As elaborated in Appendix F.2, these *differences likely stem from the type of reasoning involved*: linguistic analogies depend on syntactical patterns, factual and semantic relations on key–value associations, and hierarchies or mathematical analogies on abstract inference.

### 5.3 COMPARISON WITH LOGIT ATTRIBUTION

For validation, we compare with conventional output distribution inspection. Specifically, we apply DLA (Logit Lens; Section 2) to all models using $\bar{S}$ to highlight the limitations of token-aligned logit–based analysis. Appendix G provides details on our comparison choices and further analysis.

We use fuzzy token-to-concept matching with our concept set (e.g., "pes" $\mapsto$ `peso`), and consider projected next-token predictions from the model's middle to last layers (Appendix G.3) of the last token, as VSA probing. DLA produces no concepts in nearly 30% of analogies on average ($\mathcal{D}$ in Table 2; +17% compared to VSA in Table 1). It also exhibits lower performance and greater variability in target concept retrieval, with a similarly large drop in comparison to VSA-based precision@1 (Figure 2). In instances without concepts from DLA ($\mathcal{D}$ in Table 2), our VSA probing extracts, on average, the key-target pair in 57% of all analogies, while returning none for 28%. For instance, for the analogy `king is to queen as son is to` $\mapsto$ `daughter`, using OLMo-2, our probe extracts the key-target concepts (`son` and `daughter`), while DLA produces no concepts. The model predicts the next token prediction as ? with a softmax score of 0.06, followed by `father` (0.05); the target word has a rank of 37. Since conventional output inspections operate within the model's vocabulary space, *token-level logit analyses often produce shallower findings and show greater variability* caused likely by exogenous factors, such as prompt design, or endogenous ones.

## 6 FROM INPUT-COMPLETION TASKS TO TEXT GENERATION

Lastly, this section examines the model's internal state during text generation (RQ6). We apply our methodology (Section 4) to a question-answering scenario, which represents a more realistic setting than the previous controlled testbed, using the popular SQuAD dataset (Rajpurkar et al., 2016).

This dataset evaluates extractive QA, where each answer is a text span within a given context, through questions generated by crowdworkers over Wikipedia articles. This aligns with our concept-focused probing, as questions and answers target contextual concepts, allowing us to benchmark feature extraction against features derived from both. We extract question- and answer-related concepts grounded in lexical semantics by identifying nouns, verbs, and adjectives using WordNet (Miller, 1995) and DBpedia (Lehmann et al., 2015). Using bundling operations, we construct 693,886 training examples $\mathcal{Q}$ by progressively pairing SQuAD questions with their associated lexical features:

$$\text{"What was the \textit{name}"} \mapsto \phi_{\text{name}} \tag{5}$$

$$\text{"What was the \textit{name} of the \textit{ship}"} \mapsto (\phi_{\text{name}} + \phi_{\text{ship}})$$

Our trained encoder (Section 4.4) achieves a test-set cosine similarity of 0.44, and a binary accuracy of 0.70 for mapping embeddings of Llama 3.1 into these VSA encodings. For our probing experiments, we consider 10,000 sampled items $\bar{\mathcal{Q}}$ from SQuAD, each formatted as a question preceded

Figure 3: Concepts extracted *before* and *after* the LLM's text generation, with respect to *question* and *answer* features. Red denotes the subset of failure instances, while green the full sample $\bar{\mathcal{Q}}$.

by its contextual text. See Appendix I for details. We then apply our probe to the model's state both before and after text generation, comparing VSA encodings with the codebook $\Phi$ using cosine similarity with a concept detection threshold of 0.1, as unbinding is not required. Our probe extracts an average of three concepts before both the first and last token generation. The LLM's QA-related performance achieved an average token-based F1 score of 0.69 (95% CI: 0.68–0.70), an exact match of 0.52 (95% CI: 0.51–0.53), and 68% of outputs (95% CI: 67–69) contained the target answers.

**Observing concept drift in generation (RQ6).** We evaluate semantic-based concept relevance by computing cosine similarity between concept embeddings and question-answer features. The average similarity of extracted concepts related to the question decreases after text generation: by 4.8% for the entire sample and by 8.0% in the LLM error subset (32% of the sample), while no significant differences are observed prior to generation (Figure 3; left). For answer-related concepts, overall no change is observed before and after text generation, but the LLM error subset shows a slight increase (+3.2–3.5%; Figure 3, right). *This suggests that LLM failures may stem from losing focus on the question rather than from a lack of answer-related knowledge.* This hypothesis is supported by a weak positive Spearman correlation (0.2 with a p-value of $1e^{-99}$; Appendix K) between LLM's F1 score and the proportion of question-related concepts extracted after text generation. For example, for the SQuAD query "What do laboratories *try* to *produce hydrogen* from?" (target answer: "*solar energy* and *water*"), the model erroneously outputs "*water* and heat" (F1 = 0.57). Before the model's text generation, our proposed approach extracts the concepts $\phi_{try}$, $\phi_{produce}$, $\phi_{hydrogen}$ (question) and $\phi_{solar}$, $\phi_{water}$ (answer); after generation, the question-related concept set reduced to $\phi_{produce}$ and the answer set gained the concept $\phi_{energy}$. During text generation, the model lost the concept of hydrogen while refining concepts related to answering. This leads to a response that no longer focuses on the question's subject, but reflect the more general concepts still active in the model's internal state.

## 7 CONCLUSIONS

This work provides empirical evidence that the latent features of neural embeddings can be faithfully captured using VSA-based representations (RQ7; Section 4.4). Building on this, VSA probing enables a unified, concept-oriented analysis of features aligned with both inputs and outputs (RQ8; Section 5.2, Section 6); integrating the complementary viewpoints of prior approaches for probing.

Our method combines the top-down interpretability of supervised probes (Section 4.2), SAE's sparsity-driven proxy space (Section 4.4), and output-oriented investigation (Section 4.5) of logit-based methods. Additionally, our ingestion algorithm (Section 4.3) bypasses layer selection of conventional layer-wise approaches. It enables joint input–output feature extraction that uncovers non-trivial insights into neural embeddings. Section 5.2 shows differences in conceptual richness (RQ1–RQ3), including OLMo-2's richer structure compared to GPT-2 (RQ5), and the common sparsity in mathematical analogies (RQ4). Section 6 reveals concept-related patterns during text generation (RQ6); concept extraction before and after generation suggests that LLM failures may stem from losing focus on question-related concepts rather than answer-related ones.

Our methodology applies to any autoregressive model, is compatible with all Hugging Face models, and introduces a layer-agnostic, lightweight probe (Appendix Q). Moreover, combining VSA generality with hypervector algebra provides a promising way to investigate multimodal features in neural embeddings, as displayed in the proof of concept in Appendix P.

**Limitations.** The primary limitations of our work are its dependence on unidirectional transformation that prevents causal validation and on a set of predefined features (Appendix A). While we implemented multiple strategies to reduce confounding effects in probe-induced learning, such as evaluating on syntactically diverse inputs ($\bar{\mathcal{S}}$, and $\bar{\mathcal{Q}}$), we were unable to directly measure their effectiveness. Appendix F.1 shows two validation tests to assess confounding effects on decoding.

## REPRODUCIBILITY STATEMENT

The submission includes both the source code and our synthetic corpus, which will be made publicly available upon acceptance. A *README.md* file is provided with the code, containing detailed instructions to reproduce our methodology (Section 4) and experimental results (Section 5).

Section 4 presents our methodology, covering the entire pipeline from data creation (Section 4.1 and Section 4.2) to the training process of our proposed method (Section 4.3 and Section 4.4). Additional details of the training procedure are provided in Appendix D.1, while the overall model architecture is shown in Appendix C. The ingestion algorithm for LLM embeddings described in Section 4.3 is further illustrated in Appendix B. Finally, Appendix D.2 provides the Hugging Face links for all the LLMs used in our work, and Appendix Q reports the computational workload of our methodology.

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

## A  LIMITATIONS

While we apply data augmentation and test on syntactically different inputs to mitigate confounding on information decoding (Section 4.1 and Section 5.1), we could not measure the effectiveness of these strategies.

Table 1, Table 2 and Table 9 report on the actual concepts identified by our probing method. The label `Key values` denotes instances where the probe retrieves a key of an analogy pair with a concept linked to it in a different domain (see `Australia` in Section 4.1). This outcome can be viewed as an artifact of our probe, revealing the confounding influence of memorized key-value associations. Nevertheless, such cases constitute only a small fraction, 2% of the 114,099 textual inputs processed across all models, covering *Key — Key Values*, *Example Value — Key Values*, and *Key Values — Target* shown in Table 1. To further investigate potential confounding effects from probe learning, we introduce two control tests, as proposed in (Hewitt & Liang, 2019). Using randomly-permuted input embeddings ($e_s$) as a null model (Gotelli et al., 2012), and applying the unbinding operation on VSA encodings ($y_s$) with concept pairs unrelated to inputs, respectively, *permuted* and *unrelated baseline* in Appendix F.

While our approach avoids dependence on the LLM's vocabulary of DLA-based methods (Section 2) due to the data-agnostic nature of VSAs, it still requires a predefined set of concepts. This set can however be seen as an alphabet with no practical constraints on the cardinality, type and source of its symbols.

## B  ALGORITHM TO PROCESS LLM EMBEDDINGS AS DESCRIBED IN SECTION 4.3

---

**Algorithm 1:** Ingestion procedure $F$

**Data:** Textual sequence $s \in \mathcal{S}$

**Result:** Compressed model state for its next token prediction

**begin**

    // Get the residual stream from the language model
    $\mathbf{H} \leftarrow \text{LLM}(s) \in \mathbb{R}^{L \times T \times d}$ ;

    // Retain embeddings of the last token from the bottom half of the layers
    $\mathbf{H}^{\star} \leftarrow \mathbf{H}[L/2 : L, -1]$;

    // Apply K-Means clustering
    $\mathbf{C} \leftarrow \text{KMeans}_{K=5}(\mathbf{H}^{\star}) \in \mathbb{R}^{K \times d}$ ;

    // Sum pooling across the centroids
    $\mathbf{e}_s \leftarrow \sum_{k=1}^{5} \mathbf{C}_k \in \mathbb{R}^d$ ;

    **return** $\mathbf{e}_s$

---

# C  ARCHITECTURE OF OUR *Hyperdimensional probe*

Table 3: Configuration of the neural VSA encoder $\mathcal{M}$ for an input embedding dimension equal to $d$.

| Component | Input Dim | Output Dim | Note |
|---|---|---|---|
| **Input Layer** | | | |
| Linear Layer | $d$ | 4096 | - |
| Normalization | - | - | LayerNorm (4096) |
| Activation | - | - | GELU |
| **Residual Block 1** | | | |
| Linear Layer | 4096 | 4096 | GELU activation |
| Normalization | - | - | LayerNorm (4096) |
| Dropout | - | - | $p = 0.5$ |
| Residual Connection | - | - | Identity |
| **Residual Block 2** | | | |
| Linear Layer | 4096 | 4096 | GELU activation |
| Normalization | - | - | LayerNorm (4096) |
| Dropout | - | - | $p = 0.5$ |
| Residual Connection | - | - | Identity |
| **Output Layer** | | | |
| Normalization | - | - | LayerNorm (4096) |
| Linear Layer | 4096 | 4096 | - |
| Activation | - | - | Tanh |

Trainable parameters with: $d = 1024, 55M$
$d = 2048, 59M$
$d = 4096, 67M$
$d = 5120, 71M$

# D  TRAINING PERFORMANCE OF THE NEURAL VSA ENCODERS

Table 4: Training performance of our neural VSA encoder $\mathcal{M}$ on the test set. Order by model size.

| Large Language Model | | | | Cosine similarity | Binary accuracy |
|---|---|---|---|---|---|
| **Name** | **Parameters** | **Embedding dimension** | **Layers from residual stream** | | |
| Llama 4, Scout, 17B-16E | 109 B | 5120 | 24th to 48th \|25\| | 0.890 | 0.934 |
| OLMo-2 | 32 B | 5120 | 32nd to 64th \|33\| | 0.878 | 0.926 |
| Phi 4 | 14 B | 5120 | 20th to 40th \|21\| | 0.881 | 0.930 |
| Llama 3.1-8B | 8 B | 4096 | 16th to 32nd \|17\| | **0.892** | **0.937** |
| Pythia-1.4b | 1.4 B | 2048 | 12th to 24th \|13\| | 0.861 | 0.916 |
| GPT-2, medium | 355 M | 1024 | 12th to 24th \|13\| | 0.865 | 0.920 |
| | | | **AVERAGE** | 0.878 ± 0.01 | 0.927 ± 0.01 |

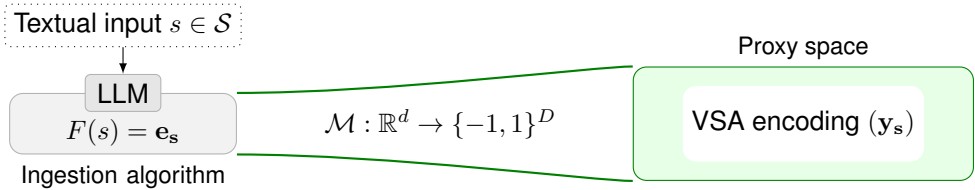

Figure 4: The regression model that maps the neural representations into a controlled vector space.

## D.1  TRAINING DETAILS

The neural VSA encoder $\mathcal{M}$ was trained for 421 epochs on average via PyTorch Lighting,[6] using early stopping (patient set at 100 epochs) and a batch size of 32. The optimal learning rate was automatically determined using the learning rate finder provided by the aforementioned library, and was approximately set to $3e^{-5}$ on average. We use *AdamW* as the optimizer (weight decay of $1e^{-4}$), applying a learning rate schedule based on Cosine Annealing with Warm Restarts, starting from the 100th epoch and doubling the restart period thereafter. To adapt the batch size after LR restarts, we employed a Gradient Accumulation Scheduler: the effective batch size was doubled at the 110th epoch, quadrupled at the 310th, and increased eightfold at the 410th epoch. During training, the model's outputs are dynamically binarized using the sigmoid function to ensure compatibility with the loss function (Section 4.4). This approach demonstrated better empirical performance than linear min-max normalization.

## D.2  HUGGING FACE REPOSITORIES FOR THE CONSIDERED LLMS

1. Meta AI's **Llama 4, Scout**, huggingface.co/meta-llama/Llama-4-Scout-17B-16E

2. Meta AI's **Llama 3.1**, huggingface.co/meta-llama/Llama-3.1-8B

3. Microsoft's **Phi-4**, huggingface.co/microsoft/phi-4

4. EleutherAI's **Pythia**, huggingface.co/eleutherai/pythia-1.4b

5. AllenAI's **OLMo-2**, huggingface.co/allenai/OLMo-2-0325-32B

6. OpenAI's **GPT-2**, huggingface.co/openai-community/gpt2-medium

---

[6]lightning.ai/docs/pytorch/stable

## E  UNBINDING STAGE FROM SECTION 4.5

Table 5: **Unbinding stage**: Proportions of the best unbinding concepts used for extracting concepts from VSA encodings across different models, with overall mean and standard deviation. Key refers to cases where the candidate concept corresponds to the key of the target pair ($b_1$), while NONE indicates that no unbinding operations were applied to the probed VSA encoding. Example denotes a concept where the key ($a_1$) and value ($a_2$) from the in-context example were pre-bound. Lastly, Context represents a scenario where the in-context example ($a_1, a_2$) was pre-bound together with the key of the target pair ($b_1$). On the other hand, Greedy means using a concept candidate from the vocabulary, rather than picking it among those of the input. The table has been trimmed to highlight the relevant and common items across the models. We consider the first four strategies to be the most relevant, as they account for 97% of all unbinding operations across models.

| Concept for unbinding (%) | GPT-2 | Pythia | Llama 4, Scout | OLMo-2 | Phi-4 | Llama 3.1 | AVERAGE |
|---|---|---|---|---|---|---|---|
| Key | 65.9 | 74.4 | 83.2 | 83.2 | 87.4 | **87.9** | 80.3 ± 7.8 |
| NONE | **22.0** | 16.9 | 11.6 | 8.6 | 7.7 | 7.0 | 12.3 ± 5.4 |
| Example Key | **6.0** | 2.6 | 1.0 | 2.1 | 1.5 | 1.7 | 2.5 ± 1.7 |
| Context | 1.2 | 2.0 | 2.6 | **4.5** | 1.3 | 1.5 | 2.2 ± 1.2 |
| Greedy | 2.1 | 1.9 | 1.3 | 0.9 | 1.2 | 0.9 | 1.4 ± 0.5 |
| Example Value | 1.6 | 1.5 | 1.0 | 0.4 | 0.7 | 0.5 | 0.9 ± 0.5 |
| Cleaned Example Key | 0.2 | 0.5 | 0.0 | 0.1 | 0.2 | 0.1 | 0.2 ± 0.2 |
| Cleaned Example Value | 0.9 | 0.1 | 0.0 | 0.1 | 0.1 | 0.1 | 0.2 ± 0.3 |
| Cleaned Key | 0.0 | 0.0 | 0.0 | 0.0 | 0.0 | 0.1 | 0.0 ± 0.0 |
| Cleaned Original | 0.0 | 0.0 | 0.0 | 0.0 | 0.0 | 0.0 | 0.0 ± 0.0 |
| Example | 0.0 | 0.0 | 0.0 | 0.0 | 0.0 | 0.0 | 0.0 ± 0.0 |
| Example Value & Key | 0.0 | 0.0 | 0.0 | 0.0 | 0.0 | 0.0 | 0.0 ± 0.0 |
| Example Key & Key | 0.0 | 0.0 | 0.0 | 0.0 | 0.0 | 0.0 | 0.0 ± 0.0 |

## F  EXPERIMENTAL RESULTS

Table 6: Experimental results on the LLM's analogy-style completion tasks, along with our probing method for retrieving the target concept from model's internal state. The model are ordered based on precision@1 for next-token prediction. Statistical variability is reported using 95% confidence intervals, which more appropriately capture variability for metrics bounded within the [0,1] range. To control for randomness, we also introduce two control tests using Llama 3.1-8B: a comparison against a null model with randomly-permuted input embeddings ($e_s$, *permuted baseline*), and extraction of concept pairs unrelated to inputs ($y_s$, *unrelated baseline*).

| MODEL | LLM's Next Token | | DLA-based Probing | VSA-based Probing | |
|---|---|---|---|---|---|
| | Precision@1 | Precision@5 | Precision@1 | Precision@1 | Precision@5 |
| *Permuted baseline* | - | - | - | 0.080 (079-082) | 0.103 (101-104) |
| *Unrelated baseline* | - | - | - | 0.099 (097-101) | 0.105 (103-107) |
| Llama 4 Scout, 17B-16E | 0.081 (079-082) | 0.501 (498-504) | **0.773** (771-775) | 0.866 (864-868) | 0.875 (873-877) |
| GPT-2, medium | 0.267 (265-270) | 0.548 (545-551) | 0.300 (298-303) | 0.692 (689-695) | 0.702 (699-704) |
| Pythia-1.4b | 0.369 (366-372) | 0.654 (651-657) | 0.413 (410-416) | 0.778 (776-781) | 0.790 (788-792) |
| Llama 3.1-8B | 0.352 (349-354) | 0.546 (543-549) | 0.467 (464-470) | **0.891** (889-893) | **0.908** (907-910) |
| Phi 4 | 0.519 (516-522) | **0.753** (750-755) | 0.585 (582-588) | 0.887 (886-889) | 0.904 (902-905) |
| OLMo-2 | **0.529** (526-532) | 0.721 (719-724) | 0.714 (712-717) | 0.879 (877-881) | 0.892 (890-894) |
| AVERAGE | 0.352 | 0.621 | 0.542 | 0.832 | 0.845 |

### F.1  VALIDATION STRATEGY

To assess the effectiveness of our probe, we conduct two control tests (see Table 6) as proposed in *"Designing and interpreting probes with control tasks"* by Hewitt & Liang (2019):

1. **Permuted Baseline**: We compared our outputs against a null model by inputting the trained probe with randomly permuted LLM embeddings;

2. **Unrelated Baseline**: We attempt to extract concepts that are unrelated to the input using VSA-based probing.

Both tests yielded very low precision probing scores, reinforcing the effectiveness of our method. These results show that:

- Applying our VSA-based probing (see Equation 4) using concepts irrelevant to input texts results in meaningless outputs;
- Corrupted or nonsensical input embeddings also produce poor results.

That said, it is crucial to recognize a fundamental limitation of all probing approaches: by definition, the human-interpretable information encoded in LLM embeddings is not explicitly known. Consequently, no probing method can provide absolute certainty in decoding such information. To address this, we further validated our method by evaluating the trained probes on textual inputs distinct from those used during training, thereby reinforcing the reliability of our information decoding approach.

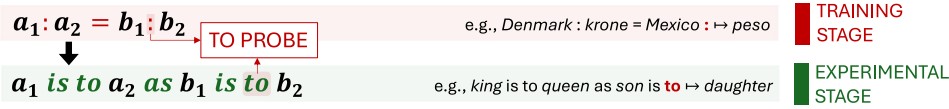

Figure 5: Experimental design for the analogy-style setup. Our experimental setup uses textual inputs with syntactic structures unseen during training.

### F.2 DISTRIBUTION OF INSTANCES WITH NO CONCEPTS EXTRACTED

We examine probe performance across different LLM input types, defining success and failure by the presence or absence of concepts extracted by VSA probing. Table 7 displays the distribution of instances with no concept extracted grouped by input categories. While we observe model-wise variability, this preliminary analysis shows a common pattern in representation blankness.

1. **Linguistic analogies** yield the lowest rate of missing concept extraction (1–1.8%), suggesting richer LLM representations, likely due to reliance on all concepts to capture *implicit syntactic patterns*.

2. **Factual knowledge** and **semantic relations** show slightly higher but still low blank rates (5.3–7%). Since these analogies rely on *key–value associations*, blanks may reflect missing associations in the model.

3. **Semantic hierarchies** (34.8%) and **mathematical analogies** (89.5%) yield the highest blank rates. Both require more *abstract reasoning*, but the large gap in mathematics likely stems from the rarity of analogical tasks with numbers, compared to equation solving or standard math problems more common in training data.

Table 7: Analogies by Area (%) for the subset of instances with no retrieved concepts for Llama 4 and OLMo2, mentioned in Section 5.2. OLMo-2 shows richer embeddings than Llama 4, with lower proportions of instances with conceptually-blank representations for most of the areas. Llama 4 slightly outperform OLMo2 in mathematical and grammatical analogies.

| Area | Llama 4 (docs, %) | OLMo2 (docs, %) | AVG | Sample | Domain |
|---|---|---|---|---|---|
| Mathematics | **87.8** | 91.1 | 89.5 | 80 is to 160 as 98 is to | math double |
| Semantic Hierarchies | 38.8 | **30.8** | 34.8 | limousine is to car as monorail is to | hyponyms |
| Semantic Relations | 10.0 | **3.9** | 7.0 | Croatia is to Croatian as Switzerland is to | nationality adjective |
| Factual Knowledge | 5.5 | **5.1** | 5.3 | euclid is to Greek as galilei is to | name nationality |
| Verbal & Grammatical Forms | **1.4** | 2.1 | 1.9 | seeing is to saw as describing is to | past tense |
| Morphological Modifiers | 1.1 | **0.8** | 1.0 | agree is to agreement as excite is to | verb+ment |

### F.3 Diagnosing erroneous answers from Llama 4

Llama 4 most frequently generated a white space token for our corpus $\bar{S}$, accounting for 76% of its outputs, considerably higher than the 8% average observed in the other models (30% for Llama 3.1). Its next most common tokens were: ? (9%), what (6%) and x (0.7%). The target token had a median rank of 5, with its SoftMax score trailing the top-1 token by a median difference of 0.85 (Appendix L.1), which starkly contrasts other models with 0.05. Thus, the model confidently predicted a space, with the target word often within its top five predictions. These insights, and the strong performance of our hyperdimensinal probe (probing@1 = 87%), suggest issues in handling the syntactical structure of our corpus rather than lack of analogical reasoning. Possibly influenced by its tokenizer (see space-token frequency in the other Llama), which emphasizes prompt engineering importance and variability caused by models' tokenizers. This may be further worsened by the model's multimodality and the complexity of its MoE architecture.

## G Experimental comparison

We compare our VSA-based results to those yeled by the Direct Logit Attribution (DLA) technique; because, unlike SAEs, it requires no extra steps such as feature-naming, making it the most direct and unambiguous comparison for our approach.

Our neural VSA encoder (Section 4.4) does qualify as a supervised probe, as it is trained to map LLM internal representations (i.e., residual stream activations) into interpretable, human-understandable features (i.e., VSA encodings). Supervised probes are typically designed for specific experimental goals or target features, ranging from syntactic structure, as in *"A Polar Coordinate System Represents Syntax in Large Language Models"* (Diego Simon et al., 2024); to real-world knowledge, as in *"Language Models Represent Space and Time"* (Gurnee & Tegmark, 2023); and to abstract semantics, as in *"The Geometry of Truth"* (Marks & Tegmark, 2023). Our probe is specifically designed around VSA principles, so direct comparisons with non-VSA probes would require fundamentally different approaches not grounded in VSAs.

While our controlled vector space (VSA encodings) parallels the SAE proxy layer, our approach uses a top-down strategy by querying it with predefined concepts (Equation 4), whereas SAEs rely on a bottom-up process that names all triggered features post hoc. This bottom-up approach reveals an unbounded set of latent features without relevance filtering, requiring exhaustive feature naming and additional filtering to isolate those aligned with our bounded input-output concept framework. In addition, while SAEs typically target a single layer, our probing approach examines nearly the entire residual stream simultaneously, complicating direct and precise comparisons. This manual intervention involved in SAE-based methods, from feature naming to filtering, prevent them from being fully automated, and directly comparable to our supervised approach. By contrast, DLA outputs a single, unique and unambiguous feature (token) constrained by the model's output vocabulary, enabling a direct comparison through a fuzzy token-to-concept matching with our concept set.

In summary, DLA is the most direct comparison, as SAE comparisons require additional steps, making them indirect and ambiguous, and supervised probes reflect only a generic mapping paradigm.

### G.1 DLA-based experimental results

To validate our results, we apply DLA to all models using $\bar{S}$, as it allows direct baseline without extra steps such as feature naming or filtering required in SAE analysis. See Appendix G for details.

We adopt simple, fuzzy token-to-concept matching approach with our concept set (e.g., pes ↦ peso), and consider projected next-token predictions (Appendix G.3) from the model's middle to last layers of the last token, as VSA probing. DLA produces no concepts in nearly 30% of analogies on average (see NONE in Table 8; +17% compared to VSA, Table 1), while yielding the target with its key in 26% of the cases (-50%). In instances without concepts from DLA, our VSA-based probe extracts, on average, the key-target pair in 57% of all analogies (Table 2), while returning none for 28%. For instance, for the analogy king is to queen as son is to ↦ daughter, using OLMo-2, our probe extracts the key-target concepts (son and daughter), while DLA produces no concepts. The model predicts the next token prediction as ? with a softmax score of 0.06, followed by father (0.05); the target word has a rank of 37. Focusing on next-token rep-

resentations, and thus capturing surface-level features, DLA exhibits inferior probing capabilities compared to ours, which compromise subsequent interpretability analyses of LLM embeddings. On the other hand, we observe substantial variance within this subset during VSA probing. Across models (Table 2), our probe fails to retrieve any concepts in 43% of cases for Llama 4, compared to only 14% for Llama 3.1. GPT-2 confirms greater representativeness for the in-context example. There is also variation across analogy categories in this subset (Table 9): for OLMo-2, linguistic analogies show the highest retrieval rates for Context | Target (7.4% and 4.4%), whereas mathematical analogies shows nearly no concept retrieval (91%), confirming common blank representations. Appendix G.2 shows that, in cases where VSA fails, also DLA frequently yields no concepts rather than other relevant concepts.

Table 8: Concepts extracted using the DLA probing technique on the full corpus $\bar{\mathcal{S}}$ with all LLMs. Likewise in our VSA-based probing, we focus on the same middle-to-bottom range of model's hidden layers of the last token. The table highlights the key common items across models, with the first six cases covering over 95% of all extracted concepts.

| Extracted Concepts (docs, %) | GPT-2 | Pythia | Llama4, Scout | OLMo-2 | Phi-4 | Llama 3.1 | AVERAGE | Δ VSA |
|---|---|---|---|---|---|---|---|---|
| NONE | 33.9 | 32.8 | 15.4 | 14.6 | 33.1 | **47.4** | 29.5 ± 11.4 | +17.3 |
| **Target** | 15.0 | 18.0 | **36.7** | 29.0 | 34.4 | 22.5 | 25.9 ± 8.1 | +25.8 |
| Key | **Target** | 12.6 | 19.3 | 38.4 | **38.5** | 22.7 | 22.1 | 25.6 ± 9.7 | - 50.4 |
| Key | 9.7 | **10.4** | 6.3 | **10.4** | 6.0 | 4.5 | 7.9 ± 2.4 | +5.0 |
| Example Value | **12.8** | 5.7 | 0.3 | 0.7 | 0.9 | 0.5 | 3.5 ± 4.6 | +3.5 |
| Example | **9.0** | 5.3 | 0.3 | 1.7 | 1.0 | 0.5 | 3.0 ± 3.2 | +0.7 |
| Example Value | **Target** | 1.0 | **2.1** | 0.7 | 0.6 | 0.3 | 0.9 | 0.9 ± 0.6 | +0.9 |
| Example Key | **3.0** | 1.5 | 0.1 | 0.2 | 0.1 | 0.1 | 0.8 ± 1.1 | +0.7 |
| Context | **Target** | 0.6 | 0.3 | 0.5 | **1.5** | 0.3 | 0.4 | 0.6 ± 0.4 | -1.5 |

Table 9: Percentages of extracted factors by analogy category considering the subset of instances when the DLA yields no concept for OLMo-2.

| Extracted concepts (docs, %) | Morphological Modifiers | Verbal & Grammatical Forms | Factual Knowledge | Semantic Relations | Mathematics | Semantic Hierarchies | AVERAGE |
|---|---|---|---|---|---|---|---|
| Key | Target | **90.3** | 83.4 | 70.1 | 79.4 | 0.0 | 41.5 | 60.8 ± 30.3 |
| NONE | 1.6 | 2.7 | 14.3 | 1.7 | **91.1** | 21.1 | 22.1 ± 31.1 |
| Example | 0.7 | 0.7 | 4.7 | 8.5 | 0.0 | **15.0** | 4.9 ± 5.1 |
| Key | 1.6 | 2.7 | 3.8 | 1.4 | 0.0 | **5.1** | 2.4 ± 1.7 |
| Key | Pair Values | 1.3 | 0.9 | 0.0 | 5.1 | 0.0 | **11.6** | 3.2 ± 4.2 |
| Context | Target | 4.4 | **7.4** | 0.8 | 1.6 | 0.0 | 0.6 | 2.5 ± 2.6 |
| Out-of-Context | 0.2 | 0.6 | 1.3 | 0.0 | **8.8** | 0.9 | 1.9 ± 2.9 |
| Context | 0.0 | **0.3** | 0.1 | 0.0 | 0.0 | **0.3** | 0.1 ± 0.1 |

### G.2 CONCEPTS EXTRACTED BY DLA WHEN VSA YIELDS NO CONCEPTS

Table 10: Concepts extracted though DLA-based probing when VSA yields no concepts. The table highlights DLA also extract no concepts in the majority of the instances ($59 \pm 15$ %), highlighting high variability among models.

| Extracted concepts (docs,%) | GPT-2 | Pythia | Llama3 | Phi-4 | OLMo-2 | Llama4 | AVERAGE |
|---|---|---|---|---|---|---|---|
| None | 40.9 | 46.5 | 83.1 | 72.7 | 55.9 | 53.9 | $58.8 \pm 14.8$ |
| **Target** | 8.1 | 9.0 | 5.3 | 15.7 | 14.3 | 23.3 | $12.6 \pm 6.4$ |
| Key | 7.5 | 11.7 | 3.4 | 4.3 | 13.2 | 9.1 | $8.2 \pm 3.7$ |
| Key \| **Target** | 6.9 | 8.7 | 5.7 | 3.6 | 11.8 | 11.9 | $8.1 \pm 3.3$ |
| Example Value | 17.9 | 11.1 | 1.3 | 1.0 | 1.3 | 0.6 | $5.5 \pm 6.7$ |
| Example | 12.1 | 6.6 | 0.4 | 2.2 | 1.7 | 0.1 | $3.9 \pm 4.3$ |
| Example Key | 3.3 | 2.7 | 0.1 | 0.1 | 0.2 | 0.2 | $1.1 \pm 1.2$ |
| Example Value \| **Target** | 1.0 | 1.7 | 0.1 | 0.0 | 0.2 | 0.2 | $0.5 \pm 0.7$ |
| Example Value \| Key | 1.0 | 0.6 | 0.1 | 0.0 | 0.4 | 0.3 | $0.4 \pm 0.4$ |
| Example Key \| Key | 0.3 | 0.4 | 0.1 | 0.0 | 0.1 | 0.1 | $0.2 \pm 0.2$ |
| Example Value \| Key \| **Target** | 0.3 | 0.3 | 0.0 | 0.0 | 0.3 | 0.2 | $0.2 \pm 0.1$ |
| Context \| Target | 0.3 | 0.2 | 0.3 | 0.2 | 0.4 | 0.1 | $0.3 \pm 0.1$ |
| Example Key \| Target | 0.2 | 0.1 | 0.0 | 0.0 | 0.0 | 0.0 | $0.1 \pm 0.1$ |
| Target \| Example | 0.1 | 0.1 | 0.0 | 0.0 | 0.1 | 0.0 | $0.1 \pm 0.1$ |

### G.3 RAW RESULTS OBTAINED THOUGH THE DLA PROBING TECHNIQUE

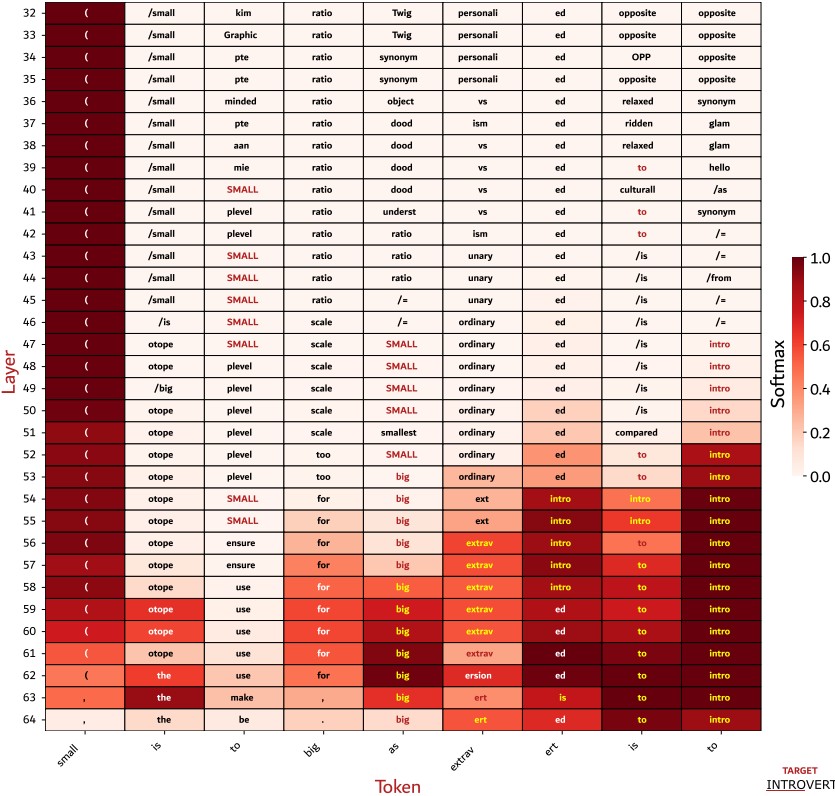

Figure 6: Comprehensive raw outputs obtained though DLA on OLMo-2 for a sampled analogy.

# H APPLICABILITY TO OTHER DOMAINS

## H.1 GENERALIZATION OF INPUT REPRESENTATION

VSA representations are automatically generated from input features, with their construction guided by the probing objective and the target latent features. While our work focuses on textual inputs with well-defined semantics, allowing straightforward extraction of input features (i.e., words), the underlying principle is flexible and generalizable. Equation 2 illustrates the creation of input representations via binding and bundling operations for our specific input template and downstream task. The hyperdimensional algebra underlying VSA allows this approach to generalize to other textual formats, NLP tasks, and even multi-modal data (see appendix P).

Scalability challenges depend largely on the nature of the input features. For tasks such as toxicity detection, expert-labeled data or specialized feature extraction pipelines may be required. For example, mapping the phrase *"You are a pathetic excuse for a human just like the rest of your kind"* to a conceptual form such as $(\phi_{\text{attack}} \odot \phi_{\text{insult}}) + (\phi_{\text{attack}} \odot \phi_{\text{identity}})$ requires human expertise. Once features are extracted, however, constructing VSA encodings is automatic, efficient, and scalable. VSA probing can then uncover encoded concepts in the LLM vector space, for instance:

$$y_s \oslash \phi_{\text{attack}} = \phi_{\text{identity}} + \text{noise}$$

In contrast, tasks based on syntactic structures offer more scalable input extraction. For example, the sentence *"The city of Turin is in Italy"* can be processed with conventional techniques such as POS tagging and Semantic Role Labeling (SRL). A VSA encoding can then be automatically created:

$$(\phi_{\text{NOUN}} \odot \phi_{\text{city}}) + (\phi_{\text{PROPN}} \odot \phi_{\text{Turin}}) + (\phi_{\text{VERB}} \odot \phi_{\text{be}}) + (\phi_{\text{PROPN}} \odot \phi_{\text{Italy}})$$

## H.2 APPLICABILITY TO OTHER DOWNSTREAM TASKS

Although we demonstrate VSA-based probing using analogy-competition tasks, the methodology is generalizable to other experimental settings. The analogy-based dataset was chosen to:

- provide a simple, controlled, and interpretable evaluation environment;

- elicit LLMs to focus on concepts and their inherent relationships;

- probe the LLM vector space with inputs spanning a spectrum of reasoning tasks.

Thanks to the flexibility of VSAs and hypervector algebra, VSA-based probing can be applied to a wide variety of experimental settings with different:

1. **Downstream tasks**. Our decoding paradigm can be used for linguistic feature extraction, toxicity detection, or bias classification;

2. **Textual templates**. For example, in question-answering setting, an input text in such as *"Who wrote the play Romeo and Juliet?"* can be encoded as

   $$(\phi_{task} \odot \phi_{question}) + (\phi_{relation} \odot \phi_{writtenBy}) + (\phi_{play} \odot \phi_{Romeo\&Juliet})$$

   allowing the VSA to query LLM representations and reveal which concepts are strongly represented or linked to the predicted answer;

3. **Modalities**. As discussed in Appendix P, inputs combining text with other modalities could also be encoded and probed via VSAs.

VSA-based probing thus provides a unified, flexible framework for examining how LLMs encode and relate abstract input features, from syntactic structures to high-level concepts such as gender bias or toxic language.

## I QUESTION-ANSWERING SETTING FROM SECTION 6

We generate 693,886 training examples $\mathcal{Q}$ from the SQuAD dataset using an augmenting strategy by incrementally considering textual questions with their corresponding features:

$(A_1)$ "What was the `name`" $\mapsto \phi_{\text{name}}$

$(A_2)$ "What was the `name` of the `ship`" $\mapsto \phi_{\text{name}} + \phi_{\text{ship}}$

$(A_3)$ "…" $\mapsto \ldots$

$(A_{n-1})$ "What was the `name` of the `ship` that `Napoleon` sent to `the Black Sea`?" $\mapsto \phi_{\text{name}}$

$\qquad \mapsto \phi_{\text{name}} + \phi_{\text{ship}} + \phi_{\text{napoleon}} + \phi_{\text{send}} + \phi_{\text{theBlackSea}}$

$(A_n)$ "What was the `name` of the `ship` that `Napoleon` sent to `the Black Sea`?"

`Charlemagne`" $\mapsto (\phi_{\text{name}} + \phi_{\text{ship}} + \phi_{\text{napoleon}} + \phi_{\text{send}} + \phi_{\text{theBlackSea}}) + \phi_{\text{charlemagne}}$

For our experiments, we generate another corpus $\bar{\mathcal{Q}}$ including also the contextual text (Wikipedia article) provided for each SQuAD's items:

> "Napoleon III responded with a show of force … by the Greek Orthodox Church.
>
> Q: What was the name of the ship that Napoleon sent to the Black Sea? $\qquad$ (6)
>
> A ($\leq$ 3 words):"

Lastly, we apply our entire pipeline by probing the final state of a language model at the last token (colon) and extracting concepts through comparison with the codebook $\Phi$. We analyze the model's internal state across the text generation process, considering the residual stream at initialization ($\mathbf{H}[\text{seq}_0]$) and after the autoregressive generation of $t$ tokens ($\mathbf{H}[\text{seq}_t]$).

## J COSINE SIMILARITIES AMONG THE ITEMS OF THE VSA CODEBOOK

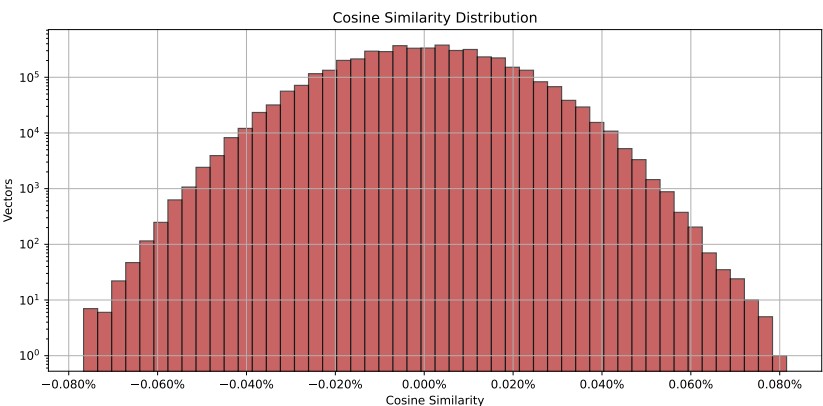

Figure 7: Distribution of pair-wise cosine similarities among the items of the codebook.

## K  SPEARMAN CORRELATION FOR THE QA-RELATED EXPERIMENTS

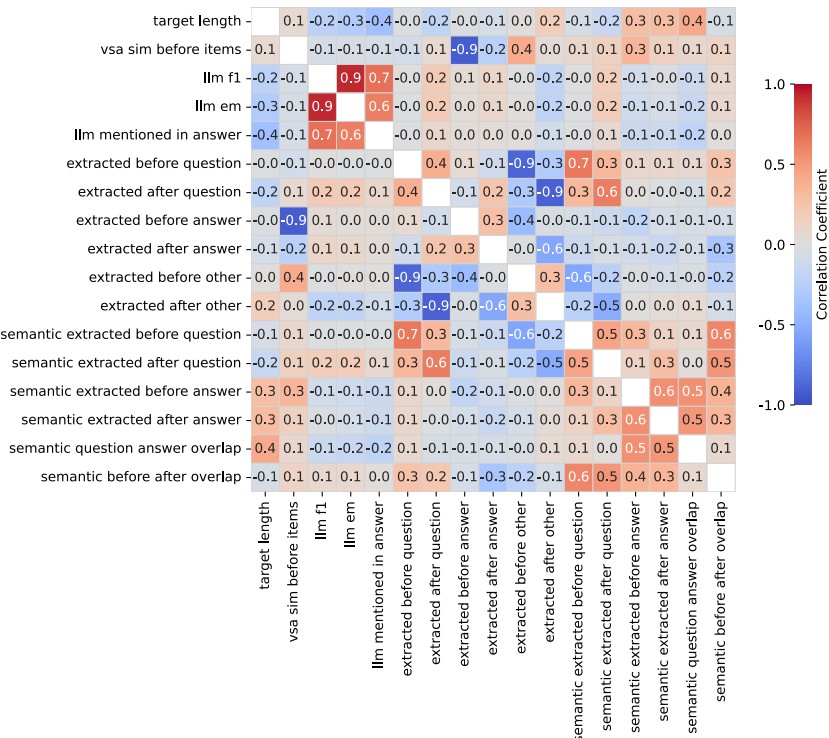

Figure 8: Spearman correlation coefficients computed on $\bar{\mathcal{Q}}$.

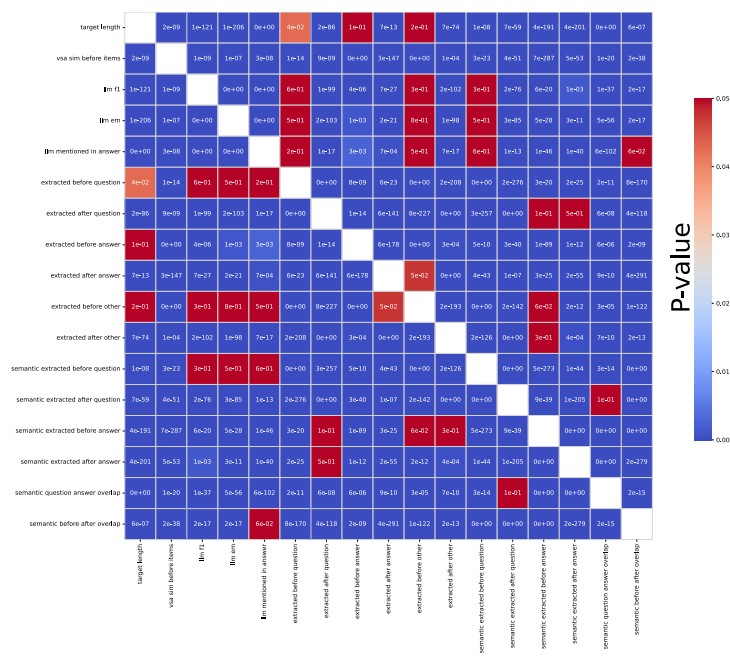

Figure 9: P-values of the Spearman correlation coefficients.

# L  OVERVIEW OF THE EXPERIMENTAL METRICS

## L.1  LLAMA 4, SCOUT

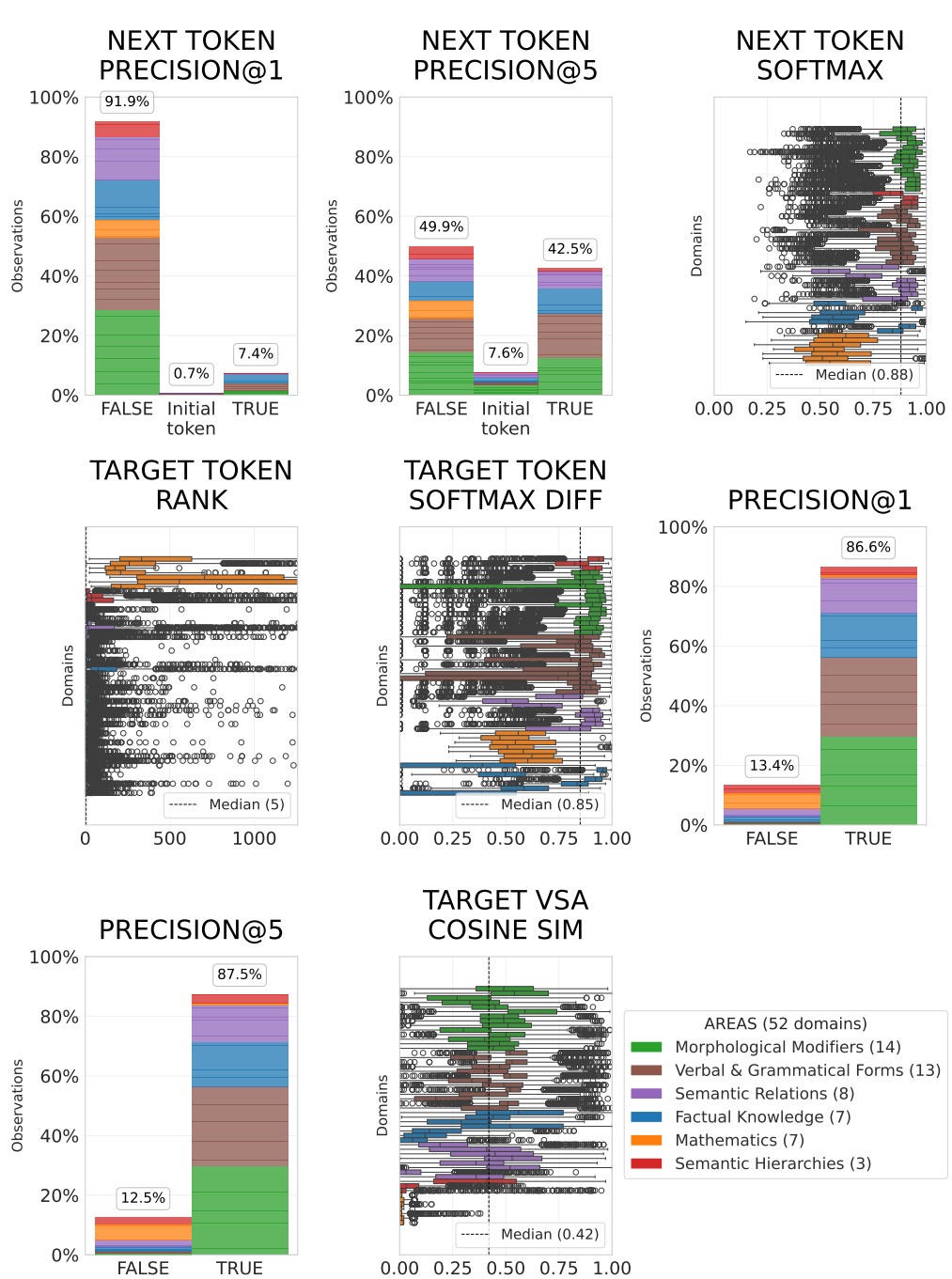

Figure 10: Experimental metrics of the LLM's next-token prediction task and probing performance for Llama 4. *Precision@k* is displayed as a categorical variable, with its binary values portrayed as boolean. The category *initial token* is associated to the special case (0.5) introduced in Section 5.1. We measure VSA noise by computing the cosine similarity between the retrieved target concept and its codebook version Φ.

## L.2   OLMO-2

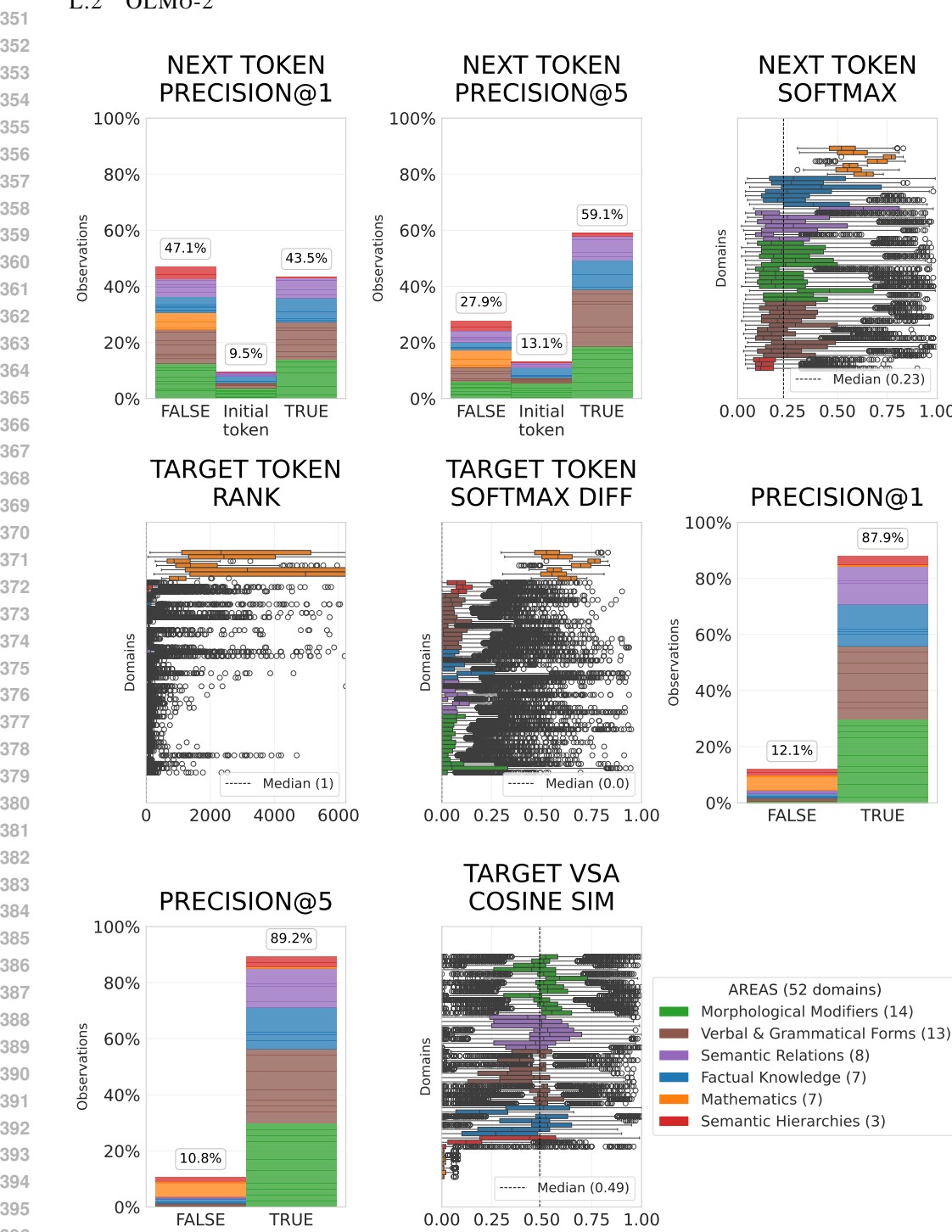

Figure 11: Experimental metrics of the LLM's next-token prediction task and probing performance for OLMo-2. *Precision@k* is displayed as a categorical variable, with its binary values portrayed as boolean. The category *initial token* is associated to the special case (0.5) introduced in Section 5.1.

# M  SYNTHETIC CORPUS

Table 11: Knowledge bases for our synthetic corpus $\mathcal{S}$.

| Dataset | | Domains | | Sample example |
|---|---|---|---|---|
| Google Analogy Test Set | 12 | capital world, currency, plural, . . . | 33,812 | `Denmark : krone = Mexico : peso` |
| Bigger Analogy Test Set | 33 | verb+ment, occupation, gender, . . . | 73,471 | `queen : king = mother : father` |
| Mathematics | 7 | double, square, division2, . . . | 6,816 | `4 : 16 = 5 : 25` |
| | 52 | | 114,099 | |

Table 12: Overview of our experimental set, grouped by tasking an LLM to cluster the domains.

| Category | | Domains | Docs |
|---|---|---|---|
| Morphological Modifiers | 14 | noun+less, adj+ness, . . . | 34,308 (30%) |
| Verbal & Grammatical Forms | 13 | past tense, plural, . . . | 31,219 (27%) |
| Factual Knowledge | 7 | country capital, occupation, . . . | 18,800 (17%) |
| Semantic Relations | 8 | family, genders, . . . | 16,831 (15%) |
| Mathematics | 7 | math double, math division5, . . . | 6,816 (6%) |
| Semantic Hierarchies | 3 | hypernyms, hyponyms, . . . | 6,125 (5%) |
| | 52 | | 114,099 (100%) |

Table 13: All domains, and their corresponding cardinality after data augmentation for training.

| Domain | Examples | Domain | Examples | Domain | Examples |
|---|---|---|---|---|---|
| country_capital | 21801 | capital_world | 18561 | country_language | 12299 |
| antonyms_gradable | 11268 | adj_superlative | 10942 | un+adj_reg | 10614 |
| adj+ly_reg | 10576 | adj_comparative | 10519 | male_female | 10236 |
| noun_plural_reg | 10216 | noun_plural_irreg | 10206 | verb_Ving_Ved | 10164 |
| verb_inf_3pSg | 10112 | animal_sound | 10083 | verb_inf_Ving | 10008 |
| name_nationality | 9998 | verb+er_irreg | 9865 | verb_Ving_3pSg | 9861 |
| verb+able_reg | 9849 | adj+ness_reg | 9849 | animal_shelter | 9833 |
| hypernyms_animals | 9831 | over+adj_reg | 9828 | re+verb_reg | 9821 |
| verb+ment_irreg | 9807 | verb_inf_Ved | 9805 | UK_city_county | 9805 |
| name_occupation | 9801 | noun+less_reg | 9801 | verb_3pSg_Ved | 9801 |
| verb+tion_irreg | 9801 | hypernyms_misc | 9719 | antonyms_binary | 9603 |
| past_tense | 6313 | plural | 4129 | comparative | 3765 |
| present_participle | 3401 | plural_verbs | 3055 | currency | 2983 |
| adjective_to_adverb | 2977 | math_double | 2918 | nationality_adjective | 2818 |
| superlative | 2545 | math_division2 | 2498 | opposite | 2221 |
| math_division5 | 641 | family | 529 | math_squares | 402 |
| math_division10 | 258 | hyponyms_misc | 102 | math_root | 77 |
| math_cubes | 29 | | | | |
| DOMAINS: 52 | | | | TEXTUAL EXAMPLES: 395,944 | |

# N  DECLARATION OF LLM USAGE

The paper presents a pipeline that treats LLMs as subjects of study, not tools. To enhance interpretability, we adopted an LLM (GPT-4o) to categorize the 52 distinct analogy domains into semantically coherent macro categories (Table 12 in Appendix M).

## O    DIMENSIONALITY REDUCTION

### O.1    AVERAGE CORRELATIONS AMONG MODEL'S HIDDEN LAYER

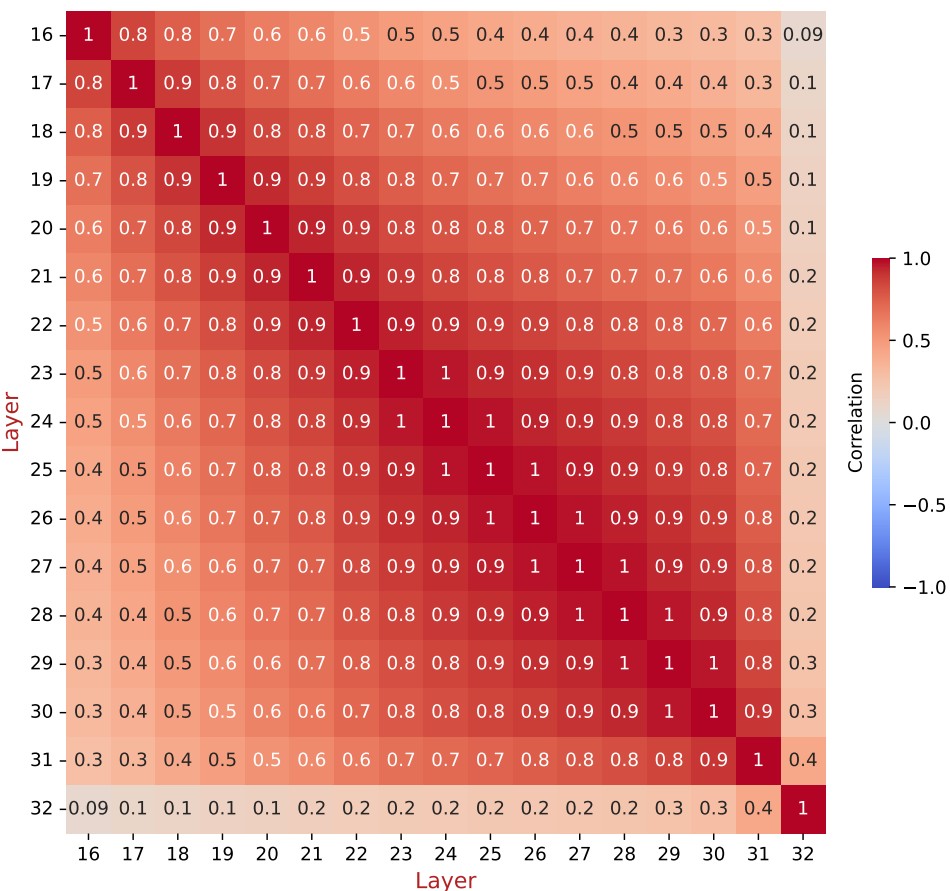

Figure 12: Average Person correlations among the second half of model's hidden layers for Llama3.1

### O.2    ANALYSIS OF REPRESENTATION REDUNDANCY

In Section 4.3, we hypothesize that highly correlated rows (model's adjacent layers) could cause redundant representations, since they likely encode similar numerical patterns, and thus information.

Here, we present an analysis of representation redundancy, defined as approximate linear dependence among LLM hidden layer embeddings. We computed the Gram matrix $G = HH^T$, where $H$ is the model's residual stream, and analyzed its eigenvalues. Table 14 shows results for the OLMo-2 model (considering the 32nd-to-64th range of hidden layers; Appendix D), averaged on a 100K training input sample. The spectrum reveals a few dominant eigenvalues (around 3-4 modes) followed by many smaller ones, indicating that the embedding space is approximately low-rank. This suggests that, when considering the full matrix ($\mathbb{R}^{33 \times 5120}$ for OLMo-2), most hidden layer representations (rows) are redundant, since only a few rows (or their combinations) contribute meaningful structure. The first mode is by far the most dominant, with a normalized eigenvalue of 0.65, compared to 0.17 for the second. We hypothesize that this leading component might correspond to next-token prediction representations, while the remaining modes capture secondary structures or auxiliary information. Our hyperdimensional probe aims to capture also these auxiliary latent structures, rather than limiting solely on the single predominant component.

Table 14: Eigenvalues (EV) of the Gram matrix from OLMo-2's residual stream.

| Comp. | EV (mean ± std) | Norm. EV |
|---|---|---|
| 0 | 58084 ± 5293 | 0.650 |
| 1 | 15450 ± 2056 | 0.170 |
| 2 | 5972 ± 608 | 0.070 |
| 3 | 2539 ± 330 | 0.030 |
| 4 | 2057 ± 220 | 0.020 |
| 5 | 1187 ± 166 | 0.010 |
| 6 | 727 ± 119 | 0.010 |
| 7 | 505 ± 83 | 0.010 |
| 8 | 363 ± 59 | 0.000 |
| 9 | 282 ± 48 | 0.000 |
| 10 | 230 ± 37 | 0.000 |
| . . . | . . . | . . . |
| 30 | 30 ± 6 | 0.000 |
| 31 | 27 ± 6 | 0.000 |
| 32 | 22 ± 6 | 0.000 |

## O.3 SILHOUETTE ANALYSIS FOR DETERMINING OPTIMAL RANGE OF CLUSTERS

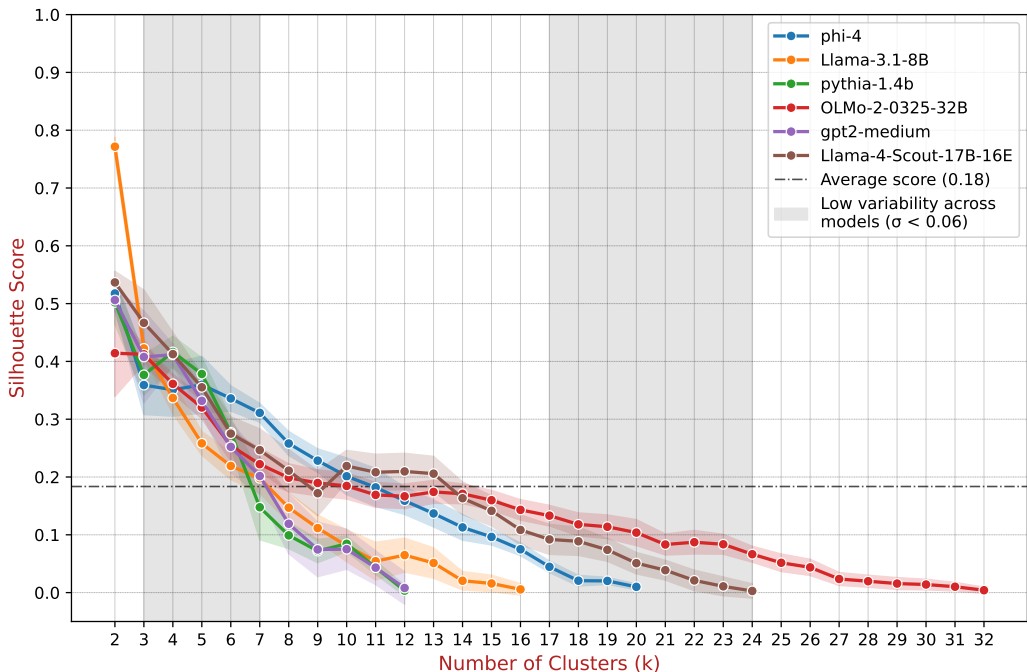

Figure 13: Silhouette scores for varying numbers of clusters, computed using a random sample of 10,000 textual inputs from $\mathcal{S}$. The six language models have varied layer counts (see Table 4), which results in different maximum possible cluster numbers.

## O.4 Distribution of cluster assignments for grouping model's hidden layers

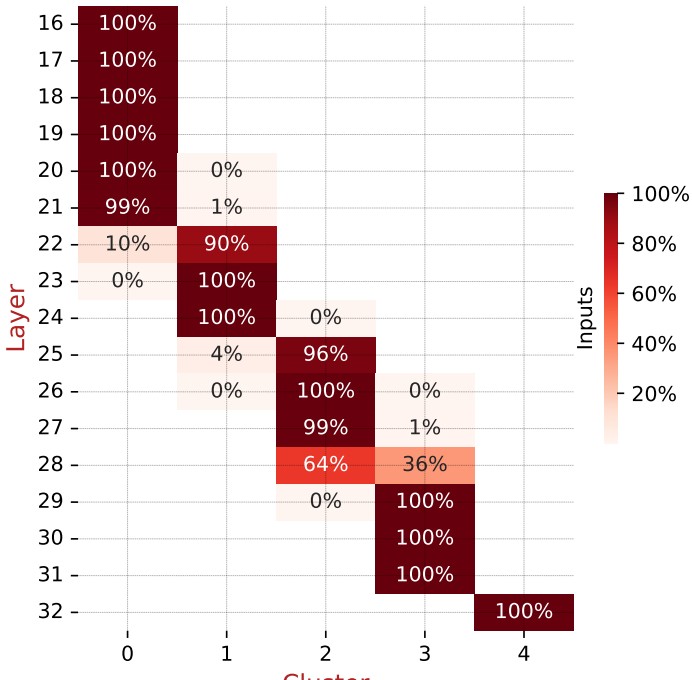

Figure 14: Distribution of model's hidden layers grouped by k-means clustering within the ingestion algorithm $F$ for Llama3.1-8B. It portrays the percentages of cluster assignments across all instances.

## O.5 Ablation study on the dimensionality-reduction steps

This section presents an analysis of skipping the dimensionality reduction steps introduced in Section 4.3. While our VSA-based methodology would work without these compression steps, the overall computational cost of probing would dramatically increase. For example, our ingestion procedure (Appendix B; Section 4.3) reduces the probed OLMo-2's embeddings from $\mathbb{R}^{33 \times 5120}$ to $\mathbb{R}^{5120}$. This allows our neural VSA encoder to have an input dimension $d = 5012$ with only 71M trainable parameters (see Appendix C).

If the two steps are eliminated, and thus the entire residual stream of the model $\mathbb{R}^{33 \times 5120}$ is considered, the encoder receives a flat input vector, creating an input dimension $d = 168960 \in \mathbb{R}^{168960}$. Although the encoder would internally handle feature extraction, since the flattened input holds all the information encoded in the LLM embeddings, this approach would increase the number of trainable parameters to 742 million, representing a tenfold increase. Additionally, adopting a lazy feature extraction stage in an input vector space of size $\approx 10^5$, which is approximately low-rank (see Appendix O.5), would result in a computationally inefficient approach.

Removing one of the two steps, such as sum pooling, should lead to just an increase of the overall computational cost for the encoder ($\mathbb{R}^{5 \times 5120} \mapsto \mathbb{R}^{25600}$; $d = 25600$; 155M trainable parameters; x2), rather than affecting probe's outputs. Further, since our neural VSA encoder is found effective to extract latent features even from our heavily-compressed input representation (Section 5), other dimensionality reduction approaches could also be as effective as ours (Appendix B).

In summary, skipping the compressing steps is possible and the only drawbacks should be the increase of footprint of both the training and inference stages of the VSA-based probing (see also Appendix Q).

## P   PROOF OF CONCEPT FOR HYPERDIMENSIONAL PROBE IN MULTIMODAL SETTINGS

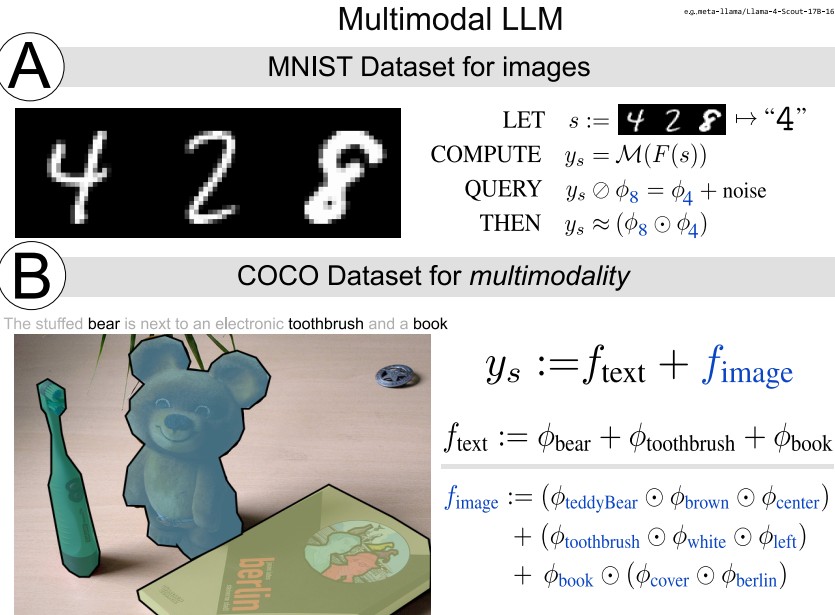

Figure 15: Proof of concept for using hyperdimensional probe in multimodal settings. Figure A shows a complete probing procedure for a MNIST-based mathematical analogy. Figure B exhibits a VSA encodings describing a multimodal input using textual and image features.

## Q   COMPUTATIONAL WORKLOAD

The computational workload of this work is split into two parts: LLM inference (exogenous, Section 4.3) and the training and probing stages of our method (endogenous, Section 4.4 and 4.5).

The exogenous factor, running the Large Language Models, was the most computationally demanding task. For our experiments, we tested six different Large Language Models in inference mode, caching their embeddings for our training phase and probing them dynamically during the inference phase of our work (Figure 1). We worked with LLMs ranging from 355M parameters (GPT-2) to 109B parameters (Llama 4, Scout), using between one and three NVIDIA® A100-80GB GPUs, depending on the model size. Quantization is not employed.

In contrast, the computational demands of our VSA-based methodology is relatively low. The most resource-intensive stage was training our neural VSA encoder, but due to its modest size (ranging from $55M$ to $71M$ parameters, see Appendix C), this process remained lightweight. We performed this training on a single GPU, though it could easily be handled with much less powerful and lower-memory GPUs. Regarding GPU usage, we trained our encoder for approximately 8 hours on each LLM's embeddings, though the process could have been shortened with a less conservative early stopping criterion or by reducing the amount of training data.

The probing stage is then composed of simple vector multiplications (unbinding, Section 3), after loading the heavy LLM and our lightweight trained neural VSA encoder into memory (from 800 MB of the 55M version to 1 GB of the biggest one). Future research could explore even further reducing the latent dimension of our neural VSA encoder (Appendix C) or adopt VSA encodings with lower dimensionality (e.g. $D = 512$, leading to a more lightweight encoder. The time cost of probing is thus comparable to simple LLM inference with a slight increase due to feedforward our lightweight trained model, with the time demand for vector multiplications being negligible. Accordingly, the GPU hours for probing depends mainly on the amount of test data and model size; for example, we used around 95 hours of GPU computation for probing embeddings of Llama 3.1-8B on $\bar{S} \approx 10^5$, processing each instance in around 3 seconds.

