# OpenReview forum: "Hyperdimensional Probe: Decoding LLM Representations via Vector Symbolic Architectures"
_ICLR.cc/2026/Conference — Submitted to ICLR 2026_

### Official Review · Reviewer_dtKf · 2025-10-16

**Soundness:** 3
**Presentation:** 2
**Contribution:** 1
**Rating:** 0
**Confidence:** 4

**Summary:**

The work proposes a novel probing procedure, in which a new tokenization is first constructed in order to orthogonally represent the concepts in the training corpus. Empirical results are shown on a synthetic dataset and a real one to show that this method can successfully extract interpretable concepts from the LLM's hidden space. A novel method for reducing the dimensionality of the full LLM hidden space is also proposed as part of the procedure.

**Strengths:**

The paper is laid out in a logical way, making it easy to follow. The method is novel, and some interesting decisions are made as to its implementation.

**Weaknesses:**

1.Experimental design is highly questionable

1(a). The method is only compared against naively prompted LLMs, despite the description of a full body of related work above. Also, since your method is trained, fine-tuning the LLM on your dataset should be considered as a baseline. The free-form setting which was chosen both conveniently eliminates most probes, and obscures LLM evaluation.

1(b) In 5.1 (l. 345), the LLM is considered half-correct by a 0.5 accuracy value when its next-token prediction is correct but the full answer to the question is multi-token. This seems extremely questionable.

1(c) LLM comparison throughout the results section seems to be conducted with naive-prompting. Few-shot examples are not provided. Certainly, the very low performance on what should be a very simple task (simple analogical reasoning) reported for the LLMs is indicative of problematic treatment. Without considerate prompt design, or few-shot examples demonstrating output format, LLMs are liable to provide answers in a variety of forms, such as "the answer is [answer]", rather than "[answer]", making next-token measurement unsuitable. Your later evaluation on the real dataset, based on inclusion of the answer in the text, shows much more reasonable performance (supporting this suspicion).

1(d) l. 443: the reported F1 and EM statistics for the real-world dataset show exceptionally large variances ( 0.69 ± 0.38, 0.52 ± 0.5 respectively). How were these variances calculated? Statistical significance of your empirically supported claims should be provided.

1(e) a single non-synthetic dataset is tested on. The dataset used does not capture notions of reasoning, despite the synthetic data being reasoning-focused.

2. Impact: the impact of this paper seems weak. Correctly, no claims of performance gains are made. The main claim to impact in this paper is the demonstration that VSAs can work, and some promisory demonstrations of potential downstream use cases.

**Questions:**

1. In practice, this method effectively mirrors the tuned lens framework in which a mapping is learned from the hidden state through to the decoding matrix, in order to address layer-decoder un-alignment that may be present due to the intermediate layers' lack of explicit training for this application. The main distinction is that in this approach, a new decoder is designed before training. Your claim is that the nature of this new vectorization addresses some of the weaknesses of supervised probes. Can you demonstrate this empirically in any way?
2. Presumably, if a latent representation learned for interpretability contains more information on the important concepts pertaining to a problem than a different representation, then it has a larger potential performance as a model-input. How do you support your claims that VSA leads to a richer and more conceptually faithful space relative to the various related work discussed, when performance compared to even vanilla LLM prompting is similar.

---

> ### Author Response · Authors · 2025-11-20
> **Rebuttal by Authors (part 1)**
>
> Thanks for highlighting that our work:
> 1. proposes a *novel method*;
> 2. adopts *interesting decisions* in its implementations;
> 3. is *structured in logical manner*.
>
> However, there appears to be **a misunderstanding of the scope of our work**, which might have resulted to the notably low rating of zero (outside the range of 4–6 given by other reviewers).
>
> In the revised version, we plan to use the additional page to *clarify our objective* and *its corresponding experimental setup*.
>
> # (1) Experimental design and the scope of our work
> Our work **does not propose** a *new tokenization method*, *vectorization scheme*, *LLM architectural component*, or *tuned-lens–style framework*, *nor does it address layer–decoder misalignment* or *depend on any decoding matrix*. Instead, **we present a new probing paradigm for information extraction from a LLM embeddings** by combining ideas from neural probing and symbolic representations (VSAs; L47).
>
> Our approach focuses on decoding information encoded in LLM embeddings **without any intervention during LLM inference**, treating **LLMs as subject of study** rather than competitors. Our probing approach inspects an LLM's internal state (L222) prior to its next-token prediction (Section 5.2), and during text generation (Section 5.3). It does not attempt to address layer–decoder misalignment or compensate for the absence of explicit, application-specific training in intermediate layers. **Our work frames within the domain of *interpretability*, where competitors are other representation decoding paradigms such as SAEs and DLA-based techniques (Section 2), *rather than fine-tuned LLMs* or *prompt techniques* such as few-shot learning.**
>
> *Our experimental setup reflects this aim*: it is intended to **demonstrate effective information decoding across different LLMs, rather than to claim superior performance on vanilla LLMs**. Our comparison with LLM performance aimed to highlight the debugging potential of our VSA-based probing (L76): the target concept was decoded from the LLM's internal state, but the model fails to output the correct word; maybe due to exogenous factors (L357; e.g., prompt design), or endogenous ones (e.g., tokenization).
>
> To *validate effective information decoding* of our probe, we design an experimental setup with:
> 1. six diverse LLMs (from OpenAI's GPT-2 to MetaAI's Llama 4);
> 2. varied embedding dimensions (1024-5120), and hidden layer counts (13-32);
> 3. testing inputs different from those seen during probe training.
> To assess *generalizability*, we evaluate our proposed methodology in two experimental settings: one analyzing the model’s internal state during next-token prediction in input-completion tasks (Section 5.2), and the other examining the model’s autoregressive text generation in question-answering tasks (Section 5.3).
>
> While LLM performance on downstream tasks can certainly be improved via fine-tuning or prompt-based techniques (e.g., few-shot examples), this was not the objective of our work. **Our experimental design and its corresponding results serve to validate two novel paradigms in LLM interpretability**:
> 1. the hypothesis that latent features encoded in LLM vector space can be represented with VSA-based representations;
> 2. the effectiveness of VSA-based information decoding on six different LLM vector spaces.
>
> # (1.a) Experimental comparison
> **LLM fine-tuning is not a relevant comparison for our probing approach**, as our work treats *LLMs as a subject of study* (information decoding) rather than competitors.
>
> We apologize; **we did contrast our findings against a relevant probing approach**: Direct Logit Attribution (DLA). *While key insights from the DLA comparison are mentioned in the main text (Section 5; L328), a comprehensive analysis and our comparison choice are detailed in Appendix G.*
>
> To adhere to space limitations, we prioritized demonstrating the experimental generalizability of our proposed approach in the main text. Specifically, Section 5.3 demonstrates the applicability of our VSA-based probing method to a separate use case: it shifts the application from probing a model's internal state prior to its next-token prediction (in input completion tasks) to using question-answering to probe the model's text generation process. Prompted by your comment, we will use the additional page in *the revised version to move the comparison from Appendix G into the main text as a new section*, numbered 5.4.
>
> ## DLA-based comparison (from Appendix G.1)
> To validate our results, we apply DLA to all models using $\bar{S}$, as it allows direct baseline without extra steps such as feature naming or filtering required in SAE analysis.
>
> We adopt simple, fuzzy token-to-concept matching approach with our concept set (e.g., pes $\mapsto$ peso), and consider projected next-token predictions (Appendix G.3) from the model's middle to last layers of the last token, as VSA probing.
>
> See Appendix G.1 for further details.

---

> ### Author Response · Authors · 2025-11-20
> **Rebuttal by Authors (part 2)**
>
> DLA fails to produce concepts in nearly 30% of analogies (see NONE in Table 7; +17% vs. VSA in Table 1) and yields the key-target pair in only 26% of cases (−50%). In these DLA “misses,” our VSA-based probe extracts the key-target pair in 57% of analogies, returning none in 28% (Table 8).
>
> For example, in "king is to queen as son is to" $\mapsto$ "daughter using OLMo-2, our probe identifies son and daughter, while DLA produces no concepts; the model predicts "?" (0.06) and "father" (0.05), with daughter ranked 37th. *By focusing only on next-token representations, DLA captures surface-level features, yielding weaker probing results and limiting interpretability of LLM embeddings compared to VSA-based probing.*
>
> We observe substantial variance in VSA probing within this subset. Across models (Table 8), our probe retrieves no concepts in 43% of cases for Llama 4 versus 14% for Llama 3.1, while GPT-2 shows stronger in-context example representation. Variation also occurs across analogy types (Table 9): for OLMo-2, linguistic analogies have the highest "Context|Target" retrieval rates (7.4% and 4.4%), whereas mathematical analogies show almost no retrieval (91%), reflecting common blank representations. Appendix G.2 shows that when VSA fails, DLA also frequently produces no concepts rather than alternative relevant ones.
>
> On the other hand, **the free-form setting is itself an advantage over traditional token-based interpretability paradigms such as DLA-based probes (L360). **Because these methods depend on the model’s output vocabulary, they are limited to surface-level features (L329, e.g., next-token representations) and* become sensitive to both exogenous factors* (L358, such as prompt design) and *endogenous ones* (e.g., tokenization), resulting in findings that likely vary across settings and models.
>
> In contrast, **our VSA-based insights are largely unaffected by these factors**, as is not restricted to token–level surface-level representations.
>
> # (1.b) Metrics for LLM performance
> Due to space constraints, we briefly explain LLM-related metrics, and the motivation of introducing a partial-correct score when evaluating LLM’s precision in input-competition tasks (Section 5.2). While we already include some remarks in the extended limitation section (Appendix A), we will use the additional page in the revised version to clarify this aspect directly in the main text.
>
> We introduce a score of 0.5 in precision@k for partially matching tokens to **address cases where tokenization and morphological similarity introduce ambiguity in next-token evaluation**. In our dataset, several concepts, particularly in linguistic analogy pairs, have *closely related surface forms* (e.g., accomplish vs. accomplishment). When an LLM predicts a token that represents the shared prefix of multiple valid concepts, we cannot reliably infer which concept the model intended without implementing autoregressive generation. Treating such predictions as:
> - fully correct (1) would overestimate model performance in completing analogies,
> - while treating them as fully incorrect (0) would underestimate it.
> - The intermediate score therefore reflects genuine uncertainty in model intent rather than model error.
>
> Empirically, *ambiguous cases are relatively rare* (Appendix A), owing to the extensive vocabularies of modern LLMs: on average they occur in only 8.6\% of next-token precision@1 instances across all the six models (Llama 4: 0.7\%, Pythia: 16.2\%). Thus, while this choice slightly lowers precision@k for LLM performance, the **0.5 weighting provides a conservative but fair way to account for token-level ambiguity inherent in our data** and in current LLM tokenization schemes.
>
> In response to your comment, we recalculated the LLM’s performance by converting all 0.5 scores to 0 or 1. **This adjustment resulted in only minor changes, with average changes in the 4–7\% range, leaving our overall findings unaffected.** The following table reports the metrics under the original metrics with partial-matching scheme and the two converting strategies:
> | MODEL                 | P@1 with 0.5 | P@5 with 0.5 | P@1 (0.5→0) | P@5 (0.5→0) | P@1 (0.5→1) | P@5 (0.5→1) |
> |----------------------|----------------|----------------|------------------|------------------|------------------|------------------|
> | Llama 4 Scout, 17B-16E | 0.077 | 0.463 | 0.074 | 0.425 | 0.081 | 0.501 |
> | GPT-2, medium         | 0.227 | 0.471 | 0.187 | 0.394 | 0.267 | 0.548 |
> | Pythia-1.4b           | 0.288 | 0.541 | 0.207 | 0.429 | 0.369 | 0.654 |
> | Llama 3.1-8B          | 0.309 | 0.490 | 0.266 | 0.435 | 0.352 | 0.546 |
> | Phi 4                  | 0.478 | 0.683 | 0.436 | 0.614 | 0.519 | 0.753 |
> | OLMo-2                | 0.482 | 0.656 | 0.435 | 0.591 | 0.529 | 0.721 |
> | **AVERAGE**           | 0.310 | 0.551 | 0.268 | 0.481 | 0.352 | 0.621 |
> | Delta (%)             |        |        | -4.2% | -7% | +4.2% | +7% |

---

> > ### Author Response · Authors · 2025-11-20
> > **Rebuttal by Authors (part 3)**
> >
> > On the other hand, the question-answering scenario (Section 5.3) requires adopting different LLM-related metrics to evaluate the LLM performance, as the focus is on model's text generation, yielding multi-token textual answers rather than single next-token predictions as input-completion tasks (Section 5.2).
> >
> > For the LLM’s question-answering outputs, we report performance (L443) using two other standard evaluation metrics: F1 score and Exact Match (EM). We follow the same metrics used by the creators of the SQuAD dataset (The Stanford Question Answering Dataset) and rely on the [official evaluation script]({https://rajpurkar.github.io/SQuAD-explorer/) for implementation. The F1 score measures token-level overlap between the model’s output and the ground-truth answer, while EM indicates whether the generated text matches the ground truth exactly.
> >
> > We acknowledge that the lack of an explicit clarification on LLM-related metrics, due to space constraints, can generate confusion in the readers. We will clarify the metrics-related description in the 10-page revised version.
> >
> > # (1.c) Prompting and prompt design
> >  **The objective of our work was not to boost LLM performance on downstream tasks via prompt engineering such as few-shot learning.** Our work treats *LLM failures as opportunities* to showcase the debugging capabilities of our probing framework (L76), rather than as errors to be corrected.
> >
> > For example, Section 5.2 unveils that llama-family models achieves strong probing performance while struggling to accomplish generate the correct next token (L356). This indicates that their LLM embeddings encoded the target concept, but the model often fails to output it, likely due to exogenous factors such as prompt design. Conversely, this demonstrate VSA-based probing is unaffected by prompt design, overcoming the impact of exogenous and endogenous factors of token-based probing methods such as DLA (L361).
> >
> > Furthermore, we employ next-token metrics only for input-competition tasks in Section 5.2, given the single-word outputs (L338). For question-answering (Section 5.3), on the other hand, we adopt Natural Language Generation metrics such as the F1 score computed on overlapping token (see also ``Metrics for LLM performance''). On the other hand, Equation 5 in Section 5.3 showcases an example of a VSA-based input representation for our training example, also adopting an augmenting strategy (L427).
> >
> > During LLM inference for the QA task, VSA-based probing in Section 5.3, we did not include the answer in the prompt (see the example shown in Equation 6).  A detail description of the QA-related setup can be found in Appendix I.
> >
> > # (1.d) Variance of F1 and EM statistics
> > Statistical variability for all metrics is measured using standard deviation. Although the LLM performance on the Exact Match (EM) metric suggests that it consistently fails to reproduce the precise target answers from SQuAD, resulting in EM scores close to random; the relatively high F1 scores indicate that the model’s generated text is still largely aligned with the reference answers.
> >
> > For instance, consider the SQuAD question “What is a choice that users of pesticides can make that will lower risks?” The ground-truth answer is “choosing less toxic pesticides”, whereas the model outputs “use less toxic pesticides”. Although the two expressions convey the same meaning, the prediction receives a low Exact Match score $(EM = 0)$ but a relatively high F1 score of $0.75$ (an overlap of 3 tokens out of 4).
> >
> > Importantly, we evaluate foundation language models without optimizing them for downstream tasks, such as through prompt engineering or fine-tuning, as this was not the goal of our study. Our purpose in testing the approach within a question-answering setting was to analyze the concepts extracted before and after the LLM’s generation in relation to question-and-answer features (L438), rather than to maximize the model’s performance on the downstream tasks.
> >
> > # (1.e) Datasets and reasoning-focused inputs
> > The synthetic and non-synthetic datasets serve different purposes, supporting the validation of our VSA-based methodology in two distinct contexts and yielding different types of insights.
> >
> > The *synthetic dataset* enables us to probe the LLM’s vector space using inputs that require different forms of reasoning, ranging from syntactic pattern recognition to factual and semantic associations. As shown in Section 5.2, representational richness varies substantially across input categories (L408), offering reasoning-focused insights.
> >
> > In contrast, the non-synthetic SQuAD dataset enables VSA-based probing of LLM text generation. Section 5.3 analyzes concepts before and after generation, revealing that failures often arise from losing focus on question-relevant concepts rather than lacking answer-related knowledge.

---

> ### Author Response · Authors · 2025-11-20
> **Rebuttal by Authors (part 4)**
>
> # (2) Weak contribution
> Our primary contribution is methodological; we introduce a novel probing paradigm overcoming some constraints of existing information decoding approaches (L73), *from the reliance of LLM's output vocabulary of DLA-based methods* to *the need of feature naming/filtering of SAE-based analysis.*  **Our work introduces methodological novelties**, from representing LLM's latent features with VSA encodings (Section 4.2), a novel ingestion algorithm for LLM embeddings (Section 4.3), a neural VSA encoder (Section 4.4), and VSA-based probing procedure (Section 4.5).
>
> The experimental results (Section 5) serve to validate VSA-based methodology with:
> 1. six diverse LLMs (from OpenAI's GPT-2 to MetaAI's Llama 4);
> 2. varied embedding dimensions (from 1024 to 5120);
> 3. two distinct experimental setting;
>       - model's state prior to its next-token prediction in input-completion tasks
>       - text generation in question-answering;
>  4. Textual inputs $\{\bar{\mathcal{S}}, \bar{\mathcal{Q}}\}$ different from those seen during training $\{\mathcal{S}, \mathcal{Q}\}$.
>
> In summary, our work advances information decoding in LLM vector spaces by probing the model’s full internal state **without the surface-level constraints of DLA-based methods** imposed by next-token representations and output vocabularies, while also **avoiding the post-hoc feature naming and filtering required in SAE-based methods.**
>
> # (Q1) Advantages over existing approaches
> *Our method neither mirrors the tuned-lens framework nor addresses layer–decoder misalignment*, *relies on a decoding matrix*, or *introduces a new vectorization schema* or a *decoder prior to training*.
>
> Our work proposes a **novel information decoding paradigm by overcoming some conceptual and experimental constraints of existing methods**, advancing interpretability of neural embeddings. While these aspects are currently discussed in Appendix G extensively, we will exploit the additional page of the revised version to include some remarks within the main text.
>
> The following table summarises the key advantages of our VSA-based methodology compared to existing probing paradigms:
> | Advantage of VSA-based methodology                                  | Limitation of               |
> |----------------------------------------------------------------------|----------------------------|
> | VSA features can be seen as a vocabulary with no practical constraints on its creation | DLA-based techniques       |
> | Enable targeting latent features that are not task-tied             | Conventional probes        |
> | Avoid additional naming and filtering stages for isolating relevant features | SAE-based analysis         |
>
> ### Limitation of supervised probes
> Due to space constraints, we briefly describe the limitations of supervised probes. As discussed in Section 2, evaluating decoding capabilities is challenging because probe outputs may reflect either actual information decoding or internal probe learning (L99). Hewitt and Liang (2019) address this via control tests, such as random baselines.
>
> To mitigate these confounds, we adopt several strategies. We validate our encoders on unseen syntactic structures ($\bar{\mathcal{S}}$ vs. $\mathcal{S}$; Section 5.1) and perform two control tests: a permuted baseline (randomly shuffled embeddings) and an unrelated baseline (extracting concepts irrelevant to the input; Table 5, Appendix F). Low precision scores (0.08–0.09) confirm that irrelevant or corrupted inputs produce meaningless outputs, as expected.
>
> Conventional supervised probes often target task-specific features (e.g., truthfulness, geographic info, syntax), whereas our VSA-based probe learns a vector transformation mapping embeddings into VSA encodings, **without directly targeting downstream tasks.**
>
> In summary, while our method is a supervised probe (L118), our conceptual and experimental strategies effectively separate information decoding from probe learning, demonstrating robust decoding of LLM embeddings.
>
> ### Limitation of DLA (lens-style methods)
> The primary constrains of lens-style methods, also referring to as Direct Logit Attribution (DLA) is their reliance on the model's output vocabulary. This leads *to rely solely on surface-level latent features encoded in LLM embeddings*, namely next-token representations. **On the other hand, VSA-based probing can target latent features of LLM embeddings defined by any chosen criteria**, such as contextually relevant concepts (L197), or potentially even multi-modal concepts (L477).
>
> Consequently, DLA-based probing identifies surface-level subspaces linked to next-token prediction, whereas **VSA-based probing recovers a broader latent subspace for each concept**, one that extends beyond surface- or token-level latent directions.

---

> ### Author Response · Authors · 2025-11-20
> **Rebuttal by Authors (part 5)**
>
> ### Limitation of SAEs
> As noted in Appendix G, VSA-based probing requires supervision to map to a proxy space but *enables top-down decoding by querying LLM embeddings with relevant concepts* (Eq. 4; Sections 5.2–5.3).
>
> In contrast, SAE-based analysis uses a bottom-up decoding process, naming all triggered features post hoc. This is constrained by DLA-based methods (limited to the model’s vocabulary) or input clustering (input-dependent), **revealing an unbounded set of features that require exhaustive naming and filtering.** VSA decoding could even be applied to SAE representations to improve naming.
> Unlike SAEs, which typically target a single hidden layer, our method examines nearly the entire residual stream, **avoiding layer-selection constraints through our ingestion algorithm** (Section 4.3).
>
> Overall, the interventions required by SAE, layer selection, feature naming, and post-hoc filtering, *impose greater conceptual and experimental constraints than learning a supervised mapping to the proxy space*.
>
> # (Q2) Interpretability of VSA-based space
> Our probing paradigm uses VSA encodings to inspect the LLM’s full internal state before next-token predictions (Section 5.2) and during text generation (Section 5.3) without intervening in inference. *This reveals conceptual richness in embeddings and tracks concept drift over autoregressive generation.*
>
> Meanwhile, DLA-based methods (also SAE-analysis with DLA-based feature naming) rely on projected next-token representations (L101), which restricts their probing capabilities to surface-level features and *undermines the depth of their interpretability analyses.*
> Appendix G.1 (L1010) shows that for the analogy "king is to queen as son is to" $\mapsto$ "daughter", using OLMo-2, our probe extracts the key-target concepts ("son" and "daughter"), while DLA produces no concepts. The model predicts the next token prediction as \texttt{?} with a softmax score of 0.06, followed by \texttt{father} (0.05); the target word has a rank of 37. **While DLA performance is prompt-dependent**, and could be *boosted via prompt engineering*, **our VSA-based insights are largely unaffected by exogenous factors such as prompt design** (L357).
>
> On the other hand, VSA-based space eliminates also the need for a post hoc feature-naming and feature-filtering stage, skipping one of the most problematic stages of SAE-based analysis (L115).  Thereby, the manual interventions required in SAE-based analysis, from feature naming/filtering to layer selection, *prevent the interpretability of SAE features from being fully automated or independent from experimental choices and data.*
>
> Lastly, *LLM prompting falls under instruction engineering, a domain distinct from our approach*, which is grounded in *interpreting the semantics encoded in the LLM vector space*,
> In Section 5.2, we analyse such **LLM failures by treating these failures not as errors to be corrected but as opportunities** to demonstrate the diagnostic and debugging capabilities of our probing framework.
>
> Although prompting and probing serve different purposes, *our VSA-based method reveals cases where the LLM’s final internal state correctly encodes the target concept, yet the model fails to produce it*, likely resulting from external factors such as prompt design, which does *not affect VSA-based probing*.

---

> ### Comment · Reviewer_dtKf · 2025-11-20
> **Response to Initial Rebuttal (part 1)**
>
> (1)
> (1) "Our experimental design and its corresponding results serve to validate [...] the **effectiveness** of VSA-based information **decoding* [...]"
>
> You argue that the LLM is a "subject of study" in this work. If this is the case, the aim of your work should be to draw some novel conclusions about the underlying mechanisms or behaviors of LLMs. Otherwise, your study on the subject of LLMs bears no fruit. Why, then, do you wish to evaluate the **effectiveness** of your approach.
> Conversely, the evaluation of the **effectiveness** of a strategy demands its comparison with other approaches, since "effectiveness'' is a subjective concept. For example, naively, I could design a "concept decoder'' by taking an LLM and prompting it with "what/which concept(s) is this text concerning?''. I could then fine tune the LLM, so that it is as proficient as possible in this extraction. This would likely give some notion of "maximum attainable effectiveness''. For this reason, baselines are important. By the same token, I can naively prompt an un-tuned LLM for the same, to get a notion of "minimum reasonable effectiveness''. On the downstream QA task, your method performs near-identically to this "minimum reasonable effectiveness" (ignoring the very strange variability values). While you say that your overall goal is not a downstream task such as QA but a more general interpretability outcome, is it not true that any classifier trained on some extraction of the LLM's hidden state can be turned to be a "concept extractor'' or "interpreter'' by naming concepts as multiple choices for it to choose from in a classification setting?
>
> "It does not attempt to address layer-decoder misalignment". Notably, you mention that your method **decodes** the LLM hidden state using a VSA-based technique. So, since you replace the standard decoding scheme, your method would indeed address misalignments in the standard LLM decoding.
>
>
> "it does not attempt to [...] compensate for the absence of explicit, application-specific training". Do you not train your representations on a specific task?
>
> 1(b) In the example you give, accomplish and accomplishment are functionally the same concept. Fundamentally, they share the same root, and are only syntactically (not semantically) different. I would not say that "accomplishment" is a half-right guess in the place of "accomplish"; it's a "pretty much right" guess and should be counted fully. With that said, this problem is why most experimental settings of this nature are designed to be multiple choice. In any case, short free-form generation is effectively multiple choice, since the language model is only ever likely to produce a few distinct answers if you sample it a fixed number of times.
>
> 1(c) "The objective of our work was not to boost LLM performance on downstream tasks via prompt engineering such as few-shot learning". You report accuracy on the downstream QA task to demonstrate its real-world utility: "To further validate our proposed approach in real-world scenarios, and beyond the controlled analogy task explored previously, we apply our methodology to a question answering task, using the popular SQuAD dataset." (line 415). The conclusion, based on the accuracies you present, is that your method is no more useful to real-world tasks than vanilla-prompting the LLM: you report 69% F1 on the LLM and 70% accuracy on your method. Why is your method, in this example, preferable to few-shot learning, self-consistency, or any of the many prompt/sampling-based methods which will surely increase the LLM accuracy meaningfully?
>
> 1(d) I think you missed the critical element here. You report EM on the LLM as having a mean of 0.52 with standard deviation of 0.5. This means that, under your assumption that the performance is Gaussian-distributed (or else you would not provide standard deviation), there is a 16% chance that EM is above 1.02. This isn't possible. The maximum EM score is 1.0. The same goes for your report of F1 on LLM.

---

> ### Comment · Reviewer_dtKf · 2025-11-20
> **Response to Initial Rebuttal (part 2)**
>
> (2)
> There is a near-infinite number of possible techniques to solve any problem. Most, however, are not counted, as they are not more useful than the most obvious or easiest solution. You claim, despite including experiments to demonstrate real-world utility on downstream tasks, that your work's goal is introduce a general paradigm. How is this paradigm useful? Does it provide insights that existing work does not? Is it practically useful? I don't think this paper managed to demonstrate either.
>
> (Q1)Each of these advantages of your VSA-based methodology that you mention are also advantages of prompting an LLM. LLMs do not have restricted vocabularies, their latent spaces are not task-specific, and they do not require naming and filtering stages during prompting. Why is your method more useful than LLM-prompting? In practice, it seems that it is not.
>
> You say that your method does not mirror the tuned-lens framework, and yet you replace the standard LLM decoding procedure with a learned network which results in the decoding in the hidden state (as you say, you propose a "decoding paradigm".
>
> You say you do not introduce a new vectorization schema or decoder prior to training, but you describe the building of a codebook and embeddings in the paper.
>
>
> ============
>
> As it stands, I think the only demonstrated claim of this paper is that there exists at least one VSA-based methodology which can be built to represent the information in the LLM's latent state. However, there is little reason provided as to why it is important that we know that this is true.
>
> I don't think the paper sufficiently demonstrates why we would use this method over either other probing methods or LLM prompting methods. As an exploratory work, the work does little to offer novel and widely insightful findings, and as a solution to a problem the work neither specifies a specific problem to be solved (i.e. with a Problem Statement) nor demonstrates that the proposed method does a better job than other methods even specifically of similar cost. Additionally, the reported empirical results seem at times impossible.

---

> ### Author Response · Authors · 2025-11-21
> **Response to Reviewer's comment (part 1)**
>
> # Problem Statement
> You are correct that the current version of our paper does not clearly articulate the research questions or the problem statement in relation to existing methods. *Because of space constraints, these elements were discussed only briefly in the main text.* We appreciate the opportunity to clarify these key aspects for you and the other reviewers, helping your final assessment.
>
> We will incorporate this clarification more explicitly in the revised version, specifically by fully rewriting the introduction section.
>
> ## Problem Statement
> Our primary research question is to **understand which concepts were represented internally during LLM reasoning**, specifically, *which concepts are encoded in the internal state of a language model* before completing analogy-style inputs and during text generation in a QA-like setting. Probing an LLM's final state to answer the following research questions related to LLM reasoning:
> - **RQ1**: Does the LLM represent of *all in-context concepts* when solving an analogy? (Section 5.2)
> - **RQ2**: Does the LLM rely on *key–value associations* to solve an analogy? (Section 5.2)
> - **RQ3**: Does the LLM represent *only the output concept*? (Section 5.2)
> - **RQ4**: Are there any identifiable *patterns across different input types*? (Section 5.2)
> - **RQ5**: Are there meaningful differences in behavior *across different LLMs*? (Section 5.2)
> - **RQ6**: What concepts are *present prior to text generation*, and which ones *emerge afterward*? How many *question-related* and *answer-related* concepts appear before and after text generation? (Section 5.3)
>
> The task is related to *general-purpose interpretability of LLM embeddings* rather than their misalignment with model's output (ELK); **focusing on which concepts are encoded rather than sorely assessing whether the target concept is present.**
> Specifically, we address the representation-learning problem, aiming at identifying latent features (concepts) presented in LLM embeddings with an information decoding technique.
>
> The *misalignment between these latent features and the model's outputs, is a related, yet subsequent problem*, addressed by Eliciting Latent Knowledge (ELK). Information decoding extracts human-interpretable symbols/features from neural embeddings, while *ELK exploits these extracted features to reveal misalignment* between model's state and its output.
> Information decoding is one possible approach to achieve ELK by narrowing the scope of information decoding on output-related latent features.
>
> Because **existing approaches to neural-embedding interpretation impose *significant limitations for addressing our research questions*, we introduce a new methodology.**
> Our approach leverages core properties of Vector Symbolic Architectures (VSAs) and hypervector algebra to overcome *several conceptual and experimental limitations of prior methods that hinder our ability to address the research questions*.
> Specifically, our new methodology addresses the following constraints of existing methods:
> ### (1) Direct Logit Attribution (DLA; lens-style methods)
> The primary limitation of Direct Logit Attribution (lens-style methods) for addressing our research questions, is its *reliance on token representations*, surface-level latent features.
> Their dependence on the model’s output vocabulary undermines our research goals in two key aspects:
> 1. it **restricts concept representation** to single-token words, and
> 2. by **directly pursuing the ELK-style misalignment problem**, it interferes with our broader objective of general-purpose concept extraction.
> ### (2) Sparse Autoencoders (SAEs)
> On the other hand, SAEs provide *general-purpose concept extraction* using a proxy space learned via sparse dictionary learning.
> The shortcomings of SAEs that limit their usefulness for our research questions are threefold:
> 1. **Unbounded and non-compositional feature space:** SAEs induce an effectively *unbounded* and *hard-to-control set of latent features*, making it difficult to ensure coverage, comparability, or principled organization of the feature space.
> 2. **Constrained and ambiguous feature naming:** SAE feature naming typically relies either on DLA; carrying forward its dependence on surface-level token associations and their limitations, or on input-driven clustering methods, which are highly task- and dataset-dependent. As a result, *the semantic grounding of features remains inconsistent and not compositional.*
> 3. **Layer-localized representations:** SAEs operate on a single hidden layer, requiring researchers to *choose a supposedly optimal model's hidden layer*. This choice is inherently task-specific and undermines our goal of analyzing the model’s internal state holistically across layers.

---

> ### Author Response · Authors · 2025-11-21
> **Response to Reviewer's comment (part 2)**
>
> ### (3) Conventional supervised probes
> While our approach meets the criteria of a supervised probe, we introduce the term *“hybrid supervised probe”* (L118) to describe a method that combines the *mapping framework of conventional supervised probes with the SAE’s ability to disentangle superposed subspaces through a sparsity constraint.*
> Our novel supervised mapping paradigm, with the sparsity constraint implemented via VSA encodings, allows us to combine
>  SAE-alike capabilities with interpretability of conventional probes:
> 1. **learning a sparse proxy space** (SAEs) rather than *directly the downstream task* (probes)
> 2. defining a **bounded feature space** (probes) rather than *unbounded* (SAEs)
> 3. querying a **learned proxy space for feature extraction** (SAEs) rather than *conventional classification* (probes).
> ### Hyperdimensional probe 🕵🏻
> As a result, we design a novel approach:
> |Characteristic|Component|Limitation of|
> |------------------------------------------------------------------------|------------------------|------------------|
> |representing unique, well-defined abstract concepts |VSA codebook|DLA|
> |inspecting the model's internal state without introducing layer selection|Ingestion algorithm|SAEs|
> |learning a sparse proxy space with a bounded feature space|neural VSA encoder|SAEs/probes|
> |querying LLM embeddings with a compositional feature space|hypervector algebra|SAEs|
> |recovering concept from neural encoding for general-purpose probing|VSA-based probing|DLA|
>
> While Appendix G describes our experimental comparison choices in detail, the DLA-based results indicates that:
> - DLA's dependence on surface-level, token-level latent features limits its capacity to provide deeper answers to the experimental questions.
> - DLA may introduce both exogenous and endogenous variability typical of token-oriented methods into the results.
>
> While our **VSA-based paradigm *advances information decoding in LLM embeddings***, it may also be *applied to SAE feature naming*, thereby enhancing also the **interpretability of the proxy space of SAEs.**
> ### Research questions
> Our work addresses the six experimental questions outlined above (RQ1-RQ6) with two additional methodological questions:
> - **RQ7**: Can LLM embeddings be faithfully represented using VSA encodings? (Section 4.4)
> - **RQ8**: To what extent can VSA-based probing recover information from the LLM vector space? (Section 5.2 and Section 5.3)
> # Concepts and partial-token matching
> While showing (1b; metrics for LLM performance) that *our experimental findings remain unchanged regardless of whether this strategy is adopted*; our problem statement might have clarified the rational of introducing this strategy. For answering the above research questions, **our work considers each unique and atomic concept, determining its contextual role in the input**:
> - Is it the target concept?
> - Is it from the in-context example?
> - is the key to the target analogy?
>
> Although they share the same conceptual root, “accomplish” and “accomplish**ment**” represent distinct concepts in our setting (verb+ment; analogies of morphological modifies), analogous to how “German” and “German**y**” occupy different roles within the domain of nationality-related analogies.
>
> Here is an example of LLM's metrics on the dataset used in Section 5.2 (see also Appendix M) :
> |Input from $\bar{\mathcal{S}}$|Domain|LLM output|P@k - LLM|Meaning|
> |-----------------------------------------------------------------|---------------|---------------|-------------|----------------------------------|
> |"*accomplish* is to *accomplishment* as *announce* is **to**"|verb+ment|announ|0.5|🤔announce? announcement?|
> |"*accomplish* is to *accomplishment* as *announce* is **to**"|verb+ment|announcement|1|🏅ANALOGY SOLVED|
> |"*Mexican* is to *Mexico* as *German* is **to**"|nationalities|German|0.5|🤔German? Germany?|
> |"*Mexican* is to *Mexico* as *German* is **to**"|nationalities|Germany|1|🏅ANALOGY SOLVED|
>
> *Precision@1 and Precision@5 captures softmax-level uncertainty* in LLM's next-token predictions; which would be absent in multi-token generation.
>
> # Purpose of QA task (RQ6)
> Although we report LLM performance on the QA task for completeness, the aim of testing our probe in a QA-style setting (Section 5.3) is different. **We apply our probing technique to natural-language generation setting (real-world scenario) to compare extracted concepts before and after text generation** (RQ6), rather than **focusing strictly on model's state for its next-token prediction** (RQ1 - RQ5).
>
> Accordingly, Section 5.3 applies our full pipeline (Section 4) to the QA setting by:
> 1. outlining SQuAD feature extraction (Section 4.1) in L418–L423;
> 2. constructing input representations (Section 4.2) in L424 and Equation 5;
> 3. reporting probe training performance (Section 4.4) in L427–L428;
> 4. performing VSA-based probing (Section 4.5) in L441–L442; and
> 5. offering concept-level insights for this generation-oriented setup in L446–L460.

---

> ### Author Response · Authors · 2025-11-21
> **Response to Reviewer's comment (part 3)**
>
> Our method aims to map latent features in LLM embeddings to VSA-based input representations. The reported *binary accuracy (0.70, L429) measures the encoder’s training performance; its ability to learn the transformation from LLM embeddings to VSA encodings in the SQuAD-based setting ($\mathcal{Q}$; Equation 5). It does not indicate performance on question-answering*. As noted in Section 4.4, this accuracy reflects the element-wise agreement between generated and ground-truth VSA encodings.
>
> In contrast, L443 reports the LLM’s QA performance using F1 and EM metrics computed on textual outputs generated by the language model.
> Thus, the *two reported scores serve entirely different purposes*: **F1 score measures downstream QA capabilities of the language model, whereas the binary accuracy reflects how well our trained model aligns LLM embeddings with their VSA encodings** during probe training (Equation 5).
>
> Afterward, **Section 5.3 answers our sixth research question** by applying our trained encoder before and after text generation on $\bar{\mathcal{Q}}$ (L446–L460; see also Appendix I), comparing the extracted concepts in relation to question-related and answer-related features (Figure 3).
> # Our contribution
> With your help, we can share the following twofold contributions of our work with the ICLR community next spring in Brazil:
> 1. **Methodological novelties** 🛠️ (RQ7 and RQ8):
>     - Proposing a **thorough and well-motivated approach for improving existing interpretability methods** (ufU2), overcoming some of their conceptual and experimental constraints, for example:
>         - DLA-based constraints of the model's output vocabulary, causing a reliance on surface-level latent features: next-token representations.
>         - Task-tied features of conventional supervised probes; avoiding directly targeting a downstream task by learning a vector transformation from LLM embeddings to VSA encodings, our proxy space.
>         - Important intervention in SAE-based analysis, from selecting the optimal model's hidden layer to post-doc feature naming/filtering.
>     - Applying **Vector Symbolic Architectures for probing** in a novel (gn8b, s5VU, ufU2) and interesting (s5VU) manner.
>     - Proposing a **novel method for reducing the dimensionality of the full LLM hidden space** (dtKf).
>     - Demonstrating the **generalizability of the VSA-based probing** by testing *two distinct experimental settings* (ufU2), and across *tasks of varying complexity*, *model sizes,* and *families* (gn8b).
> 2. **Non-trivial insights into LLM embeddings** 🔍(RQ1-RQ6):
>     - Section 5.2 **reveals differences in conceptual richness** (RQ1-RQ3) **across input types** (RQ4; L408) and **LLMs** (RQ5; L393).
> Uncovering, for instance, OLMo-2's conceptual richness in contrast to GPT-2's, and the majority of conceptually-empty representations in mathematical analogies.
>     - Section 5.3 **uncovers distinct concept-related patterns during text generation** (RQ6) that point to *potential sources of the model’s incorrect answers* (L452).  Probe's concept extraction before and after text generation suggests that LLM failures may stem from losing focus on question-related concepts rather than a lack of answer-related knowledge.
> # Distribution of the EM score
> You are correct that interpreting the reported mean and standard deviation under a Gaussian assumption would assign nonzero, and in this case, substantial, probability outside the valid interval $[0,1]$. Since EM cannot exceed 1, such an interpretation would indeed be inconsistent.
>
> Below, we clarify why the **reported values are nevertheless mathematically valid, why the *Gaussian interpretation leads to the apparent contradiction***, and how we correct the presentation to avoid this confusion.
> ## The reported mean and SD do not imply a Gaussian assumption.
> The values we provided (e.g., mean $\mu \approx 0.52$ and sample standard deviation $\sigma \approx 0.50$) were computed directly from empirical per-example scores. They are *sample statistics*, not parameters of a
> Gaussian model. No normality assumption was intended.
>
> For a *bounded binary-like metric such as EM*, the sample variance naturally approaches its theoretical maximum $p(1-p)$ when the system is correct on roughly half the examples. For example, with $p = 0.5165$, the Bernoulli variance is
> $$ p(1-p) = 0.5165 \times 0.4835 \approx 0.24998 $$
> which implies an SD of $ \sqrt{p(1-p)} \approx 0.49998 $.
>
> This matches our empirical SD and is completely consistent with a bounded $0$--$1$ variable:
> |Statistic|F1 score|EM|Is mentioned|
> |-----------|------------|------------|------------|
> |COUNT|10000|10000|10000|
> |MEAN|0.68857987|0.5165|0.6755|
> |STD| 0.382711801| 0.499752664|0.468211141|
> |MIN| 0|0|0|
> |25%|0.347|0|0|
> |50%|1|1|1|
> |75%|1|1|1|
> |MAX|1|1|1|
>
> |Metrics|0|1|Total|
> |------------|-------------------|-------------------|---------------|
> |EM|4835 (48.3%) |5165 (51.7%) | 10000 (100%)|
> |Is mentioned| 3245 (32.5%)|6755 (67.5%)| 10000 (100%)|

---

> ### Author Response · Authors · 2025-11-21
> **Response to Reviewer's comment (part 4)**
>
> ## Why the Gaussian interpretation leads to an "impossible'' result.
> If one *does* treat EM as Gaussian with $\mu = 0.52$ and $\sigma = 0.50$, then the probability of exceeding $1$ would be
> $$
> P(X > 1) = 1 - \Phi\!\left( \frac{1 - 0.52}{0.50} \right)
>         \approx 1 - \Phi(0.96)
>         \approx 0.16,
> $$
> i.e., as you mentioned, roughly 16\%. Since EM is bounded above by $1$, such a model is clearly invalid.
>
> **This contradiction arises only *under the normality assumption***; which we did not intend to make.
>
> ## Why SD $\approx 0.50$ is common for EM.
> For a Bernoulli variable,$ \mathrm{SD} = \sqrt{p(1-p)} $
>
> which is maximized at $p = 0.5$ and equals $0.5$ at that point. Thus, whenever a model achieves EM close to $50\%$, the empirical SD will naturally be close to $0.5$.
>
> *This behavior is inherent to the metric and does not reflect any distributional assumption.* Rounding then could have further misled.
>
> ## Correction to avoid implying a Gaussian model.
> To prevent confusion, we will revise our reporting methodology:
> 1. For EM, we will treat the score as a binomial proportion and report standard errors $\sqrt{p(1-p)/N}$ together with bootstrap confidence intervals.
> 2. For F1, which is not Bernoulli-distributed, we will report bootstrap confidence intervals computed from per-example scores.
> 3. We will also report the binary metric “*is_mentioned*,” which uses a looser check to determine whether the generated text contains the ground-truth answer (rather than requiring an Exact Match). This metric is better suited to our needs.
> 4. We will avoid presenting EM or F1 as ``mean $\pm$ SD'' since this formatting implicitly suggests a Gaussian interpretation.
>
> ## Summary
> - The reported values (e.g., mean $\approx 0.52$, SD $\approx 0.50$) are fully consistent with the bounded nature of EM and F1 *when interpreted as sample statistics*.
> - The *apparent inconsistency arises only if a Gaussian distribution is assumed.*

---

> > ### Comment · Reviewer_dtKf · 2025-11-25
> > **Statistical Significance Discussion**
> >
> > OK. When we provide a measurement of variability of our metric, it is important to consider its actual meaning to the reader. Confidence intervals are clear: some % of the time, we will be below/above these values. The simplest understanding of  standard deviation is in its natural context, the gaussian. Reporting CIs instead of SD will solve this issue for me.

---

> ### Comment · Reviewer_dtKf · 2025-11-25
> **Discussion regarding outlined RQs, clarity, their representation in the paper**
>
> The research questions (RQs) you have outlined in your response are helpful, and certainly help the reader understand the author's aims with this work. With that said, they do not actually appear in the paper. 5 research questions are attributed to section 5.2, but only two distinct paragraphs are written, and none of the research questions appear explicitly. This is one of the reasons I (and, it seems, the other reviewers) am having such a hard time understanding this work and its aims. Each research question should be stated clearly, and experimental results should be plainly described in terms of the ways in which they answer the question.
>
> The second reason that this paper is difficult to understand, in terms of its aims and claims, is that the underlying insight or meaning behind each result is left unsaid. You say "Section 5.2 reveals differences in conceptual richness (RQ1-RQ3) across input types (RQ4; L408) and LLMs (RQ5; L393). Uncovering, for instance, OLMo-2's conceptual richness in contrast to GPT-2's, and the majority of conceptually-empty representations in mathematical analogies.". What is the significance of the fact that most conceptually-empty representations are in mathematical analogies? Does it tell us something about the distinction between math and other reasoning forms? More generally, what does it mean that LLM representations are conceptually empty? These types of questions are answered very naturally if research questions are introduced explicitly, since typically they will come in the text with some form of motivation so that we understand what the answer will tell us.
>
> The paper needs a major re-structuring to highlight clearly these research questions, both in the introduction and experimental sections. Experiments need to be both proposed and answered in terms of the RQs. The conclusion then must also be re-written. While the authors have come a long way in terms of making the intended message of their work clearer, they have not proposed a new version of the text. A new version of the text including these changes, however, would be massively re-written, to the point that it should be a new submission.

---

> ### Comment · Reviewer_dtKf · 2025-11-25
> **Discussion of Concepts vs Parts of Speech**
>
> You said that "Although they share the same conceptual root, “accomplish” and “accomplishment” represent distinct concepts in our setting (verb+ment; analogies of morphological modifies), analogous to how “German” and “Germany” occupy different roles within the domain of nationality-related analogies."
>
> Since your setting is synthetic and, as you state, solely for the purpose of scientific investigation of your research questions, you can design it to be as appropriate as possible to your aims. In particular, you stress that you want to learn conceptual representations rather than "surface-level representations'' which you claim DLA learns. In my view, identification of part-of-speech (verb vs noun) is a superficial task relative to conceptual identification, since one is syntactic and the other is semantic. I don't see why you would expressly choose to evaluate model discernment of part-of-speech when your goal is to learn semantically meaningful concept-representations.

---

> ### Author Response · Authors · 2025-11-25
>
> # Clarification on concepts
> Our methodology treats concepts as atomic and abstract, **associating concepts such as *Germany*, *German*, *accomplishment*, and *accomplish* to their corresponding latent feature/direction**. In regards of their comprehensive semantics encoded in LLM vector space, not sorely their part-of-speech, such as being a noun or verb.
>
> **This is the key advantages of our VSA-based methodology**: mapping latent feature/direction/subspace in LLM embeddings to a vocabulary (VSA codebook $\Phi$) with no practical constraints on the cardinality, type and source of its symbols.
>
> We refer to different roles in terms of **their contextual position** rather than their linguistic aspects:
> 1. for analogy-style inputs ($\bar{\mathcal{S}}$) in Section 5.2
> |TEXT|IN-CONTEXT KEY|IN-CONTEXT VALUE | KEY | TARGET | EXTRACTED CONCEPTS
> -------| ---| -----| ---- |-----|--
> "Mexican is to Mexico as German is to"|Mexican| Mexico  |German|Germany|🏅$\phi_{German}$ (KEY), $\phi_\text{Germany}$ (TARGET)
> "Euclid is to Greek as Galilei is to" | Euclid | Greek |Galilei| Italian|🏅 $\phi_{Galilei}$ (KEY), $\phi_\text{Italian}$ (TARGET)
> "Kabul is to Afghanistan as Berlin is to" | Kabul |Afghanistan |Berlin | Germany | 🕵🏻$\phi_{Kabul}$ (IN-CONTEXT KEY), $\phi_\text{Afghanistan }$ (IN-CONTEXT VALUE)
>
> 2. for QA-oriented inputs ($\bar{\mathcal{Q}}$)  in Section 5.3
> |QUESTION| ANSWER |QUESTION CONCEPTS | ANSWER CONCEPTS| CONCEPTS EXTRACTED BEFORE TEXT GENERATION| CONCEPTS EXTRACTED AFTER TEXT GENERATION
> ----| ----| ---| ----- |----| ---
> "What do laboratories *try* to *produce* hydrogen from?" |"*solar* *energy* and *water*"|try, produce, hydrogen | solar, energy, water|$\phi_{\text{try}}, \phi_{\text{produce}}, \phi_{\text{hydrogen}}$ for question and $\phi_{\text{solar}}$ $\phi_{\text{water}}$ for answer|$\phi_{\text{produce}}$ for question and $\phi_{\text{solar}}, \phi_{\text{energy}}, \phi_{\text{water}}$ for answer
>
> While the example in our previous response was presenting for better explaining partial-correct matching strategy, **we do create a dataset with analogy-style tasks with varying degree of abstraction** (see Appendix M): from linguistic analogies and semantic relations to factual analogies.

---

> ### Author Response · Authors · 2025-11-25
>
> Thank you for all the *constructive* feedback.
> Presenting a paper that includes both *methodological* and *experimental contributions* can be **challenging for readers and authors** alike.
>
> We are **currently preparing a revised 10-page version**, incorporating as many clarifications as possible to address the reviewers’ comments, and plan to upload it **before the end of the rebuttal period:**
> 1. **Clarification of the problem statement**: to better frame the objectives of our work (dtKf, gn8b, s5VU), we have *fully rewritten the introduction, related work, conclusion sections* and *abstract*. It now explicitly includes our research questions and the underlying problem statement. Additionally, we clarified the distinct purposes of the analogy-style synthetic dataset versus the QA-style non-synthetic datasets (dtKf).
>
> 2. **Better framing of the advantages of VSA-based methodology**: in the revised introduction and related work section, we provide a *clearer rationale for using VSA probing* and highlight its characteristic overcoming the key constraints of prior methods (dtKf, gn8b, s5VU).
>
> 3. **Experimental comparison moved to the main text**: we relocated our *experimental comparison from Appendix G.1 to Section 5.3* (ufU2) to better contextualize the comparison (dtKf, s5VU) and emphasize non-trivial insights from the DLA-based analysis. Table 7 has been moved into the main text, and DLA-based scores are now integrated into Figure 2.
>
> 4. **Clarification of Table 1**: the *description of the concept-extraction results* in Table 1 (gn8b, s5VU, ufU2) has been improved and expanded in Section 5.2.
>
> 5. **Clarification of LLM metrics**: we have enhanced the *description of LLM metrics throughout the paper*. This includes clarifying the rationale behind the partial-matching score (s5VU) in Section 5.1, separately describing metrics for the QA-style setup (dtKf) in Section 6, and providing improved reporting of statistical variance for LLM metrics (dtKf).
>
> 6. **Unidirectional transformation listed as a limitation**: we added a discussion of the reliance on *unidirectional transformation to the limitations section* (Section 7), noting that it currently prevents causal validation (gn8b, s5VU).
>
> 7. **Improved presentation of QA-style setup**: to improve the manuscript’s logical flow (gn8b), *the validation of our full pipeline within the QA setup has been moved from Section 5.3 to its own dedicated section* (Section 6). This reorganization allows for clearer distinction between experimental setups and better presentation of inputs, LLM metrics (dtKf), and the text-generation process (s5VU).
>
> While our revised version will provide a clearer framing of the problem statement and our underlying research questions/answers, **the emphasis of our work is necessarily on describing the methodology rather than on the insights, given the novelty of applying VSAs to neural probing in two distinct experimental settings.** Due to space constraints, a more in-depth interpretation of the experimental findings remains an open challenge for future research.
>
> We sincerely *appreciate your time and consideration*. We believe that the revisions addressing your feedback, as well as that of other reviewers, have substantially improved the manuscript.
> We hope that **our *revised version* will support your overall evaluation of our submission**.

---

### Official Review · Reviewer_ufU2 · 2025-10-23

**Soundness:** 3
**Presentation:** 3
**Contribution:** 3
**Rating:** 6
**Confidence:** 3

**Summary:**

The authors present an approach to probing that is based on Vector Symbolic Architecture (VSA) and avoids the limitations that direct logit attribution (DLA)/logit lens-based methods and sparse autoencoders (SAEs) have. Their approach requires three steps: (1) computing the input representations of transformers by feeding inputs into the model, (2) training a supervised model to map token embeddings from a transformer into VSA encodings, and (3) examining the VSA encodings and extracting the concepts embedded into them. Performance with the hyperdimensional probe shows that probing reveals concepts more accurately than LLM next token prediction on the analogy task.

**Strengths:**

The paper is well motivated with the goal of improving over existing interpretability methods. The paper applies a unique approach to interpretability by techniques in vector symbolic architectures to analyze a transformer's residual stream. Furthermore, the approach is also applied to two settings which enables understanding of the generalizability of the approach. References to the appendix are comprehensive and the paper is generally thorough.

**Weaknesses:**

1. A comparison against either probe or SAE should be included in the main paper, if the claim is that the hyperdimensional probe outperforms (in some way) these existing methods. Although it is noted in the appendix, the baseline is important enough that it should be included in the main text.
2. It is unclear in Table 1 what each of the extracted concepts mean, from first glance. For instance, Key Values is only briefly mentioned in the caption, and as is Out-of-context. It would be better to include a section that also elaborates on the evaluation (and each "extracted concept") since this is a core finding of the paper.

**Questions:**

1. How are the features decided for 5.3 for question answering? It seems like the approach does not seem to be easy to apply to different tasks since it is not obvious what features to enumerate when the format is not analogy-style.
2. How compute intensive is constructing the hyperdimensional probe? Meaning, how many GPU hours are used per probe + dataset?
3. How does this approach generalize out-of-domain on datasets that the hyperdimensional probe is not trained on?

---

> ### Author Response · Authors · 2025-11-20
> **Rebuttal by Authors (part 1)**
>
> We appreciate you noting that our work:
> 1. is *thorough* and *well-motivated* in improving existing interpretability methods, such as avoiding the limitation of SAEs and lens-based methods;
> 3. *adopts a unique interpretability perspective* to analyze the transformer’s residual stream;
> 4.  *demonstrates the generalizability* of the proposed approach by testing two distinct experimental settings
> 5. includes *comprehensive references in the appendix*.
>
> # Experimental comparison in the main text
> We apologize that, due to space constraints, the main text focused on demonstrating our approach’s generalizability: next-token probing in input-completion tasks (Section 5.2) and text generation in question-answering (Section 5.3).
> In the revised version, we will *move the experimental comparisons from Appendix G to a new main-text section (5.4) using the additional page.
> # Interpretation of Table 1
> We apologize; due to space constraints, we focused on presenting high-level insights from Table 1 in the main text and overlooked a detailed description of the table.  Prompted also by the comments from others (s5VU), we will *provide a clearer explanation of the table’s components in Section 5.2 of the revised version*, making use of the additional page. We also plan to incorporate relevant table-related considerations currently included in the extended limitations section in Appendix A.
>
> While Section 4.5 and Equation 4 describe the procedure for concept extraction using the unbinding operation, **Table 1 shows the final results of this extraction procedure**. Each row indicates the set of concepts yielded by our probe via the concept extraction procedure (Section 4.5) for all the six LLMs (columns). A concept set groups the concept used for the unbinding operation, and the concepts found after applying unbinding (see Equation 4). For example, the first entry "Key$|$Target'' indicates the probe extracts the set of concept representing the key and the value of the target analogy (e.g., $\{\phi_\textrm{mexico}, \phi_\textrm{peso}\}$ in Equation 4) by using the "Key'' to perform the unbinding operation ($\phi_\textrm{mexico}$) which yielded the ``Target'' concept ($\phi_\textrm{peso}$).  Lastly, the cell indicates the relative frequency of a concept set for a specific language model: for example, the probe extracts the key and value of the target analogy (Key$|$Target) in 85.4\% of the instances ($\bar{\mathcal{S}}$) for Llama 3.1.
>
> # (Q1) Feature selection for question-answering
> For the question–answering dataset (SQuAD), we extract features relevant to each question-answer pair by examining their lexical semantics. Specifically, we use two lexical knowledge bases, WordNet and DBpedia (L422), to identify words with well-defined semantic properties: nouns, verbs, and adjectives. This aims to focus on the most pertinent concepts presented in questions and answers. A word is considered to have a relevant lexical semantics when it appears in at least one of the two knowledge bases; that is, when a corresponding entry is found in WordNet (i.e., synset) or an entry exists in DBpedia.
>
> For example, in the instance shown in Equation 5, name and ship are nouns, Napoleon and The Black Sea are proper nouns, and send is the lemmatized predicate present in the question.
>
> # (Q2) Computational cost of our methodology
>  We describe the computational workload of our methodology in Appendix Q.  The computational demand of our work is split into two parts: *LLM inference* (exogenous, Section 4.3) and *the training and probing stages of our method* (endogenous, Section 4.4 and 4.5).
>
> The most computationally demanding task was running the LLMs. We tested six models (355M–109B parameters) in inference mode, caching embeddings for training and probing them dynamically, using 1–3 NVIDIA A100-80GB GPUs. No quantization was applied. In contrast, our VSA-based method is lightweight; training the 55M–71M parameter neural VSA encoder on a single GPU was the most resource-intensive step and could run on smaller GPUs.
>
> Regarding GPU usage, we trained our encoder for approximately 8 hours on each LLM’s embeddings, though the process could have been shortened with a less conservative early stopping criterion or by reducing the amount of training data. Afterwards, VSA-based probing consists of simple vector multiplications (unbinding), after loading the heavy LLM and our lightweight trained neural VSA encoder into memory (from 800 MB of the 55M version to 1 GB of the biggest one). The time cost of probing is thus comparable to simple LLM inference with a slight increase due to feedforward our lightweight trained model, with the time demand for vector multiplications being negligible. Accordingly, the GPU hours for probing depends mainly on the amount of test data and model size; for example, we used around 95 hours of GPU computation for probing embeddings of Llama 3.1-8B on $\bar{\mathcal{S}} \approx 10^5$, processing each instance in around 3 seconds.

---

> ### Author Response · Authors · 2025-11-20
> **Rebuttal by Authors (part 2)**
>
> # (Q3) Out-of-domain generalizability
> The neural VSA encoder, a central component of the hyperdimensional probe (Section 4.4), is designed to map concepts with their corresponding neural representations, specifically, to map VSA encodings onto the appropriate directions or subspaces within LLM embeddings. To achieve this, the encoder $\mathcal{M}$ *must be exposed to a few examples of the latent features being targeted*, whether drawn from in-domain or out-of-domain distributions. In other words, **the hyperdimensional probe cannot extract latent features (i.e., concepts) that it has never encountered.**
>
> For instance, one might focus on toxicity by training on toxic content from social platforms, thereby mapping textual inputs to toxicity-related concepts (see Appendix H.1).  An example could be the phrase “You are a pathetic excuse for a human just like the rest of your kind”, which would be associated with the concept $\phi_\text{discrimination}$. Once trained, the encoder can then be applied to out-of-domain text to identify potential toxicity-related concepts.
>
> Moreover, we evaluate our probe on textual inputs $\bar{\mathcal{S}}$ that differ from those used during training $\mathcal{S}$ (L332). This allows us both to control for confounding effects and to validate the probe’s out-of-domain performance.  While $\mathcal{S}$ and $\bar{\mathcal{S}}$ share the same underlying semantic concepts, they consist of distinct texts with a different syntactic structure. Consequently, they can be seen as out-of-domain inputs with respect to the LLM embeddings, the inputs processed by our trained encoder.
>
> Importantly, *the probe’s performance is not affected by the size or construction strategy of the codebook.* The codebook, essentially the list of all features, functions like an alphabet, unconstrained in terms of cardinality, type, or source of its symbols, thanks to the properties of VSA.
>
> However, as with all interpretability probes, *its effectiveness ultimately depends on the type of feature being targeted*. Consequently, the strategy used to construct input representations (i.e., VSA encodings) directly impacts probe performance.  For example, if the probe attempts to map or extract latent features that are unlikely to be encoded in the LLM’s vector space, the resulting outputs may be unreliable or meaningless (see the “unrelated baseline” in Appendix F).

---

> > ### Author Response · Authors · 2025-11-27
> >
> > Can we offer ***any* additional clarification**?
> >
> > If our responses, such as the clarified *interpretation of Table 1* and the *discussion of potential out-of-domain generalizability*, address your main concerns, we hope they provide a clearer understanding of our contributions.
> >
> > **We are also *preparing a revised version*, scheduled for upload within the next few days**, which will incorporate our *experimental comparison* into the main text and include several improvements prompted by your and others’ feedback, including, just to list a few, a *fully rewritten introduction and related work sections*, as well as an enhanced *explanation of the results in Table 1.*
> >
> > Thank you for the time and care you have devoted to evaluating our work; we hope that *our clarifications* and the *forthcoming revised version* support your overall assessment.

---

### Official Review · Reviewer_s5VU · 2025-10-23

**Soundness:** 2
**Presentation:** 1
**Contribution:** 3
**Rating:** 4
**Confidence:** 3

**Summary:**

This paper proposes Hyperdimensional Probe: a method based on Vector Symbolic Architectures to extract concepts from a latent. With controlled settings the authors validate the proposed approach and show that it can successfully extract meaningful concepts.

**Strengths:**

* This paper tackles a very important problem of Eliciting Latent Knowledge (ELK), although in a controlled setting where the concept space is known.
* I think the proposed method, building on ideas from Vector Symbolic Architectures, is both interesting and novel. This approach may unlock new ways for concept extraction or interpretability in general.

**Weaknesses:**

* Poor readability: I had a very hard time reading parts of the paper. Even after multiple passes I couldn't fully grasp certain sections. See my clarifying questions below.
* Missing obvious baselines and comparisons to prior work on ELK. See Question 1 below.

**Questions:**

1. Baselines: Eliciting Latent Knowledge (ELK) being a very important problem in interpretability, has been widely studied in the literature. [nostalgebraist, 2020](https://www.lesswrong.com/posts/AcKRB8wDpdaN6v6ru/interpreting-gpt-the-logit-lens), [Belrose et al, 2023](https://arxiv.org/abs/2303.08112), [Ghandeharioun et al, 2024](https://arxiv.org/abs/2401.06102), [Hernandez et al, 2024](https://arxiv.org/pdf/2308.09124) (among others) have all proposed different methods for ELK. The authors should compare their method against at least some of these baselines to better position their contributions.

    1.1. Correct me if I am wrong, but *probing@1* corresponds to accuracy in detecting the correct concept, right? If so, it should be easy to compare against the mentioned baselines.


2. Line 20-21: *"This probe combines the strengths of SAEs and conventional probes while overcoming their key limitations."*: This is a strong claim and I am not sure I understand the authors' justification for this.

    2.1. For SAEs, Hyperdimensional Probes overcome the feature labeling part of the limitation, while introducing the need for supervision to learn mapping to the proxy space, exchanging one limitation for another.

    2.2. For probing: Line 37-38 says "... though their decoding capabilities have been debated". I basically didn't get this statement. Which part of [Hewitt and Liang, 2019](https://arxiv.org/abs/1909.03368) are you referring to? And how does your approach overcome this?

    2.3. [Belinkov, 2021](https://arxiv.org/abs/2102.12452) lists some shortcomings of probing methods, such as the need for causal validations. I believe that Hyperdimensional Probes also may suffer this shortcoming. However, I am not sure what could be done to test this as I don't think the mapping to proxy space, $\mathcal{M}$, is invertible. But I would like to hear the authors' thoughts on this.


3. Line 244-245: *"We apply sum pooling, which consists of summing all centroid embeddings."*: In modern transformer LMs, latents in later layers tend of have higher magnitudes compared to earlier layers. So, summing all latents may bias the resulting vector towards later layers. Was there any normalization step involved to address this?

4. Assigning 0.5 precision score for multi-token targets (lines 345-346) definitely underestimates the actual performance of the LM. Why not decide correctness based on the first token only? Or, even better, just generate multiple tokens and check if the full target is generated?

    4.1. I also think the probe performance should be reported in terms of the samples where the LM generated the correct target. After all, the goal is to extract concepts that the LM actually uses in its computation. So, reporting probe performance on samples where the LM failed to generate the correct target seems misleading.

5. I didn't understand Table 1 and. Adding illustrative examples for each of the extracted concepts (what was used for unbinding, what was the target, ...) would help a lot.

    5.1. I also didn't understand the subsequent discussion around Table 1. For example, in Lines 397-399 the authors discuss how GPT-2 struggles with the task. This is probably because I didn't fully understand Table 1.

6. In Section 5, how were the VSA encodings compared? Just cosine similarity? Was there a threshold for concept detection?

7. In Figure 3 and Lines 456-460, you seem to be expecting the concepts to persist even after text generation. Why is that? Doesn't it make sense that once the concept is used in generation, it is not needed anymore? Also, how many tokens were generated? For the given example did the authors mean that the LM deletes the input concepts (*hydrogen*) immediately after generating (*water*) leading to errors?

---

> ### Author Response · Authors · 2025-11-20
> **Rebuttal by Authors (part 1)**
>
> Thanks for all your thoughtful questions. We appreciate you noticing that our proposed approach:
> 1. builds on ideas from Vector Symbolic Architectures in a *interesting and novel fashion*
> 2. tackles an *important problem in interpretability*.
>
> We apologize that, due to space constraints, *some methodological and experimental details were moved to the Appendix, which may have affected clarity*; given the multiple VSA-related concepts introduced.
>
> In the revised version, we will use the extra page to restore key content to the main text and expand the discussion.
>
> ---------
> # (Q1) Baselines and comparison with other methods
> *We did contrast our VSA-based results against those yielded by a lens-style method*, also referring to as Direct Logit Attribution (DLA).
> The methods of **Nostalgebraist (2020)**, and **Belrose et al. (2023)** *build on Direct Logit Attribution (DLA)*, projecting the LLM’s embedding space onto vocabulary space either directly via the model's *unembedding matrix* (Logit Lens; $\mathcal{W}$ in the Nostalgebraist’s work), or via *learned affine transformation* (Tuned Lens; Belrose et al., 2023).
>
> As noted in Section 2 (L102), *lens-style methods, grouped under DLA, decode LLM vector spaces into the model’s output vocabulary for inspecting output distributions*. Belrose et al. (2023) propose the Tuned (logit) Lens, training an affine probe per hidden layer. While more reliable than the vanilla logit lens, it remains token- and layer-oriented, tied to the model’s vocabulary, and *inherits many conceptual limitations of DLA-based approaches*.
>
> **In our experiments, we contrast our VSA-based results against those yielded by the Direct Logit Attribution (L327), also known as Logit Lens by Nostalgebraist (2020); reporting a lens-style baseline comparison in Appendix G.**
> We apologize that, due to space constraints, we prioritized demonstrating the generalizability of our approach in the main text, from model's state for its predictions (Section 5.2) to text generation (Section 5.3). Key DLA insights are noted in the main text (L328), while full analysis and comparisons are in Appendix G, with lens-like visualizations in G.3.
>
> The table shows the probing@1 metrics computed using the DLA-based outputs (see Table 7 in Appendix G.1).  Our method improves average performance by 30%, but the approaches can capture different latent features: DLA focuses on surface-level next-token subspaces, while VSA probes broader concept subspaces beyond token-level directions.
> | MODEL                   | DLA (P@1) | VSA (P@1) | Δ        |
> |-------------------------|---------------------------------|---------------------------------|----------|
> | Llama 4 Scout, 17B-16E  | 0.773 ± 0.41                    | 0.866 ± 0.34                    | +9.3%    |
> | GPT-2, medium           | 0.300 ± 0.45                    | 0.692 ± 0.46                    | +39.2%   |
> | Pythia-1.4b             | 0.413 ± 0.49                    | 0.778 ± 0.42                    | +36.5%   |
> | Llama 3.1-8B            | 0.467 ± 0.49                    | 0.891 ± 0.31                    | +42.4%   |
> | Phi 4                   | 0.585 ± 0.49                    | 0.887 ± 0.32                    | +30.2%   |
> | OLMo-2                  | 0.714 ± 0.45                    | 0.879 ± 0.33                    | +16.5%   |
> | **AVERAGE**             | 0.542 ± 0.46                    | 0.832 ± 0.35                    | +29.0%   |
>
> **Hernandez et al. (2024)** propose Linear Relational Embedding (LRE), which maps a subject token embedding to its corresponding object embedding via an affine transformation, capturing relations like “playing the instrument.” In contrast, our method analyzes a single token’s embedding (the last one) to inspect the full internal state, capturing all concepts prior to prediction (Section 5.2) and during generation (Section 5.3).
>
> **Ghandeharioun et al. (2024)** use activation patching (Patchscopes) to probe LLM embeddings, covering both output distribution inspection and object-oriented feature extraction. Unlike these token-oriented methods—which depend on the model’s vocabulary and can be influenced by prompt design or tokenization—our VSA-based approach offers a broader perspective on latent concept representations.
>
> We frame our method as an information-decoding approach, extracting interpretable semantics from LLM embeddings. Unlike ELK methods, which address alignment between embeddings and outputs, information decoding focuses on uncovering the full semantic content of embeddings. ELK can be viewed as a narrower use case of information decoding, targeting output-related knowledge. While Figure 2 demonstrates ELK-style misalignment, the main contribution of the hyperdimensional probe is extracting concept representations from LLMs (Table 1). This full extraction allows comparing question- and answer-related concepts, revealing that QA errors often stem from losing focus on question-related concepts rather than lacking knowledge (L452).

---

> ### Author Response · Authors · 2025-11-20
> **Rebuttal by Authors (part 2)**
>
> The following table summarises the key differences between the mentioned approaches and ours:
> | Authors | Proposed approach | Paradigm | Output |
> |---------|-----------------|----------|--------|
> | Nostalgebraist, 2020 | Logit Lens (DLA); projects the LLM’s embedding space onto its vocabulary space through the model's unembedding matrix | Direct Logit Attribution | Next-token representation |
> | Belrose et al, 2023 | Tuned Lens; a trained probe, based on DLA, that learns an affine mapping from the embedding space to the vocabulary space | Direct Logit Attribution | Next-token representation |
> | Hernandez et al, 2024 | Linear Relational Embedding; a learned relational mapping between embeddings related to a subject and its object | Relation Learning | Vector transformation between token embeddings |
> | Ghandeharioun et al, 2024 | Patchscopes; a unifying framework that alters an LLM inference to decode the semantics of LLM embeddings | Activating Patching | Token representations / Natural Language Text |
> | **OUR** | Hyperdimensional probe | VSAs and hypervector algebra | Set of encoded concepts |
>
> In summary, we have *already reported a comparison with a lens-style baseline in the appendix section*, and its content, and the above tabular data, *will be integrated in the main text of the revised version, exploiting the additional page.*
> Specifically, we will move our experimental comparison from Appendix G into the main text as a new section, numbered 5.4.
> We will also better frame our contribution, including also some remarks on the dichotomy between information decoding and ELK.
> # (Q2) Overcoming limitations of exiting approaches
> ### (Q2.1) Limitation of SAEs
> While our proposed approach has limitations as any other methods (see also Appendix A for an extended discussion), **the limitations of SAE-based analysis and VSA-based probing have a different impact on the conceptual and experimental setting for extracting latent features from LLM vector space.**
>
> As noted in Appendix G; while requiring supervision to learn mapping to the proxy space, it enables a *top-down decoding strategy* by querying LLM vector space (Equation 4) with concepts relevant for the probing objective: next-token concept in Section 5.2; question-related and answer-related concepts in Section 5.3.
>
> On the other side, *SAE-based analysis relies on a bottom-up process* that names all triggered features post hoc. This naming stage is currently constrained by two strategies (L112): DLA-based methods, limiting features to model's output vocabulary; and input clustering, making the naming process rather input-dependent. *This bottom-up approach reveals an unbounded set of latent features without relevance filtering, requiring exhaustive feature naming and additional filtering to isolate those aligned with bounded concept framework.* One might even apply the VSA-based information decoding paradigm on SAE's latent representations; introducing a new feature naming methods for SAE-based analysis.
>
> In addition, while SAEs typically target a single model's hidden layer, our probing approach examines nearly the entire residual stream simultaneously. Determining the optimal hidden layer for a target task introduces a further degree of freedom in SAE-based analysis, which is avoided by our approaches though our novel ingestion algorithm (Section 4.3).
>
> We argue that the *constraints introduced by these interventions*; selecting the optimal hidden layer, assigning feature names via DLA or input clustering, and applying post-hoc feature filtering; *outweigh our need to learn a mapping to the proxy space, both conceptually and experimentally.*
>
> Exploiting the additional page, we will include some of these remarks in the main text of the revised version.
>
> ### (Q2.2) Limitation of supervised probes
> We apologize; due to space constraints, we describe the limitations of supervised probes in the main text concisely. As better described in Section 2, the evaluation of decoding capabilities of supervised probes suffers from the *challenge of separating information decoding from probe learning* (L99), the debate refereed in the introduction section (L37-38). Namely, assessing *whether the outputs of the trained probe are actual information decoded from the probed embeddings* (information decoding) or rather a result of an internal learning of the probe.
>
> *Hewitt and Liang (2019)* propose control tests, such as random baselines, to separate actual information decoding from probe learning. To address this, we validate encoders on unseen syntactic structures (Section 5.1) and perform two control tests (Table 5, Appendix F): a permuted baseline using randomly shuffled embeddings and an unrelated baseline extracting irrelevant concepts. Low precision scores (0.08–0.09) confirm that our VSA-based probe produces meaningful outputs only for relevant inputs.

---

> ### Author Response · Authors · 2025-11-20
> **Rebuttal by Authors (part 3)**
>
> Lastly, conventional supervised probes are generally tied to task-specific features, targeting directly the downstream task:
> 1. Binary classes (truthfulness; Marks et al., 2023)
> 2. World information (geographical coordinates; Gurnee et al., 2023)
> 3. Input syntax (syntactic attributes; Simon et al., 2024).
> On the other hand, our VSA-based probe *learns a vector transformation, mapping neural embeddings into VSA encodings*, rather than directly targeting a downstream task (L121) such as input-completion.
>
> In summary, while our proposed method does qualify as a supervised probe (L118), we address the ongoing challenge of separating information decoding from probe learning through several conceptual and experimental strategies, demonstrating effective information decoding via our VSA-based probe.
>
> ### (Q2.3) Invertibility and causal validation
> As logical extension, we are actually working on testing invertibility of $\mathcal{M}$; a bidirectional transformation enabling VSA-based steering (i.e., causal validation), shifting from a encoder-only model (Section 4.4) to an encoder-decoder architecture: $$\mathcal{M}: \mathbb{R}^d \to \{-1,1\}^D \to \mathbb{R}^d$$
>
> We have already interesting preliminary results in steering the language of text generation, by removing the language concept from the input embeddings, and adding another language concept before text generation. To learn LLM subspace for a language concepts, we first train an autoencoder-alike model with a combined loss function to reconstruct the LLM embeddings while internally represent the target VSA concept:
> $$e(\text{“Die Hauptstadt Frankreichs}) \mapsto \phi_{\text{german}} \mapsto e(\text{“Die Haup...”})$$
>
> The encoder achieved an average binary accuracy of 0.99 on VSA concepts, and the decoder reconstructed LLM embeddings with a cosine similarity of 0.96. To steer auto-regressive generation, we decode VSA concepts into LLM embeddings (e.g., German or Spanish), subtract the source-language concept, and add the target-language feature. The following table shows some preliminary results on multiple languages:
>
> |     | **DE** | **EN** | **ES** | **FR** | **IT** | **JA** | **RU** | **ZH** |
> |-----|--------|--------|--------|--------|--------|--------|--------|--------|
> | **DE** | 1.00 | 0.61 | 0.39 | 0.40 | 0.70 | 0.38 | 0.47 | 0.70 |
> | **EN** | 0.88 | 1.00 | 0.64 | 0.75 | 0.85 | 0.69 | 0.74 | 0.80 |
> | **ES** | 0.85 | 0.84 | 1.00 | 0.84 | 0.88 | 0.49 | 0.74 | 0.78 |
> | **FR** | 0.77 | 0.63 | 0.50 | 1.00 | 0.80 | 0.45 | 0.56 | 0.67 |
> | **IT** | 0.38 | 0.71 | 0.20 | 0.33 | 1.00 | 0.24 | 0.37 | 0.56 |
> | **JA** | 0.96 | 0.96 | 0.84 | 0.93 | 0.93 | 1.00 | 0.88 | 0.85 |
> | **RU** | 0.80 | 0.63 | 0.65 | 0.73 | 0.75 | 0.36 | 1.00 | 0.82 |
> | **ZH** | 0.59 | 0.58 | 0.21 | 0.50 | 0.69 | 0.27 | 0.34 | 1.00 |
>
> While causal steering is one way to validate probes, we instead validate our method via extracted concepts and control tests (L485) in our controlled unidirectional setting. Implementing a VSA-based steering approach would require additional methodological and experimental details. We note the unidirectional (encoder-only) transformation as a limitation of the current implementation.
>
> In summary:
> 1. Our VSA-based probe is a first step toward actively steering LLM text generation using VSA concepts.
> 2. It currently supports only unidirectional transformation; causal validation would require an encoder-decoder setup.
> 2. A full VSA-based steering methodology is in development and will be presented separately.
>
> # (Q3) Normalization before sum pooling
> We do not normalize before sum pooling, as we aim to preserve centroid magnitudes and the underlying data structure. High-magnitude layers may reflect strong conceptual signals for the model’s output, while low-magnitude layers may indicate weaker conceptual richness.
>
> Appendix O.2 analyzes LLM latent directions via the eigenvalues of \(G = HH^T\), where \(H\) is the residual stream.
> For OLMo-2 (layers 32–64, 100K inputs), a few dominant eigenvalues (3–4 modes) indicate an approximately low-rank space. The first mode (\(\lambda = 0.65\)) likely reflects next-token prediction, while remaining modes capture auxiliary information. Our hyperdimensional probe targets these auxiliary structures without compromising their magnitude or importance.
> # (Q4) Partial-correct precision score in LLM performance
> Due to space constraints, we only briefly explain LLM metrics and the rationale for the partial-correct score in input-completion tasks (Section 5.2). In the revised version, we will use the additional page to clarify this directly in the main text (see also Appendix A).
>
> We assign a 0.5 score in precision@k for partially matching tokens to handle tokenization and morphological ambiguities (e.g., accomplish vs. accomplishment). Fully scoring such predictions as correct would overestimate performance, while scoring them as incorrect would underestimate it. The intermediate score reflects genuine uncertainty in model intent.

---

> ### Author Response · Authors · 2025-11-20
> **Rebuttal by Authors (part 4)**
>
> Empirically, *ambiguous cases are relatively rare* (Appendix A), owing to the extensive vocabularies of modern LLMs: on average they occur in only 8.6\% of next-token precision@1 instances across all the six models (Llama 4: 0.7\%, Pythia: 16.2\%). Thus, while **this choice slightly lowers precision@k for LLM performance, the 0.5 weighting provides a conservative but fair way to account for token-level ambiguity inherent in our data** and in current LLM tokenization schemes.
>
> In response to your comment, we recalculated LLM performance by converting all 0.5 scores to 0 or 1. *This caused only minor changes (4–7% on average), leaving our overall findings unchanged.* The table reports metrics for the three cases:
>
> | MODEL                 | P@1 with 0.5 | P@5 with 0.5 | P@1 (0.5→0) | P@5 (0.5→0) | P@1 (0.5→1) | P@5 (0.5→1) |
> |----------------------|----------------|----------------|------------------|------------------|------------------|------------------|
> | Llama 4 Scout, 17B-16E | 0.077 | 0.463 | 0.074 | 0.425 | 0.081 | 0.501 |
> | GPT-2, medium         | 0.227 | 0.471 | 0.187 | 0.394 | 0.267 | 0.548 |
> | Pythia-1.4b           | 0.288 | 0.541 | 0.207 | 0.429 | 0.369 | 0.654 |
> | Llama 3.1-8B          | 0.309 | 0.490 | 0.266 | 0.435 | 0.352 | 0.546 |
> | Phi 4                  | 0.478 | 0.683 | 0.436 | 0.614 | 0.519 | 0.753 |
> | OLMo-2                | 0.482 | 0.656 | 0.435 | 0.591 | 0.529 | 0.721 |
> | **AVERAGE**           | 0.310 | 0.551 | 0.268 | 0.481 | 0.352 | 0.621 |
> | Delta (%)             |        |        | -4.2% | -7% | +4.2% | +7% |
>
> The question-answering scenario (Section 5.3) requires different LLM metrics, as it evaluates multi-token text generation rather than single-token predictions (Section 5.2). We report F1 and Exact Match (EM) scores, following SQuAD’s official evaluation script. F1 measures token overlap, EM checks exact matches.
>
> We will clarify these metrics in the revised 10-page version.
>
> ### Probe performance on the correct subset
> *Our work focuses on LLM failure instances, aiming to debug inference by analyzing the actual concepts encoded in the model’s vector space. *
>
> Unlike ELK, our probe inspects the full internal state—just before next-token prediction (Section 5.2) or during text generation (Section 5.3), *without assuming any link to the model’s outputs*, addressing a *representation-learning problem* rather than an alignment problem.
>
> While Figure 2 illustrates ELK-like debugging, **our main insights come from examining the entire set of encoded concepts.** Section 5.2 highlights differences in conceptual richness across LLMs and input types, and Section 5.3 analyzes QA failures, revealing concept-related patterns that indicate sources of incorrect answers.
>
> # (Q5) Interpretation of Table 1
> We apologize for the limited detail in Table 1 due to space constraints. Following reviewer feedback, we will clarify the table’s components in Section 5.2 and include relevant points from Appendix A in the revised version using the additional page.
>
> While Section 4.5 and Equation 4 describe the procedure for concept extraction using the unbinding operation, *Table 1 shows the final results of this extraction procedure.* Each row indicates the set of concepts yielded by our probe via the concept extraction procedure (Section 4.5) for all the six LLMs (columns). A concept set groups the concept used for the unbinding operation, and the concepts found after applying unbinding (see Equation 4).
> For example, the first entry "Key$|$Target'' indicates the probe extracts the set of concept representing the key and the value of the target analogy (e.g., $\{\phi_\textrm{mexico}, \phi_\textrm{peso}\}$ in Equation 4) by using the "Key'' to perform the unbinding operation ($\phi_\textrm{mexico}$) which yielded the "Target'' concept ($\phi_\textrm{peso}$). Lastly, the cell indicates the relative frequency of a concept set for a specific language model: for example, the probe extracts the key and value of the target analogy (Key$|$Target) in 85.4\% of the instances ($\bar{\mathcal{S}}$) for Llama 3.1, and in 60\% of instances for GPT-2.
>
> ### Insights on GPT-2
> Our key insights come from analyzing all extracted concepts, not just the target. GPT-2 shows the highest relative frequency of input-related concepts (e.g., “Example,” “Key”) and of no extraction (“None”) in Table 1, *indicating our probe more often retrieves input-related concepts, or none at all, from GPT-2 embeddings compared to other models.*
>
> These extraction results suggest that GPT-2’s final state tends to encode either no concept or primarily input-related concepts, rather than the concept needed to complete the analogy.
> This led us to hypothesize that GPT-2 struggles with the input-completion task (L397–399) due to insufficient understanding/knowledge for completing the inputs, *causing it to fall back on representations of input-related concepts in its internal state prior to next-token prediction.*

---

> ### Author Response · Authors · 2025-11-20
> **Rebuttal by Authors (part 5)**
>
> # (Q6) VSA encodings and concept detention
> Due to space constraints, Section 4.5 provides a concise description of how we probe the VSA encodings produced by our trained encoder, namely our concept extraction procedure. We compare the VSA encodings, both before and after each unbinding operation, with the VSA codebook ($\Phi$) using cosine similarity.
>
> A concept is considered detected when its similarity exceeds the threshold of 0.1 (L318). This threshold was chosen empirically: preliminary experiments showed that contextually relevant concepts consistently appear above this threshold, while irrelevant concepts fall below it. Although this threshold may seem low, it is important to note that VSAs and hypervector algebra are highly noise-tolerant (L147), allowing recovery of concepts even from very noisy representations.
>
> # (Q7) Concept persistency during text generation
> We refrain from making strong assumptions, such as *presuming that an LLM deletes a concept’s internal representation from its vector space immediately after generating the corresponding word*.
>
> Such assumptions can be *implicitly suggested in lens-style (DLA-based) approaches*, which focus mainly on surface-level, token-oriented latent features: next-token representations. In contrast, **our VSA-based methodology captures deeper latent features** by associating each VSA feature (i.e., concept) with a complete direction/subspace within the LLM vector space, **capturing also concepts not encoded in the surface-level embedding space.**
>
> Our experimental results in Section 5.3, particularly the *observed concept drift during text generation* (Figure 3), further demonstrate the *strength of probing LLM embeddings without imposing strong assumptions about which concepts are internally represented.*

---

> > ### Comment · Reviewer_s5VU · 2025-11-24
> >
> > I thank the authors for their very detailed response.
> >
> > Q1. Are the scores summarized here showing comparison with [Belrose et al, 2023](https://arxiv.org/abs/2303.08112) or Logit Lens? Did you compare your results with [Ghandeharioun et al, 2024](https://arxiv.org/abs/2401.06102) and  [Hernandez et al, 2024](https://arxiv.org/pdf/2308.09124)
> >
> >
> >   * I am not sure if I am understanding the response correctly, but you seem to be making the case that your approach is not directly comparable to many of the prior ELK approaches. Which I am still not sure why. Can you summarize informally what you plan to report in Section 5.4 here?
> >
> > Q2.
> >
> > 2.1. Thanks for elaborating your justification for making this claim. To be honest, I still find it a bit unconvincing, but I appreciate the effort. And, I still think your approach introduces new limitations while addressing some old ones. Which limitations are more important probably depends on the use-case and can be very subjective. So, I will leave it at that.
> >
> > 2.2. Thanks for the clarification. I suggest you further elaborate this in the main text for better clarity.
> >
> > 2.3. As reviewer `gn8b` also raised this concern: I aggree with them that without causal validation, it is hard to claim that the extracted concepts are actually used by the LM in its computation. And, I have not really seen the validation strategy you explored in Appendix F.1 in prior work before. So, I am not sure how much I should trust this validation.
> >
> > But, your encoder-decoder approach seems promising. I encourage you to develop it further and incorporate it in a single paper. I am still a bit concerned as shallow MLPs (meta-model) are being learned and how much should we trust them to reflect the actual LM computation. But I can't complain if this approach generalize well and can causally influence the base-LM across ood settings.
> >
> >
> > Q3. *"High-magnitude layers may reflect strong conceptual signals for the model’s output"* - I don't buy this argument. Aren't concepts specific directions in the latent space? Their "strength" should be measured with respect to the "strength" of other concepts, i.e., the average magnitude of particular layers should be taken into account, in my opinion. But, thanks for the response anyway.
> >
> > I have read Appendix O.2. But I am not sure how this is relevant to my question here.
> >
> > Q4. Thanks for reporting the recalculated results. I appreciate it.
> >
> > 4.1. I don't think your response answers my question. If the LM fails to generate the correct target answer, doesn't it mean that the concept is either not present or not stroung enough to influence the LM's output?
> >
> >
> > Q5. Thanks for your clarifications. I still don't fully understand Table 1. I believe you can use the additional page during the rebuttal period (check https://iclr.cc/Conferences/2026/AuthorGuide). Instead of promising to add stuff in the final version, you can add them now to help reviewers better understand your work.
> >
> >
> > Q6. Thanks
> >
> > Q7. Thanks for your response. I also asked some how many tokens were generated after the input concepts and clarification on the example. Can you please address these?

---

> ### Author Response · Authors · 2025-11-24
> **Response to Reviewer's comment (part 1)**
>
> # Q2.1: ELK, and our underlying objectives
> We apologize for the lack of clarity in the current manuscript. As reviewer dtKf noted, the paper does not yet present a *sufficiently explicit problem statement* or a *clear explanation of the constraints of prior methods that motivate our novel method*. Due to space constraints, these aspects were only briefly addressed in the main text.
>
> As a result, readers may have been left uncertain about our core objectives, such as the dichotomy with ELK, the specific gaps in existing approaches that we aim to address, and the reasoning behind our methodological choices.
>
> We are currently integrating these clarifications into the revised manuscript, scheduled for *upload before the end of the rebuttal period* (3/12); by *substantially rewriting the introduction*; and *better framing existing approaches in the related work section.*
>
> ## Problem Statement
> Our primary research question is to **understand which concepts were represented internally during LLM reasoning**; specifically, *which concepts are encoded in the internal state of a language model* before completing analogy-style inputs and during text generation in a QA-style setting. Probing an LLM's final state to answer the following research questions related to LLM reasoning:
> - **RQ1**: Does the LLM represents all in-context concepts when solving an analogy? (Section 5.2)
> - **RQ2**: Does the LLM rely on key–value associations to solve an analogy? (Section 5.2)
> - **RQ3**: Does the LLM represent only the output concept? (Section 5.2)
> - **RQ4**: Are there any identifiable patterns across different input types? (Section 5.2)
> - **RQ5**: Are there meaningful differences in behavior across different LLMs? (Section 5.2)
> - **RQ6**: What concepts are present prior to text generation, and which ones emerge afterward? How many question-related and answer-related concepts appear? (Section 5.3)
>
> *Our task is related to general-purpose interpretability of LLM embeddings* rather than their misalignment with model's output (ELK); **focusing on which concepts are encoded rather than sorely assessing whether the output-related concept is present.**
> Specifically, we address the representation-learning problem, aiming at identifying latent features (concepts) presented in LLM embeddings with an information decoding technique.
> *The misalignment between these latent features and the model's outputs, is a related, yet subsequent problem*, addressed by Eliciting Latent Knowledge (ELK).
> Information decoding extracts human-interpretable symbols/features from neural embeddings, while ELK exploits these extracted features to reveal misalignment between model's state and its output.
> Information decoding is one possible approach to achieve ELK by narrowing the scope of information decoding on output-related latent features.
>
> **Because existing approaches to neural-embedding interpretation impose significant limitations for addressing our research questions, we introduce a new methodology.**
> Our approach leverages core properties of Vector Symbolic Architectures (VSAs) and hypervector algebra to *overcome several conceptual and experimental limitations of prior methods that hinder our ability to address the research questions*.
>
> Specifically, our new methodology addresses the following constraints of existing methods:
> ### (1) Direct Logit Attribution (DLA)
> The primary limitation of Direct Logit Attribution, such as the lens-style methods of Nostalgebraist (2020) and Belrose et al. (2023), in addressing our research questions, is its reliance on token representations, surface-level latent features.
> Their dependence on the model’s output vocabulary undermines our research goals in two key aspects:
> 1. it **restricts *concept representation* to single-token words**, and
> 2. by *directly pursuing the ELK-style* misalignment problem, it **interferes with our broader objective of *general-purpose* concept extraction**.
>     - Nostalgebraist’s *Logit Lens*, Belrose et al.’s *Tuned Lens*, and Ghandeharioun et al.’s *Patchscopes* all tackle this ELK-style problem by examining the model’s output distribution as projected from its internal vector space; using, respectively, the *unembedding matrix*, a *learned affine transformation*, and *patched inference of an LLM*.
>
> ### (2) Sparse Autoencoders (SAEs)
> On the other hand, SAEs provide general-purpose concept extraction using sparse dictionary learning, and a learned proxy space. The shortcomings of SAEs that limit their usefulness for our research questions are threefold:
> 1. **Unbounded and non-compositional *feature space***: SAEs induce an effectively unbounded and hard-to-control set of latent features, making it difficult to ensure coverage, comparability, or principled organization of the feature space.

---

> ### Author Response · Authors · 2025-11-24
> **Response to Reviewer's comment (part 2)**
>
> 2. **Constrained or unstable *feature naming***: SAE feature naming typically relies either on DLA, carrying forward its dependence on surface-level token associations and their limitations, or on input-driven clustering methods, which are highly task- and dataset-dependent. As a result, the semantic grounding of features remains inconsistent and fragile.
> 3. ***Layer-localized* representations:** SAEs operate on a single hidden layer, requiring researchers to choose a supposedly “optimal” layer. This choice is inherently task-specific and undermines our goal of analyzing the model’s internal state holistically across layers.
> ### (3) Conventional supervised probes
> While our approach meets the criteria of a supervised probe, we introduce the term “hybrid supervised probe" (L118) to describe a **method that combines the *mapping framework of conventional supervised probes* with the *SAE’s ability to disentangle superposed subspaces* through a sparsity constraint**.
>
> Our novel supervised mapping paradigm, with the sparsity constraint implemented via VSA encodings, allows us to *combine SAE-alike capabilities with interpretability of conventional probes*:
> - learning a **sparse proxy space** (SAEs) rather than directly the downstream task (probes)
> - defining a **bounded feature space** (probes) rather than unbounded (SAEs)
> - querying a **learned proxy space for feature extraction** (SAEs) rather than conventional classification (probes).
> ### Hyperdimensional probe
> As a result, we design a novel approach:
> |Characteristic|Component|Limitation of|
> |------------------------------------------------------------------------|------------------------|------------------|
> |representing unique, well-defined abstract concepts |VSA codebook|DLA|
> |inspecting the model's internal state without introducing layer selection|Ingestion algorithm|SAEs|
> |learning a sparse proxy space with a bounded feature space|neural VSA encoder|SAEs/probes|
> |querying LLM embeddings with a compositional feature space|hypervector algebra|SAEs|
> |recovering concept from neural encoding for general-purpose probing|VSA-based probing|DLA|
>
> While our VSA-based paradigm advances information decoding in LLM embeddings, it may also be applied to SAE feature naming, thereby enhancing the interpretability of SAE-based analysis.
> ### Research questions
> Our work addresses the six experimental questions outlined above (RQ1-RQ6) with two additional methodological questions:
> - **RQ7**: Can LLM embeddings be faithfully represented using VSA encodings? (Section 4.4)
> - **RQ8**: To what extent can VSA-based probing recover information from the LLM vector space? (Section 5.2 and Section 5.3)
> # Q1: DLA-based results and experimental comparison section
> **Our experimental comparison** (DLA-based results; Appendix G.1), and **the scores provided in our previous response, can be referred to Logit Lens by *Nostalgebraist* (2020).**
> While plenty of scholars introduces advantages in output distribution investigation, such as more reliable layer-wise next-token projections via Tuned Lens (Belrose et al., 2023), and Patchscopes (Ghandeharioun et al., 2024), *they share
> many conceptual and experimental limitations of DLA-based methods and output distribution investigation*.
>
> While we contrast our VSA-based results with those yielded by DLA/Logit Lens, **our aim is to underscore the main *constraints of output-distribution-oriented analyses* for our general-purpose probing objective.**
> Their sorely focus on output distribution investigation, caused by their reliance on surface-level, token-level latent features:
> 1. **limits their ability to yield deeper insights** into our experimental research questions, such as our objective to *consider other in-context concepts* rather than only the output-related concept, and
> 2. **introduces both *exogenous* and *endogenous* variability** typical of *token-oriented methods*.
>
> On the other hand, as previously noted, Linear Relational Embedding (LRE) by ***Hernandez et al.* (2024) is not an appropriate comparison**, since it operates in a *different, though complementary, experimental setting.*
> The authors focus on *learning relations between two specific input token embeddings*, rather than analyzing the model’s internal state prior to next-token prediction (Section 5.2) or during text generation (Section 5.3).
> # Description of the new experimental comparison section (5.4)
> Subjected to the 10-page limit, we plan to integrate the following aspects in the experimental comparison section:
> 1. Explanation of our *choice for a DLA-based experimental comparison*, integrating some remarks presented here and in Appendix G;
> 2. Describing our *DLA-based experimental setting* reported in Appendix G.1;
> 3.  Reporting *non-trivial insights into DLA-based results*, now reported in Appendix G.1;
> 4. Moving *Table 7 from Appendix G.1* to this new section in the main text;
> 5. Integrating the *DLA-based precision@1 scores into Figure 2*.

---

> ### Author Response · Authors · 2025-11-24
> **Response to Reviewer's comment (part 3)**
>
> Initially, we *explicit connect our DLA-based methodology with the Logit Lens method* proposed by *Nostalgebraist* (2020), while motivating our experimental comparison choices.
> We will also describe its general-purpose objective, aimed to *highlight limitations of DLA-based and output-distribution-oriented analyses for our objective of a general-purpose concept extraction.*
>
> After describing the DLA-based experimental setting (L1004 - L1006), we will report *non-trivial insights into DLA-based results* (L1006 - L1022). Reporting differences in extraction-related results and a concrete example.
> For instance (L1010-1012), for the analogy *king is to queen as son is to* $\mapsto$ *daughter*, using OLMo-2, our probe extracts the key-target concepts (*son* and *daughter*), while DLA produces no concepts.
> The model predicts the next token prediction as *?* with a softmax score of 0.06, followed by *father* (0.05); the target word has a rank of 37 in the sorted softmax scores.
>
> We argue that DLA's *focus on token-level and other surface-level features* limits their effectiveness for *comprehensive concept-oriented interpretability* of the LLM vector space.
> **This positions *output-distribution–focused approaches*, such as lens-style methods, somewhere between naive LLM's next-token inspection and deeper, more general-purpose information decoding methods** such as VSA probing.
>
> We will also *relocate Table 7 from Appendix G.1 to the main text*, showing results for concept extraction using DLA-based outputs. Lastly, we will integrate the DLA-based precision@1 scores presented in our previous response into *Figure 2*, enabling a comprehensive visual comparison between LLM performance, and DLA-based and VSA-based probing.
>
> # Q2.3: Causal validation
> **Our objective is *not* to claim that the extracted concepts are actually used by the language model to produce its outputs**, as already clarified in our research questions and problem statement. Instead, our aim is to investigate the model’s internal state holistically, without making assumptions about whether the LLM actively exploits these concepts during computation or how they relate to its outputs.
>
> Moreover, analyzing active usage would require a precise definition of “usage”.
> For example: Is a concept considered “used” when its representation aligns with the model’s output distribution? Does misalignment imply non-usage? And how could one rigorously map abstract concepts onto next-token predictions or natural-language generation?
>
> While, our focus is on *passive representativeness* rather than *active usage*: identifying which concepts are represented in LLM embeddings, not determining whether or how the model operationalizes them.
> **Our validation strategies are therefore designed to *assess the representativeness of the extracted concepts* within the LLM embeddings.**
> Given the constraint of a unidirectional transformation, which we will explicitly list as a limitation, we employ multiple approaches to validate information decoding.
> Among those previously discussed in our previous response (limitations of supervised probes), **the most effective strategy is to evaluate our method on *textual inputs different* from those used during probe training.** Concretely, we use:
> - $\bar{\mathcal{S}}$ (instead of $\mathcal{S}$) for the analogy-style setting in Section 5.2, and
> - $\bar{\mathcal{Q}}$ (instead of $\mathcal{Q}$) for the QA-style setting in Section 5.3.
>
> Although $\mathcal{S}$ and $\bar{\mathcal{S}}$ share the same underlying semantic concepts, *they contain distinct textual inputs with different syntactic structures*. Similarly, in the QA-style setting (Section 5.3), we further validate our concept extraction by *training the probe on SQuAD questions* (Equation 5; $\mathcal{Q}$) and *testing it on different inputs that additionally include the corresponding contextual text* (Equation 6 in Appendix I; $\bar{\mathcal{Q}}$).
> This makes them effectively out-of-domain relative to the embeddings on which the probe was trained.
>
> # Q3: Normalization before sum pooling
> The reference to Appendix O.2 was meant to connect to your observation about higher magnitudes in later layers.
> In that analysis, we suggest that the ***dominant direction present in the LLM vector space* may correspond to next-token–prediction representations, which are typically encoded in the later layers of an autoregressive model.**
>
> Although these later layers may capture the most predominant concept, our aim is to study the model’s internal state as holistically as possible. Consequently, we deliberately *minimize assumptions on LLM embeddings, even at the risk of inheriting their bias toward this dominant direction or concept.*

---

> ### Author Response · Authors · 2025-11-24
> **Response to Reviewer's comment (part 4)**
>
> Regarding your prospective, considering *concept “strength” relative to the strength of other concepts* is an *interesting* and *valid perspective*.
> It would indeed be *valuable to test our VSA-based methodology with such a normalization step* to mitigate potential biases toward later layers and to explore how this adjustment relates to concept structure in VSA encodings.
>
> # Q4.1: LLM's outputs and embedded concepts
> While we agree that LLM *failures may stem from weak or absent output-oriented concept representations* in their latent space; an LLM’s responses are also *influenced by numerous exogenous and endogenous factors*, ranging from prompt design to tokenization effects and softmax-level uncertainty in next-token prediction.
> **Even if a concept is internally represented (to varying degrees of strength), the model may *still fail to fully leverage it* because of these influencing factors.** Prompt engineering improves querying by crafting instructions that elicit relevant knowledge, while fine-tuning addresses the same elicitation problem by training the model to focus on task-specific information.
>
> In contrast, our approach takes a *broader view*: we seek to identify the concepts encoded in the model’s internal state (see problem statement) **without assuming the model actively uses them during computation**.
> It may still be useful to *cluster instances by model correctness*, similar to the analysis in Section 5.3.
>
> # Q5: Clarification on the revised version
> We are currently integrating all reviewer-requested clarifications into the revised 10-page manuscript, **which we will upload before the end of the rebuttal period** (3/12).
> Your comments, together with those from the other reviewers, meaningfully strengthen our work and help ensure that the *final decision can be made based on a camera-ready–like version rather than on authors’ promises*.
>
> In parallel, we genuinely appreciate the opportunity to *elaborate on several key aspects* during this rebuttal that *cannot be fully addressed within the space constraints* of a 10-page main text.
>
> # Q7: Generated tokens and example in the QA setting
> In the natural-language generation setting described in Section 5.3, **we generate up to 20 tokens** (with *max\_new\_tokens* set to 20). We apply our trained VSA encoder to the model’s internal state **immediately before the first token is generated and again just before the final token**, which we refer to as the *before* and *after* text-generation moments, respectively.
>
> To prevent unrelated continuations, we **implemented a *stopping criterion* for autoregressive generation** (using *StoppingCriteria* from the Transformers library) that halts autoregressive generation when a *full stop* or a *newline* is produced.
> As a result, the **actual number of generated tokens is often below 20**; however, we did not record this quantity.
>
> Prompted by your comment, we however performed a *descriptive analysis of the number of words in the model’s generated texts*, although we note that words and tokens do not necessarily correspond:
> | **Statistics** | **Words** |
> |----------------|----------:|
> |**MEAN** | 4.6 |
> |**STD** | 4.7 |
> |**MIN** | 1.0 |
> |**25%** | 1.0 |
> |**50%** | 3.0 |
> |**75%** | 6.0 |
> |**MAX**|20.0|
>
> Regarding your *example-related question*: we observe that the concepts *try* and *hydrogen* are present in the model’s internal state prior to generating the first token (“water”), but absent before generating the final token (“heat”).
> In other words, VSA probing indicates that during text generation the LLM vector space appears to lose (delete? obscure?) two of the three question-related concepts, while the answer-related concepts remain consistent and even become more complete.
>
> The model ultimately produces the text “*water and heat*,” which aligns with the concepts extracted by our probe after text generation (i.e., before producing the last token): *produce*, *solar*, *water*, and *energy*.
> These concepts reflect a generic notion of energy production rather than anything specific to hydrogen production, ultimately leading the LLM to produce an erroneous answer.
> This, along with the patters emerged in the concept relevance analysis, leads us to hypothesize that the LLM might have lost focus on the question-relevant concepts after generating the first token, **resulting in a answer that *is no longer about hydrogen* but instead reflects the *remaining, more general concepts* still represented/active in its internal state.**
>
> # New sections and paper organization
> To improve the *logical flow of the manuscript*, we plan to relocate the content of Section 5.3 to a new Section 6, moving the validation of our full pipeline (Section 4) within the QA setting (formerly Section 5.3) into *its own dedicated section*.
>
> As a result, in the revised version, *our experimental comparison will appear in the new Section 5.3, rather than the previously referenced Section 5.4.*

---

### Official Review · Reviewer_gn8b · 2025-10-31

**Soundness:** 3
**Presentation:** 2
**Contribution:** 2
**Rating:** 4
**Confidence:** 3

**Summary:**

This paper proposes a new method to extract and interpret information contained in LLM hidden states called hyperdimensional probing. The method is built on vector-symbolic architectures (VSAs). They apply their method to analogy tasks, a question answering task, and other various tasks in the appendix to show the flexibility of their method in extracting knowledge from llm hidden states.

**Strengths:**

- The paper applies the VSA probe methodology to a wide variety of tasks of varying complexity, as well as across various model sizes and families which is helpful to show the broadness of applicability.
- The experimental details in the appendix are extensive, and several times I was able to answer questions I had while reading through some of the extra content there. (Though this sometimes felt like a weakness as well - see below).
- While I may be misunderstanding some pieces (due to lack of comparison/explanation), the paper does feel overall well-put-together. The use of VSA for probing seems novel to me.

**Weaknesses:**

- The paper was somewhat hard to follow at times and I found myself having to read things back several times regarding the VSA training details, architecture, dataset, and evaluation procedure. One challenge with reading is that because there is so much packed into this paper, constant referrals to the appendix from the main text sometimes felt overwhelming. Sometimes the question I had was quickly answered, and other times I had to read a few pages in the appendix to understand a single result. It seems like there is something in the big picture here that I'm missing because the results/paper feels well-put together, but I just didn't quite get why this VSA approach is better/how it compares to other existing methods. The main reasons I may misunderstand your method I'll describe below, but for now they seem like limitations of the current presentation in the paper.

- One downside of probing classifiers in general is that, while they are often used to “detect” whether an attribute is present in a hidden state, they are often correlational and not causal (see [Belinkov et al](https://aclanthology.org/2022.cl-1.7/) for an overview, but also related work). To address this, more recent work has begun addressing whether probes can be used to causally steer generation, which is something I didn’t see attempted in this work. I worry that the hyperdimensional probing suffers from similar problems as traditional probes, and it’s unclear to me whether the VSA probes could “steer” LLM generation, or are they only able to “extract” information from them.

- Similar analogy tasks tested here have been examined in prior work, but there is little comparison with previous methods. There is an attempt at comparison with logit lens (DLA), but there are more sophisticated methods that could be helpful points of comparison for the proposed VSA method. For example, the LRE from [Hernandez et al](https://openreview.net/forum?id=w7LU2s14kE) seems to extract entities for various relations. Similarly, the AxBench paper [Wu et al](https://openreview.net/forum?id=K2CckZjNy0) provides a benchmark for both concept detection and model steering for various concepts that could be formatted as analogies. Evaluating on this benchmark might be helpful to see how VSA probing compares to other standard probing techniques, SAEs, DAS, prompting, etc.

**Questions:**

- Line 53: what is meant by “hybrid supervised probe”? This is mentioned a few times, but not really explained.
- Several times you mention that VSA probing is an attempt to overcome some of the limitations of other information extraction methods (DLA -> vocabulary/token space, SAEs -> feature naming, etc.), but it’s not clear that these VSA features are any more interpretable/better than alternatives, particularly because it seems like you also tie your features to predefined tokens/concepts. How is this any different?
- If you were to train the VSA codebook for concept/task-steering, how would it differ from [Shao et al](https://openreview.net/forum?id=6axIMJA7ME3)’s compositional task representation codebook approach? Their method seems very related (though obviously not VSA based), as they also train a codebook of features for various LLM tasks.
- What do you expect to get out of doing unbinding with other candidates that aren't the final key/target? I didn't quite understand the breakdown, for example, of the rows in Table 1 and what one might expect to retrieve in each row. Do you similarly evaluate probes/DLA at all token positions for that same information?
___
- Other Notes:
  - Line 160 presents section 5.3 in between 4.1 and 4.2 which was a little odd. I would consider changing the order of these sentences.
  - Line 292-293: “this supports our hypothesis that LLM embeddings can be represented using fully distributed encodings such as MAP-B in VSAs“ - this feels like somewhat of an over-claim because this statement seems like it's only based on the findings from analogy dataset. I would suggest tapering this claim a bit (e.g. If LLMs hidden states can be represented w/ VSAs why are we using transformers and not VSAs).

- Minor Typos
  - Line 80: “token mbedding” -> token embedding
  - Line 275: “valide” -> validate?
  - Line 1148: “attak” -> attack?
  - Figure 2: Label says "GTP-2"

---

> ### Author Response · Authors · 2025-11-20
> **Rebuttal by Authors (part 1)**
>
> Thanks for recognizing:
> 1. the *novelty of using VSAs for probing*;
> 2.  *its flexibility* and *broad applicability* demonstrated across tasks of varying complexity, model sizes (from 55M to 109B parameter), and (six) families;
> 3. the *thorough experimental details provided in the appendix*.
>
> We also truly appreciate you catching small typos, showing how carefully you reviewed our paper.
>
> ---
> # Advantages of VSA-based approach and the "hybrid'' term
> One of the key advantages of our approach compared to existing probing paradigms is to support a set of concepts, for probing LLM latent features, with unrestricted levels of abstraction, cardinality, and data types via VSAs and hypervector algebra (L124).
>
> This enables targeting potentially any latent features inside LLM embeddings. Our approach avoids the reliance on the model's output vocabulary of DLA-based analysis, such as DLA-based probing or SAE's feature naming via DLA (L112). VSA-based probing also decodes multiple concepts from LLM vector space, overcoming the sorely reliance on next-token representations of DLA (L106).
> On the other hand, our decoding paradigm avoids the posterior feature-naming stage of SAEs (L121) by defining a domain-relevant concept set beforehand (codebook $\Phi$), enabling a top-down probing strategy rather than the bottom-up paradigm of SAE-based analysis.
>
> We introduce the term *hybrid supervised probe* to denote an approach that combines the mapping paradigm of conventional supervised probes with the SAE’s capacity to disentangle superposed subspaces achieved via a sparsity constraint (L118; represented through VSA encodings in our case).
>
> We will exploit the additional page of the revised version to better frame our VSA-based methodology in relation to exiting probing methods.
> # Applicability for steering text generation
> Steering is a fascinating use case. As logical extension, we are actually working on testing a bidirectional transformation to enable VSA-based steering, shifting from an encoder-only model (Section 4.4) to an encoder-decoder architecture:
> $$\mathcal{M}: \mathbb{R}^d \to \{-1,1\}^D \to \mathbb{R}^d$$
>
> We have already interesting preliminary results in steering the language of text generation, by removing the language concept from the input embeddings, and adding another language concept before text generation. To learn LLM subspace for a language concepts, we first train an autoencoder-alike model with a combined loss function to reconstruct the LLM embeddings while internally represent the target VSA concept:
> $$e(\text{“Die Hauptstadt Frankreichs}) \mapsto \phi_{\text{german}} \mapsto e(\text{“Die Haup...”})$$
>
> The encoder achieved an average binary accuracy of 0.99 on VSA concepts, and the decoder reconstructed LLM embeddings with a cosine similarity of 0.96. To steer auto-regressive generation, we decode VSA concepts into LLM embeddings (e.g., German or Spanish), subtract the source-language concept, and add the target-language feature. The following table shows some preliminary results on multiple languages:
>
> |     | **DE** | **EN** | **ES** | **FR** | **IT** | **JA** | **RU** | **ZH** |
> |-----|--------|--------|--------|--------|--------|--------|--------|--------|
> | **DE** | 1.00 | 0.61 | 0.39 | 0.40 | 0.70 | 0.38 | 0.47 | 0.70 |
> | **EN** | 0.88 | 1.00 | 0.64 | 0.75 | 0.85 | 0.69 | 0.74 | 0.80 |
> | **ES** | 0.85 | 0.84 | 1.00 | 0.84 | 0.88 | 0.49 | 0.74 | 0.78 |
> | **FR** | 0.77 | 0.63 | 0.50 | 1.00 | 0.80 | 0.45 | 0.56 | 0.67 |
> | **IT** | 0.38 | 0.71 | 0.20 | 0.33 | 1.00 | 0.24 | 0.37 | 0.56 |
> | **JA** | 0.96 | 0.96 | 0.84 | 0.93 | 0.93 | 1.00 | 0.88 | 0.85 |
> | **RU** | 0.80 | 0.63 | 0.65 | 0.73 | 0.75 | 0.36 | 1.00 | 0.82 |
> | **ZH** | 0.59 | 0.58 | 0.21 | 0.50 | 0.69 | 0.27 | 0.34 | 1.00 |
>
> While causal steering is one possible strategy for validating a probing approach, given the unidirectional transformation and our controlled experimental setting, we validate our method directly using the extracted concepts and control tests on concept extraction (L485). Incorporating a steering-based approach grounded in VSAs would have required additional space to describe several methodological changes (e.g., encoder-decoder architecture, and steering procedure) and its corresponding experimental setup and results. We will however include the unidirectional transformation, from neural embeddings to VSA concepts, as one of the limitations of the current implementation of our approach (encoder-only architecture).
>
> In summary:
> 1. our vsa-based probing paradigm can be seen as the first building block to actively steering the LLM's text generation using VSA concepts;
> 2. integrating a steering-based use case would have further reduced the overall readability of the paper, requiring removing existing contents from the main text;
> 3. we are actively working on a VSA-based steering methodology to be presented in a separate research work.

---

> ### Author Response · Authors · 2025-11-20
> **Rebuttal by Authors (part 2)**
>
> # Comparison with LRE, AxBench, and DAS
> These approaches adopt different, yet complementary perspectives for studying LLM embeddings.
>
> **Linear Relational Embedding (LRE)** proposed by Hernandez et. al. (2024), learns a linear transformation between the embeddings of two specific input tokens. Specifically, LRE learns an affine transformation that maps the embedding of a subject entity, such as $e(\text{"Miles Davis''})$, to the embedding of a related object, such as $e(\text{"trumpet''})$, learning the vector representation (LRE) of the relation "playing the instrument''.
> In contrast, our proposed method focuses solely on the final-token embedding to study the model’s complete internal state and capture all encoded concepts prior to text generation (L222).
>
> On the other hand, **AxBench** by Wu et. al. (2025) proposes a benchmark for steering methods based a variety of approaches from prompting and finetuning to SAE-based steering. As conventional steering-oriented pipelines, AxBench measures the effectiveness of manipulating LLM embeddings along two dimensions: the performance in injecting concepts (concept detection), and the steerability of model's text generation (model steering). Our approach, conversely, studies LLM embeddings without any active intervention, focusing on representation decoding rather than representation steering.
>
> Lastly, **Distributed Alignment Search (DAS)**, proposed by Geiger et. al. (2024) focus on mechanistic interpretability by linking representation analysis and mechanistic understanding. DAS localizes conceptual representations within the LLM’s vector space by identifying linear combinations of neuron activations that align with predefined semantic directions. It reveals where and how specific concepts are embedded within the model’s internal activations, whereas our approach focuses on which concepts are encoded.
>
> As further detailed in Appendix G, we compare our VSA-based results with DLA, which omits additional steps such as feature naming/filtering of SAEs, to report a direct comparison (L947).
>
> The following table summarises the key conceptual and experimental differences with the mentioned methods:
> | **Method** | **Authors** | **Objective** | **Domain** |
> |-----------|-------------|---------------|------------|
> | LRE | Hernandez et al. (2024) | Learn relational mappings between token embeddings | Relation Learning |
> | DAS | Geiger et al. (2024) | Localize concept subspaces | Mechanistic Interpretability |
> | AxBench | Wu et al. (2025) | Benchmark steering methods | Representation Steering |
> | **Hyperdimensional probe** | **OURS (ICLR 2026)** | **Decode conceptual content from internal state** | **Representation Decoding** |
> # (Q2) Advantages of VSA features
> While our approach needs a predefined set of concepts, VSA features (codebook $\Phi$) can be seen as a vocabulary with no practical constraints neither on its creation criterion nor on the cardinality, type, and source of its symbols (L123). VSA-based probing can target LLM's latent features defined by any chosen criteria, such as contextually relevant concepts (L197), or potentially even multi-modal concepts (L477). **In contrast, traditional probing methods are typically more constrained in how they define latent features to be targeted.**
>
> For example, in *DLA-based approaches* the features are tied to the **model’s output vocabulary**. *Conventional supervised probes*, on the other hand, are **tied to task-specific features**, such as binary classes (Marks et al., 2023), geographical coordinates (Gurnee et al., 2023), or syntactic attributes (Simon et al., 2024). Our decoding paradigm can target latent features defined by any chosen criteria, avoiding the reliance on the model’s output vocabulary, an advantage over DLA; and task-specific constraints of conventional supervised probes.
>
> Moreover, it avoids the problematic post hoc feature-naming and feature-filtering stage of **SAE analysis by using a top-down decoding approach.** SAEs instead follow a bottom-up pipeline that names all activated features only after training, producing an unbounded set of latent features with no built-in relevance filtering. As a result, *SAE-based analyses can become setting-dependent*, since identifying the relevant features requires an exhaustive naming step (e.g., via DLA or clustering) followed by manual filtering.
>
> While these aspects are currently discussed in Appendix G, we will exploit the additional page of the revised version to include some remarks within the main text.
>
> The following table summarises the advantages of interpreting the latent features from LLM vector space using VSA features:
>
> | **Advantage of VSA features** | **Limitation of** |
> |-------------------------------|--------------------|
> | No constraints on feature vocabulary | DLA-based techniques |
> | Target non–task-tied latent features | Conventional probes |
> | Avoid naming + filtering stages | SAE-based analysis |

---

> ### Author Response · Authors · 2025-11-20
> **Rebuttal by Authors (part 3)**
>
> # (Q3) VSA codebook for steering, contrasted with compositional task-representation codebook
> *Shao et al. (2023)* employ a learned task feature codebook to improve task performance, comparing against prompt-based methods like zero-/few-shot learning, genetic prompt search, and model tuning. In contrast, our approach focuses on decoding semantics from the model’s internal representations (information decoding) rather than enhancing downstream performance (representation steering).
>
> VSA codebooks define a fixed set of features to construct compositional input representations using predefined vector operations (Equations 2 and 5). *Similarly, our codebook consists of a fixed vocabulary of hypervectors, combined via binding and bundling to form input representations without any learning.* An encoder (Section 4.4) then maps these compositional inputs to LLM subspaces, such as the embedding directions for concepts like “mexico” and “napoleon.”
>
> The following table summarizes the key differences between our proposed approach and CTR by Shao et al. (2023):
> | **Approach** | **Objective** | **Codebook** | **Usage** | **Learning stage** | **Inference** |
> |--------------|--------------|--------------|-----------|---------------------|----------------|
> | **VSA** | Passive decoding | Fixed, predefined vocabulary | Build input representations via hypervector algebra | Learn mapping from representations to LLM subspaces | Extract concepts using proxy VSA encodings |
> | **CTR** | Active performance enhancement | Learned from data | Represent compositional tasks | Multi-task training | Search compositions for task performance |
>
> Since our work validates empirically that latent features of LLM embeddings can be represented with VSAs and hypervector algebra (L464; VSA encodings), *our VSA-based decoding paradigm could be adapted to task-oriented steering*. **To address a task-oriented steering setting, we would create our VSA codebook based on the considered NLP tasks**, such as question-answering $\mapsto \phi_{QA} $, and sentence completion $ \mapsto \phi_{sentCompl}$.
>
> *We would then build VSA-based input representations (Section 4.2) exploiting hypervector algebra*, for example given an input sentence concerning question-answering via input competition its compositional representation would be: $ (\phi_{QA} \odot \phi_{sentCompl}) $. Lastly, our encoder would learn to map this compositional representation to the corresponding LLM subspaces: directions in the LLM vector space representing the two task-oriented concepts.
>
> # (Q4) Result of unbinding with other candidates
> The "unrelated baseline'' reported in Table 5 (Appendix F.1; Validation strategy) attempts to perform unbinding with concept candidates unrelated to the inputs (thus, neither the key or target concept).
>
> The experimental results (precision@1 of 0.08) shows that unbinding concepts from VSA encodings (i.e., VSA-based probing) using concepts irrelevant to input texts yields meaningless outputs (L923). This implies that our probing approach produces meaningless results when attempting to extract concepts that are unlikely to be encoded in the probed LLM embeddings, an expected and desirable outcome.
>
> *Section 4.5 and Equation 4 describe concept extraction via the unbinding operation, while Table 1 presents the final results.* Each row lists the concepts extracted by our probe (Section 4.5) for all six LLMs (columns). **A concept set pairs the concept used for unbinding with the concepts recovered through the operation (Equation 4).**
>
> For example, the first entry "Key$|$Target'' indicates the probe extracts the set of concept representing the key and the value of the target analogy (e.g., $\{\phi_\textrm{mexico}, \phi_\textrm{peso}\}$ in Equation 4) by using the "Key'' to perform the unbinding operation ($\phi_\textrm{mexico}$) which yielded the "Target'' concept ($\phi_\textrm{peso}$).
> Lastly, the cell indicates the relative frequency of a concept set for a specific language model: for example, the probe extract the key and value of the target analogy (Key$|$Target) in 85.4\% of the instances ($\bar{\mathcal{S}}$) for Llama 3.1.
>
> For clarity, we presented the full concept sets without separating those used for unbinding from the resulting concepts. This distinction can be inferred from Table 4 in Appendix E. In the revised version, we will clarify this and improve the interpretability of Table 1.
>
> ### DLA-based results
> Concerning the comparison with DLA, we consider *the same LLM vector space as in the VSA-based probing approach*: by examining the projected next-token predictions from the model’s middle to last layers for the final token (L1005; also Figure 5 in Appendix G.3).
>
> To build *concept sets extracted through DLA*, we then apply a simple *fuzzy token-to-concept matching method* on the DLA predictions using our concept set (e.g., the token “pes” maps to the concept “peso”).
>
> Further details are provided in Appendix G.1.

---

> > ### Comment · Reviewer_gn8b · 2025-11-26
> >
> > Thank you for the detailed reply and clarifications, and comparison to related work. After reading others' reviews and replies, it seems many had similar concerns and misunderstandings.  As far as I can tell, the proposed changes/clarifications have not been made to the manuscript, and as ICLR allows you to update the manuscript during the rebuttal process, it would be helpful to see what changes are integrated before making further judgments as to whether to update my score. The current paper, as it stands is pretty dense and hard to parse at times, and lacks comparison to other similar (common) methods. However, I do appreciate the clear explanations and answers to my questions.
> >
> > > "we are actively working on a VSA-based steering methodology to be presented in a separate research work ... We will however include the unidirectional transformation, from neural embeddings to VSA concepts, as one of the limitations of the current implementation of our approach (encoder-only architecture)."
> > - The encoder-decoder architecture sounds interesting and I will be curious to see where it leads as a validation for the class of methods in general, and understand it should probably be presented on its own. I appreciate the noting of limitations of the current work for now in this regard (as there is already a lot in the current paper.)
> >
> > > **Comparing to other methods via AxBench:** "As conventional steering-oriented pipelines, AxBench measures the effectiveness of manipulating LLM embeddings along two dimensions: the performance in injecting concepts (concept detection), and the steerability of model's text generation (model steering). Our approach, conversely, studies LLM embeddings without any active intervention, focusing on representation decoding rather than representation steering."
> >
> > - If I understand correctly, AxBench's "concept detection" sub-benchmark does not require intervention, it is instead a binary classification problem where a "parametrized function" is given a model representation (e.g. hidden state), and asked to produce a label of whether a concept is  "present" or not.  As this is a typical use case (logistic regression) for traditional probing, it feels like an appropriate benchmark that VSA probes could also be evaluated on.

---

> ### Author Response · Authors · 2025-11-26
>
> # Comparison with AxBench by *Wu et al.* (2025)
> While **AxBench** (Wu et al., 2025) offers a valuable methodology with insightful findings, it appears to rely on *assumptions about neural embeddings that differ fundamentally from ours*.
>
> As you noted, AxBench formulates concept detection as a **binary classification problem** (Section 3.2 in Wu et al., 2025), closely resembling traditional supervised probing. Their evaluation compares **token-level predicted labels** (indicating the presence or absence of a concept in each token) against ground-truth annotations.
> This binary formulation makes direct comparison with our approach difficult.
>
> AxBench implicitly assumes a **token-wise one-to-one mapping** between embeddings and concepts, probing each token embedding independently.
> In contrast, our method focuses on the **final sequence embedding**, which we assume can encode **multiple input-related and output-related concepts simultaneously**.
> Under AxBench’s framing, concept detection is constrained to verifying whether an embedding contains a **single, atomic concept**, effectively assuming that neural representations encode only one feature/concept per token.
> This assumption rules out the detection of **compositional representations**, which our method explicitly models (see Equation 2).
>
> By contrast, our work is grounded in the assumption of **concept-oriented compositionality via superposition** (Bricken et al., 2023), where overlapping neuron activations can encode **more latent features than available dimensions** (i.e., subspaces). Inspired by the sparsity constraint used in SAEs, we leverage key properties of **Vector Symbolic Architectures (VSA)** and **hypervector algebra** to disentangle these overlapping subspaces within the LLM embedding space, operating under the premise that LLM embeddings encode **multiple latent features/concepts** simultaneously.
>
> In summary, because **AxBench operates under fundamentally different assumptions about *how concepts are represented* in neural embeddings**, it is *not directly comparable* to our experimental setting.
> Below, we provide our best attempt at a visual comparison given the textual input *"Denmark : krone = Mexico:"*:
> 1. AxBench's non-compositional assumption on concepts:
> **Token embedding** | *Encoded concept*
> ----| ----
> $$ e(\text{"Denmark"}) $$ | $$\mapsto \phi_\text{Denmark}$$
> $$ e(\text{"krone"}) $$ | $$ \mapsto\phi_\text{krone}$$
> $$ e(\text{"Mexico"}) $$ | $$ \mapsto\phi_\text{Mexico}$$
> $$ e(\text{":"})$$ | $$\mapsto \phi_\text{peso}$$
>
> 2. Our assumption on compositionality:
> **Token embedding** | *Encoded concepts*
> ----| ----
> $$ e(\text{":"}) $$ | $$\mapsto (\phi_\text{Denmark} \odot \phi_\text{krone}) + (\phi_\text{Mexico} \odot \phi_\text{peso})$$
> ---
> # Authors' final remarks
> Thank you for all the *constructive* feedback. We genuinely appreciate the opportunity to clarify these key aspects during the rebuttal, as well as the thoughtful attention you and the other reviewers have given our submission.
>
> We believe that the revisions prompted by your comments, and those of the other reviewers, have substantially strengthened the manuscript. We are **currently preparing a revised 10-page version** that incorporates as many clarifications as possible to address the full set of reviewer concerns, and we plan to upload it **before the end of the rebuttal period**.
>
> This revised version includes, for example, a *entirely rewritten introduction and related work section* to better articulate the motivation and advantages of a VSA-based methodology; the *relocation of our experimental comparison* from Appendix G.1 to the main text; additional *clarifications on LLM metrics*; further *explanation of Table 1*; and the *unidirectional transformation* mentioned in the limitation.
>
> We hope that these clarifications, and the forthcoming **revised version**, will support your *final evaluation* of our submission.

---

### Author Response · Authors · 2025-11-28
**Authors' Response to Reviewers' Feedback and Improvements of the revised version**

We sincerely appreciate the *thoughtful* and *constructive feedback* from all reviewers during rebuttal.
They have all recognized that our work:
1. applies **Vector Symbolic Architectures for probing**
   - in a *novel* (gn8b, s5VU, ufU2, dtKf), and
   - *interesting* (s5VU, dtKf) manner;
2. proposes a thorough and *well-motivated approach* for:
   -  ***improving* existing interpretability methods** (ufU2), while
   - *addressing* an **important problem in interpretability** (s5VU);
3. **demonstrates the *generalizability* of the proposed approach** by testing:
   - *two distinct experimental settings* (ufU2), and
   - *across tasks* of varying complexity,
   - *model sizes*, and
   - *families* (gn8b);
4.  introduces also a *novel approach* for **reducing the dimensionality of the full LLM hidden space** (dtKf);
5. presents a ***rich* amount of content** (gn8b):
   -  in a *logically structured* manner (dtKf), while
   - including *thorough experimental details in the appendix* (gn8b, ufU2).
-------
We believe that the revisions that address reviewer feedback have *massively improved the manuscript* 💪🏻

We have **uploaded a *revised version*** and, thanks also to the additional page, have **enhanced the following aspects**, which are highlighted in *red* and *blue* in the manuscript.

## **1)** Clarification of the *problem statement* 🕵🏻
To better frame the scope and objectives of our methodology (dtKf, gn8b, s5VU), we have ***fully* rewritten**:
1. the **introduction** section (Section 1)
2. the **related work** section (Section 2), and
3. the **abstract**.

They now explicitly include our *research questions* and the *underlying problem statement*.  Our findings have now *better linked to our experimental research questions* in Section 5.2 and Section 6.

Additionally, we clarified the distinct purposes of the *analogy-style dataset versus the QA-style dataset* (dtKf) throughout the paper.

## **2)** Better framing of the *advantages of VSA-based methodology* 🏅
In the *revised introduction* and *related work sections*, we provide:
- a *clearer* **rationale for using VSA probing** and,
- highlight **its characteristic overcoming the key constraints of prior methods** (dtKf, gn8b, s5VU).

We also completely **revised the conclusion section** to more clearly *highlight the main contributions of our work*.

## **3)** *Experimental comparison* moved to the main text 🧑🏻‍🔬
We **relocated our experimental comparison** from Appendix G.1 to the main text (ufU2) as Section 5.3 to:
   - better contextualize our findings (dtKf, s5VU) and
   - emphasize *shallower insights from the conventional logit-based analysis*.

In addition:
1. We also explicitly **connected our DLA-based comparison to the *Logit Lens*** method proposed by Nostalgebraist (2020);
2. **Table 8 has been moved into the main text** (now as Table 2), and
3. **DLA-based scores** are now **integrated into Figure 2**.
## **4)** Clarification of *LLM metrics* 🤔
We have *enhanced the reporting* of LLM metrics across the paper.
As part of this, **we have discarded our original partial-matching strategy** (s5VU, dtKf) for our experiments in Section 5 and recalculated scores ($0.5 \mapsto 1$) to **address concerns with this *non-standard* practice**, as our findings remain unaffected.

Additionally, we have **separated analogy-based** (Section 5) and **QA-based settings** (former Section 5.3, now Section 6) to *reduce confusion* between the metrics of these two distinct settings.
## **5)** Clarification of information exhibited in *Table 1* 🧐
We have included the **description of the results in Table 1** (gn8b, s5VU, ufU2) in Section 5.2.

## **6)** *Unidirectional transformation* listed as a limitation 😅
In Section 7, we identify the **reliance on unidirectional transformation as a key limitation** of our proposed method,
   - noting that **it currently hinders *causal validation*** (gn8b, s5VU).

We are currently developing a *bidirectional transformation* using an encoder-decoder architecture, which **will be the focus of a separate publication**, as anticipated in the rebuttal.
## **7)** Improved *reporting of statistical variance* 📊
We now report statistical variance **with 95\% confidence intervals instead of *standard deviations*** (dtKf) for greater clarity in Section 6 and Appendix F.

## **8)** Improved *presentation of experimental settings* 📃
To improve the manuscript’s logical flow (gn8b), the validation of our full pipeline within the QA setup has been **moved from Section 5.3 to its own dedicated section** (Section 6). This reorganization allows for **clearer distinction** between their *experimental setups* and better *presentation of their distinct purposes*, *inputs*, *metrics* (dtKf), and *processes* (s5VU).

## **9)** Correction of some *typos* in the manuscript ✍🏻
Typos highlighted by reviewer gn8b have been **corrected**. Hoping to not have created new ones.

-------
*"You will see miracles if you believe"*

---

### Meta-Review · Area_Chair_aKTg · 2026-01-12

**Summary:**

There is strong consensus on the value of applying VSAs to neural probing—and the intersection with interpretability is viewed as a high-potential direction. Also the paper has validated the method across a wide variety of tasks, model sizes, and different model families in the main paper/during rebuttal. However, the current manuscript requires a major restructuring to 1) clarify its contributions. Although the authors provided explanation to the clarify in the rebuttal comments, these are not present in the manuscript, and reviewers felt that incorporating them would require major rewrite. 2) integrate appropriate baselines into the main text, and rigorously validate its claims against standard methods.

**Reviewer Concerns:**

Most of the concerns are addressed during rebuttal, and concerns not addressed by rebuttal:

1) A fundamental disagreement persists regarding baselines after rebuttal.
2) Multiple reviewers found the paper dense and difficult to follow.
3) Causal Validation (Steering): Reviewers noted the lack of causal validation (steering) limits claims about whether extracted concepts are actually used by the model. The authors acknowledged this as a limitation of their unidirectional encoder , but Reviewer remained skeptical of the validation provided.

**Reviewer Scores:**

na

---

### Decision · Program_Chairs · 2026-01-26

Reject